# Stochastic Optimal Control Matching

**Carles Domingo-Enrich**
NYU & FAIR, Meta
cd2754@nyu.edu

**Jiequn Han**
Flatiron Institute
jhan@flatironinstitute.org

**Brandon Amos**
FAIR, Meta
bda@meta.com

**Joan Bruna**
NYU & Flatiron Institute
bruna@cims.nyu.edu

**Ricky T. Q. Chen**
FAIR, Meta
rtqichen@meta.com

## Abstract

Stochastic optimal control, which has the goal of driving the behavior of noisy systems, is broadly applicable in science, engineering and artificial intelligence. Our work introduces Stochastic Optimal Control Matching (SOCM), a novel Iterative Diffusion Optimization (IDO) technique for stochastic optimal control that stems from the same philosophy as the conditional score matching loss for diffusion models. That is, the control is learned via a least squares problem by trying to fit a matching vector field. The training loss, which is closely connected to the cross-entropy loss, is optimized with respect to both the control function and a family of reparameterization matrices which appear in the matching vector field. The optimization with respect to the reparameterization matrices aims at minimizing the variance of the matching vector field. Experimentally, our algorithm achieves lower error than all the existing IDO techniques for stochastic optimal control for three out of four control problems, in some cases by an order of magnitude. The key idea underlying SOCM is the path-wise reparameterization trick, a novel technique that may be of independent interest.

## 1 Introduction

Stochastic optimal control aims to drive the behavior of a noisy system in order to minimize a given cost. It has myriad applications in science and engineering: examples include the simulation of rare events in molecular dynamics [37, 36, 85, 41], finance and economics [63, 25], stochastic filtering and data assimilation [58, 68], nonconvex optimization [19], sampling [9], power systems and energy markets [8, 66], and robotics [77, 32]. Stochastic optimal has also been impactful in fields such as mean-field games [17], optimal transport [80, 81], backward stochastic differential equations (BSDEs) [14] and large deviations [24]. Recently, it has been the basis of algorithms to sample from unnormalized densities [84, 79, 9, 71].

For continuous-time problems with low-dimensional state spaces, the standard approach to learn the optimal control is to solve the Hamilton-Jacobi-Bellman (HJB) partial differential equation (PDE) by gridding the space and using classical numerical methods. For high-dimensional problems, a large number of works parameterize the control using a neural network and train it applying a stochastic optimization algorithm on a loss function. These methods are known as *Iterative Diffusion Optimization* (IDO) techniques [59] (see Subsec. 2.2).

It is convenient to draw an analogy between stochastic optimal control and *continuous normalizing flows* (CNFs), which are a generative modeling technique where samples are generated by solving an ordinary differential equation (ODE) for which the vector field has been learned, initialized at a Gaussian sample. CNFs were introduced by [20] (building on top of Rezende and Mohamed [70]),

38th Conference on Neural Information Processing Systems (NeurIPS 2024).

and training them is similar to solving control problems because in both cases one needs to learn high-dimensional vector fields using neural networks, in continuous time.

The first algorithm developed to train normalizing flows was based on maximizing the likelihood of the generated samples [20, Sec. 4]. Obtaining the gradient of the maximum likelihood loss with respect to the vector field parameters requires backpropagating through the computation of the ODE trajectory, or equivalently, solving the *adjoint* ODE in parallel to the original ODE. Maximum likelihood CNFs (ML-CNFs) were superseded by diffusion models [75, 40, 76] and flow-matching, a.k.a. stochastic interpolant, methods [55, 1, 65, 2], which are currently the preferred algorithms to train CNFs. Aside from architectural improvements such as the UNet [73], a potential reason for the success of diffusion and flow matching models is that their *functional landscape* is convex, unlike for ML-CNFs. Namely, vector fields are learned by solving least squares regression problems where the goal is to fit a random matching vector field. Convex functional landscapes in combination with overparameterized models and moderate gradient variance can yield very stable training dynamics and help achieve low error.

Returning to stochastic optimal control, one of the best-performing IDO techniques amounts to choosing the control objective (equation 1) as the training loss (see (12)). As in ML-CNFs, computing the gradient of this loss requires backpropagating through the computation of the trajectories of the SDE (2), or equivalently, using an adjoint method. The functional landscape of the loss is highly non-convex, and the method is prone to unstable training (see green curve in the bottom right plot of Figure 3). In light of this, a natural idea is to develop the analog of diffusion model losses for the stochastic optimal control problem, to obtain more stable training and lower error, and this is what we set out to do in our work. Our contributions are as follows:

- We introduce Stochastic Optimal Control Matching (SOCM), a novel IDO algorithm in which the control is learned by solving a least-squares regression problem where the goal is to fit a random *matching vector field* which depends on a family of *reparameterization matrices* that are also optimized.
- We derive a bias-variance decomposition of the SOCM loss (Prop. 2). The bias term is equal to an existing IDO loss: the *cross-entropy loss*, which shows that both algorithms have the same landscape in expectation. However, SOCM has an extra flexibility in the choice of reparameterization matrices, which affect only the variance. Hence, we propose optimizing the reparameterization matrices to reduce the variance of the SOCM objective.
- The key idea that underlies the SOCM algorithm is the *path-wise reparameterization trick* (Prop. 1), which is a novel technique for estimating gradients of an expectation of a functional of a random process with respect to its initial value. It is of independent interest and may be more generally applicable outside of the settings considered in this paper.
- We perform experiments on four different settings where we have access to the ground-truth control. For three of these, SOCM obtains a lower $L^2$ error with respect to the ground-truth control than all the existing IDO techniques, with around $10\times$ lower error than competing methods in some instances.

## 2 Framework

### 2.1 Setup and Preliminaries

Let $(\Omega, \mathcal{F}, (\mathcal{F}_t)_{t \geq 0}, \mathcal{P})$ be a fixed filtered probability space on which is defined a Brownian motion $B = (B_t)_{t \geq 0}$. We consider the control-affine problem

$$\min_{u \in \mathcal{U}} \mathbb{E}\Big[ \int_0^T \big( \tfrac{1}{2}\|u(X_t^u, t)\|^2 + f(X_t^u, t) \big) \, \mathrm{d}t + g(X_T^u) \Big], \tag{1}$$

$$\text{s.t. } \mathrm{d}X_t^u = (b(X_t^u, t) + \sigma(t)u(X_t^u, t)) \, \mathrm{d}t + \sqrt{\lambda}\sigma(t)\mathrm{d}B_t, \ X_0^u \sim p_0 \tag{2}$$

and where $X_t^u \in \mathbb{R}^d$ is the state, $u : \mathbb{R}^d \times [0, T] \to \mathbb{R}^d$ is the feedback control and belongs to the set of admissible controls $\mathcal{U}$, $f : \mathbb{R}^d \times [0, T] \to \mathbb{R}$ is the state cost, $g : \mathbb{R}^d \to \mathbb{R}$ is the terminal cost, $b : \mathbb{R}^d \times [0, T] \to \mathbb{R}^d$ is the base drift, and $\sigma : [0, T] \to \mathbb{R}^{d \times d}$ is the invertible diffusion coefficient and $\lambda \in (0, +\infty)$ is the noise level. In App. A we formally define the set $\mathcal{U}$ of admissible controls and describe the regularity assumptions needed on the control functions. In the remainder of the section we introduce relevant concepts in stochastic optimal control; we provide the most relevant proofs in App. B and refer the reader to Oksendal [60, Chap. 11] and Nüsken and Richter [59, Sec. 2] for a similar, more extensive treatment.

**Cost functional and value function**  The *cost functional* for the control $u$, point $x$ and time $t$ is defined as $J(u; x, t) := \mathbb{E}\big[\int_t^T \big(\frac{1}{2}\|u_s(X_s^u)\|^2 + f_s(X_s^u)\big)\,\mathrm{d}t + g(X_T^u)\big|X_t^u = x\big]$. That is, the cost functional is the expected value of the control objective restricted to the times $[t, T]$ with the initial value $x$ at time $t$. The *value function* or *optimal cost-to-go* at $(x, t)$ is defined as the minimum value of the cost functional across all possible controls:

$$V(x, t) := \inf_{u \in \mathcal{U}} J(u; x, t). \tag{3}$$

**Hamilton-Jacobi-Bellman equation and optimal control**  If we define the infinitesimal generator $L := \frac{\lambda}{2} \sum_{i,j=1}^d (\sigma\sigma^\top)_{ij}(t)\partial_{x_i}\partial_{x_j} + \sum_{i=1}^d b_i(x, t)\partial_{x_i}$, the value function solves the following Hamilton-Jacobi-Bellman (HJB) partial differential equation:

$$(\partial_t + L)V(x, t) - \tfrac{1}{2}\|(\sigma^\top \nabla V)(x, t)\|^2 + f(x, t) = 0, \qquad V(x, T) = g(x). \tag{4}$$

The *verification theorem* [62, Sec. 2.3] states that if a function $V$ solves the HJB equation above and has certain regularity conditions, then $V$ is the value function (3) of the problem (1)-(2). An implication of the verification theorem is that for every $u \in \mathcal{U}$,

$$V(x, t) + \mathbb{E}\big[\tfrac{1}{2}\int_t^T \|\sigma^\top \nabla V + u\|^2(X_s^u, s)\,\mathrm{d}s \,\big|\, X_t^u = x\big] = J(u, x, t). \tag{5}$$

In particular, this implies that the unique optimal control is given in terms of the value function as $u^*(x, t) = -\sigma(t)^\top \nabla V(x, t)$. Equation (5) can be deduced by integrating the HJB equation (4) over $[t, T]$, and taking the conditional expectation with respect to $X_t^u = x$. We include the proof of (5) in App. B for completeness.

**A pair of forward and backward SDEs (FBSDEs)**  Consider the pair of SDEs

$$\mathrm{d}X_t = b(X_t, t)\,\mathrm{d}t + \sqrt{\lambda}\sigma(t)\mathrm{d}B_t, \qquad X_0 \sim p_0, \tag{6}$$

$$\mathrm{d}Y_t = (-f(X_t, t) + \tfrac{1}{2}\|Z_t\|^2)\,\mathrm{d}t + \sqrt{\lambda}\langle Z_t, \mathrm{d}B_t\rangle, \qquad Y_T = g(X_T). \tag{7}$$

where $Y : \Omega \times [0, T] \to \mathbb{R}$ and $Z : \Omega \times [0, T] \to \mathbb{R}^d$ are progressively measurable [1] random processes. It turns out that $Y_t$ and $Z_t$ defined as $Y_t := V(X_t, t)$ and $Z_t := \sigma(t)^\top \nabla V(X_t, t) = -u^*(X_t, t)$ satisfy (7). We include the proof in App. B for completeness.

**An analytic expression for the value function**  From the forward-backward equations (6)-(7), one can derive a closed-form expression for the value function $V$:

$$V(x, t) = -\lambda \log \mathbb{E}\big[\exp\big(-\lambda^{-1}\int_t^T f(X_s, s)\,\mathrm{d}s - \lambda^{-1}g(X_T)\big)\big|X_t = x\big], \tag{8}$$

where $X_t$ is the solution of the uncontrolled SDE (6). This is a classical result, but we still include its proof in App. B. Given that $u^*(x, t) = -\sigma(t)^\top \nabla V(x, t)$, an immediate, yet important, consequence of (8) is the following path-integral representation of the optimal control:

$$u^*(x, t) = \lambda\sigma(t)^\top \nabla_x \log \mathbb{E}\big[\exp\big(-\lambda^{-1}\int_t^T f(X_s, s)\,\mathrm{d}s - \lambda^{-1}g(X_T)\big)\big|X_t = x\big]. \tag{9}$$

Remark this equation involves the gradient of logarithm of a conditional expectation, which is reminiscent of the vector fields that are learned when training diffusion models. For example, the target vector field for variance-exploding score-based diffusion loss [76] can be expressed as $\nabla_x \log p_t(x) = \nabla_x \log \mathbb{E}_{Y \sim p_{\mathrm{data}}}\big[\frac{\exp(-\|x-Y\|^2/(2\sigma_t^2))}{(2\pi\sigma_t^2)^{d/2}}\big]$. Note, however, that in (9) the gradient is taken with respect to the initial condition of the process, which requires the development of novel techniques.

**Conditioned diffusions**  Let $\mathcal{C} = C([0, T]; \mathbb{R}^d)$ be the Wiener space of continuous functions from $[0, T]$ to $\mathbb{R}^d$ equipped with the supremum norm, and let $\mathcal{P}(\mathcal{C})$ be the space of Borel probability measures over $\mathcal{C}$. For each control $u \in \mathcal{U}$, the controlled process in equation (2) induces a probability measure in $\mathcal{P}(\mathcal{C})$, as the law of the paths $X_t^u$, which we refer to as $\mathbb{P}^u$. We let $\mathbb{P}$ be the probability measure induced by the uncontrolled process (6), and define the *work functional*

$$\mathcal{W}(X, t) := \int_t^T f(X_s, s)\,\mathrm{d}s + g(X_T). \tag{10}$$

---

[1] Being progressively measurable is a strictly stronger property than the notion of being a process adapted to the filtration $\mathcal{F}_t$ of $B_t$ (see [50]).

It turns out (Lemma 2 in App. B) that the Radon-Nikodym derivative $\frac{d\mathbb{P}^{u^*}}{d\mathbb{P}}$ satisfies $\frac{d\mathbb{P}^{u^*}}{d\mathbb{P}}(X) = \exp\left(\lambda^{-1}\left(V(X_0, 0) - \mathcal{W}(X, 0)\right)\right)$. Also, a straight-forward application of the Girsanov theorem for SDEs (Cor. 1) shows that

$$\frac{d\mathbb{P}^u}{d\mathbb{P}^{u^*}}(X^{u^*}) = \exp\left(-\lambda^{-1/2}\int_0^T \langle u^*(X_t^{u^*}, t) - u(X_t^{u^*}, t), dB_t\rangle - \frac{\lambda^{-1}}{2}\int_0^T \|u^*(X_t^{u^*}, t) - u(X_t^{u^*}, t)\|^2\, dt\right), \tag{11}$$

which means that the only control $u \in \mathcal{U}$ such that $\mathbb{P}^u = \mathbb{P}^{u^*}$ is the optimal control itself.

## 2.2 Existing approaches and related work

**Low-dimensional case: solving the HJB equation**   For low-dimensional control problems ($d \leq 3$), it is possible to grid the domain and use a numerical PDE solver to find a solution to the HJB equation (4). The main approaches include *finite difference methods* [11, 57, 4], which approximate the derivatives and gradients of the value function using finite differences, *finite element methods* [47], which involve restricting the solution to domain-dependent function spaces, and semi-Lagrangian schemes [21, 13, 12], which trace back characteristics and have better stability than finite difference methods. See Greif [33] for an overview on these techniques, and Baňas et al. [4] for a comparison between them. Hutzenthaler et al. [44] introduced the multilevel Picard method, which leverages the Feynman-Kac and the Bismut-Elworthy-Li formulas to beat the curse of dimensionality in some settings [6, 46, 45, 43].

**High dimensional methods leveraging FBSDEs**   The FBSDE formulation in equations (6)-(7) has given rise to multiple methods to learn controls. One such approach is *least-squares Monte Carlo* (see Pham [63, Chapter 3] and Gobet [28] for an introduction, and Gobet et al. [30], Zhang et al. [83] for an extensive analysis), where trajectories from the forward process (6) are sampled, and then regression problems are solved backwards in time to estimate the expected future cost in the spirit of dynamic programming. A second method that exploits FBSDEs was proposed by E et al. [22], Han et al. [35]. They parameterize the control using a neural network $u_\theta$, and use stochastic gradient algorithms to minimize the loss $\mathcal{L}(u_\theta, y_0) = \mathbb{E}[(Y_T(y_0, u_\theta) - g(X_T))^2]$, where $Y_T(y_0, u_\theta)$ is the process in (7) with initial condition $y_0$ and control $u_\theta$. This algorithm can be seen as a shooting method, where the initial condition and the control are learned to match the terminal condition. Multiple recent works have combined neural networks with FBSDE Monte Carlo methods for parabolic and elliptic PDEs [5, 18, 86], control [7, 39], multi-agent games [34, 15, 16]; see [23] for a more comprehensive review.

Many of the methods referenced above and some additional ones can be seen from a common perspective using controlled diffusions. As observed in equation (11), the key idea is that learning the optimal control is equivalent to finding a control $u$ such that the induced probability measure $\mathbb{P}^u$ on paths is equal to the probability measure $\mathbb{P}^{u^*}$ for the optimal control. In the paragraphs below we cover several loss that fall into this framework. All the losses below can be optimized using a common algorithmic framework, which we describe in Algorithm 1. For more details, we refer the reader to Nüsken and Richter [59], which introduced this perspective and named such methods *Iterative Diffusion Optimization* (IDO) techniques. For simplicity, we introduce the losses for the setting in which the initial distribution $p_0$ is concentrated at a single point $x_{\text{init}}$; we cover the general setting in App. B.

**The relative entropy loss and the adjoint method**   The relative entropy loss is defined as the Kullback-Leibler divergence between $\mathbb{P}^u$ and $\mathbb{P}^{u^*}$: $\mathbb{E}_{\mathbb{P}^u}[\log \frac{d\mathbb{P}^u}{d\mathbb{P}^{u^*}}]$. Upon removing constant terms and factors, this loss is equivalent to (see Lemma 3 in App. B):

$$\mathcal{L}_{\text{Adj}}(u) := \mathbb{E}\left[\int_0^T \left(\frac{1}{2}\|u(X_t^u, t)\|^2 + f(X_t^u, t)\right) dt + g(X_T^u)\right]. \tag{12}$$

This is exactly the control objective in (1). This fact has been studied extensively [10, 31, 36, 48, 67]. Hence, the relative entropy loss is very natural and widely used; see Onken et al. [61], Zhang and Chen [84] for examples on multiagent systems and sampling.

Solving optimization problems of the form (12) has a long history that dates back to Pontryagin [64]. Note that $\mathcal{L}_{\text{Adj}}(u)$ depends on $u$ both explicitly, and implicitly through the process $X^u$. To compute the gradient $\nabla_\theta \hat{\mathcal{L}}_{\text{Adj}}(u_{\theta_n})$ of a Monte Carlo approximation $\hat{\mathcal{L}}_{\text{Adj}}(u_{\theta_n})$ of $\mathcal{L}_{\text{Adj}}(u_{\theta_n})$ as required by Algorithm 1, we need to backpropagate through the simulation of the trajectories, which is why

**Algorithm 1** Iterative Diffusion Optimization (IDO) algorithms for stochastic optimal control
---
**Input:** State cost $f(x,t)$, terminal cost $g(x)$, diffusion coeff. $\sigma(t)$, base drift $b(x,t)$, noise level $\lambda$, number of iterations $N$, batch size $m$, number of time steps $K$, initial control parameters $\theta_0$, loss $\mathcal{L} \in \{\mathcal{L}_{\mathrm{Adj}}(12), \mathcal{L}_{\mathrm{CE}}(13), \mathcal{L}_{\mathrm{Var}_v}(16), \mathcal{L}_{\mathrm{Var}_v}^{\log}(17), \mathcal{L}_{\mathrm{Mom}_v}(18)\}$

1 **for** $n \in \{0, \dots, N-1\}$ **do**
2      Simulate $m$ trajectories of the process $X^v$ controlled by $v = u_{\theta_n}$, e.g., using Euler-Maruyama updates
3      **if** $\mathcal{L} \neq \mathcal{L}_{\mathrm{Adj}}$ **then** detach the $m$ trajectories from the computational graph, so that gradients do not backpropagate;
4      Using the $m$ trajectories, compute an $m$-sample Monte Carlo approximation $\hat{\mathcal{L}}(u_{\theta_n})$ of the loss $\mathcal{L}(u_{\theta_n})$
5      Compute the gradients $\nabla_\theta \hat{\mathcal{L}}(u_{\theta_n})$ of $\hat{\mathcal{L}}(u_{\theta_n})$ w.r.t. $\theta_n$
6      Obtain $\theta_{n+1}$ with via an Adam update on $\theta_n$ (or another stochastic algorithm)
7 **end**
**Output:** Learned control $u_{\theta_N}$
---

we do *not* detach them from the computational graph. One can alternatively compute the gradient $\nabla_\theta \hat{\mathcal{L}}_{\mathrm{Adj}}(u_{\theta_n})$ by explicitly solving an ODE, a technique known as the *adjoint method*. The adjoint method was introduced by Pontryagin [64], popularized in deep learning by Chen et al. [20], and further developed for SDEs in Li et al. [54].

**The cross-entropy loss** The cross-entropy loss is defined as the Kullback-Leibler divergence between $\mathbb{P}^{u^*}$ and $\mathbb{P}^u$, i.e., flipping the order of the two measures: $\mathbb{E}_{\mathbb{P}^{u^*}}[\log \frac{d\mathbb{P}^{u^*}}{d\mathbb{P}^u}]$. For an arbitrary $v \in \mathcal{U}$, this loss is equivalent to the following one (see Prop. 3(i) in App. B):

$$\mathcal{L}_{\mathrm{CE}}(u) := \mathbb{E}\Big[\big(-\lambda^{-1/2}\int_0^T \langle u(X_t^v, t), dB_t\rangle - \lambda^{-1}\int_0^T \langle u(X_t^v, t), v(X_t^v, t)\rangle\, dt + \tfrac{\lambda^{-1}}{2}\int_0^T \|u(X_t^v, t)\|^2\, dt\big)$$
$$\times \exp\big(-\lambda^{-1}\mathcal{W}(X^v, 0) - \lambda^{-1/2}\int_0^T \langle v(X_t^v, t), dB_t\rangle - \tfrac{\lambda^{-1}}{2}\int_0^T \|v(X_t^v, t)\|^2\, dt\big)\Big]. \tag{13}$$

The cross-entropy loss has a rich literature [38, 49, 74, 85] and has been recently used in applications such as molecular dynamics [41]. Furthermore, we note that the cross-entropy loss can be significantly simplified and written in terms of the unnormalized $L^2$ error of the control $u$ with respect to the optimal control $u^*$:

$$\mathcal{L}_{\mathrm{CE}}(u) = \tfrac{\lambda^{-1}}{2}\mathbb{E}\Big[\int_0^T \|u^*(X_t^{u^*}, t) - u(X_t^{u^*}, t)\|^2\, dt \times \exp\big(-\lambda^{-1}V(X_0^{u^*}, 0)\big)\Big]. \tag{14}$$

This characterization, which is proven in Prop. 3(ii) in App. B, is relevant for us because a similar one can be written for the loss that we propose (see Prop. 2).

**Variance and log-variance losses** For an arbitrary $v \in \mathcal{U}$, the *variance* and the *log-variance losses* are defined as $\tilde{\mathcal{L}}_{\mathrm{Var}_v}(u) = \mathrm{Var}_{\mathbb{P}^v}(\frac{d\mathbb{P}^{u^*}}{d\mathbb{P}^u})$ and $\tilde{\mathcal{L}}_{\mathrm{Var}_v}^{\log}(u) = \mathrm{Var}_{\mathbb{P}^v}(\log \frac{d\mathbb{P}^{u^*}}{d\mathbb{P}^u})$ whenever $\mathbb{E}_{\mathbb{P}^v}|\frac{d\mathbb{P}^{u^*}}{d\mathbb{P}^u}| < +\infty$ and $\mathbb{E}_{\mathbb{P}^v}|\log \frac{d\mathbb{P}^{u^*}}{d\mathbb{P}^u}| < +\infty$, respectively. Define

$$\tilde{Y}_T^{u,v} = -\lambda^{-1}\int_0^T \langle u(X_t^v, t), v(X_t^v, t)\rangle\, dt$$
$$- \lambda^{-1}\int_0^T f(X_t^v, t)\, dt - \lambda^{-1/2}\int_0^T \langle u(X_t^v, t), dB_t\rangle \tag{15}$$
$$+ \tfrac{\lambda^{-1}}{2}\int_0^T \|u(X_t^v, t)\|^2\, dt.$$

Then, $\tilde{\mathcal{L}}_{\mathrm{Var}_v}$ and $\tilde{\mathcal{L}}_{\mathrm{Var}_v}^{\log}$ are equivalent, respectively, to the following losses (see Lemma 4):

$$\mathcal{L}_{\mathrm{Var}_v}(u) := \mathrm{Var}\big(\exp\big(\tilde{Y}_T^{u,v} - \lambda^{-1}g(X_T^v)\big)\big), \tag{16}$$
$$\mathcal{L}_{\mathrm{Var}_v}^{\log}(u) := \mathrm{Var}\big(\tilde{Y}_T^{u,v} - \lambda^{-1}g(X_T^v)\big), \tag{17}$$

The variance and log-variance losses were introduced by Nüsken and Richter [59]. Unlike for the cross-entropy loss, the choice of the control $v$ does lead to different losses. When using $\mathcal{L}_{\mathrm{Var}_v}$ or $\mathcal{L}_{\mathrm{Var}_v}^{\log}$ in Algorithm 1, the variance is computed across the $m$ trajectories in each batch.

**Moment loss** For an arbitrary $v \in \mathcal{U}$, the moment loss is defined as

$$\mathcal{L}_{\mathrm{Mom}_v}(u, y_0) = \mathbb{E}[(\tilde{Y}_T^{u,v} + y_0 - \lambda^{-1}g(X_T^v))^2], \tag{18}$$

where $\tilde{Y}_T^{u,v}$ is defined in (15). Note the similarity with the log-variance loss (17); the optimal value of $y_0$ for a fixed $u$ is $y_0^* = \mathbb{E}[\lambda^{-1} g(X_T^v) - \tilde{Y}_T^{u,v}]$, and plugging this into (18) yields exactly the log-variance loss. The moment loss was introduced by Hartmann et al. [39, Section III.B], and it is a generalization of the FBSDE method pioneered by E et al. [22], Han et al. [35] and referenced earlier in this subsection, which corresponds to setting $v = 0$.

## 3 Stochastic Optimal Control Matching

In this section we present our loss, *Stochastic Optimal Control Matching* (SOCM). The corresponding method, which we describe in Algorithm 2, falls into the class of IDO techniques described in Subsec. 2.2. The general idea is to leverage the analytic expression of $u^*$ in (9) to write a least squares loss for $u$, and the main challenge is to reexpress the gradient of a conditional expectation with respect to the initial condition of the process. We do that using a novel technique which introduces certain arbitrary matrix-valued functions $M_t$, that we also optimize.

**Theorem 1** (SOCM loss). *For each $t \in [0, T]$, let $M_t : [t, T] \to \mathbb{R}^{d \times d}$ be an arbitrary matrix-valued differentiable function such that $M_t(t) = \mathrm{Id}$. Let $v \in \mathcal{U}$ be an arbitrary control. Let $\mathcal{L}_{\mathrm{SOCM}} : L^2(\mathbb{R}^d \times [0, T]; \mathbb{R}^d) \times L^2([0, T]^2; \mathbb{R}^{d \times d}) \to \mathbb{R}$ be the loss function defined as*

$$\mathcal{L}_{\mathrm{SOCM}}(u, M) := \mathbb{E}\big[\tfrac{1}{T} \int_0^T \big\| u(X_t^v, t) - w(t, v, X^v, B, M_t) \big\|^2 \, \mathrm{d}t \times \alpha(v, X^v, B)\big], \quad (19)$$

*where $X^v$ is the process controlled by $v$ (i.e., $dX_t^v = (b(X_t^v, t) + \sigma(t)v(X_t^v, t)) \, dt + \sqrt{\lambda}\sigma(t) \, dB_t$ and $X_0^v \sim p_0$), and*

$$
\begin{aligned}
w(t, v, X^v, B, M_t) = \sigma(t)^\top \big( &- \int_t^T M_t(s) \nabla_x f(X_s^v, s) \, \mathrm{d}s - M_t(T) \nabla g(X_T^v) \\
&+ \int_t^T (M_t(s) \nabla_x b(X_s^v, s) - \partial_s M_t(s))(\sigma^{-1})^\top(s) v(X_s^v, s) \, \mathrm{d}s \\
&+ \lambda^{1/2} \int_t^T (M_t(s) \nabla_x b(X_s^v, s) - \partial_s M_t(s))(\sigma^{-1})^\top(s) \mathrm{d}B_s \big), \\
\alpha(v, X^v, B) = \exp\big( &- \lambda^{-1} \int_0^T f(X_t^v, t) \, \mathrm{d}s - \lambda^{-1} g(X_T^v) \\
&- \lambda^{-1/2} \int_0^T \langle v(X_t^v, t), \mathrm{d}B_t \rangle - \tfrac{\lambda^{-1}}{2} \int_0^T \| v(X_t^v, t) \|^2 \, \mathrm{d}t \big).
\end{aligned}
\tag{20}
$$

*$\mathcal{L}_{\mathrm{SOCM}}$ has a unique optimum $(u^*, M^*)$, where $u^*$ is the optimal control.*

We refer to $M = (M_t)_{t \in [0,T]}$ as the family of *reparametrization matrices*, to the random vector field $w$ as the *matching vector field*, and to $\alpha$ as the *importance weight*. We present a proof sketch of Thm. 1; the full proofs for all the results in this section are in App. C.

**Proof sketch of Thm. 1** Let $X$ be the uncontrolled process (6). Consider the loss

$$
\begin{aligned}
\tilde{\mathcal{L}}(u) &= \mathbb{E}\big[\tfrac{1}{T} \int_0^T \| u(X_t, t) - u^*(X_t, t) \|^2 \, \mathrm{d}t \, \exp\big(-\lambda^{-1} \int_0^T f(X_t, t) \, \mathrm{d}t - \lambda^{-1} g(X_T))\big] \\
&= \mathbb{E}\big[\tfrac{1}{T} \int_0^T \big( \| u(X_t, t) \|^2 - 2\langle u(X_t, t), u^*(X_t, t) \rangle + \| u^*(X_t, t) \|^2 \big) \, \mathrm{d}t \\
&\quad \times \exp\big(-\lambda^{-1} \int_0^T f(X_t, t) \, \mathrm{d}t - \lambda^{-1} g(X_T))\big].
\end{aligned}
\tag{21}
$$

Clearly, the only optimum of this loss is the optimal control $u^*$. Using the analytic expression of $u^*$ in (9), the cross-term can be rewritten as (see Lemma 5 in App. C):

$$
\begin{aligned}
&\mathbb{E}\big[\tfrac{1}{T} \int_0^T \langle u(X_t, t), u^*(X_t, t) \rangle \, \mathrm{d}t \, \exp\big(-\lambda^{-1} \int_0^T f(X_t, t) \, \mathrm{d}t - \lambda^{-1} g(X_T))\big] \\
&= \lambda \mathbb{E}\big[\tfrac{1}{T} \int_0^T \langle u(X_t, t), \sigma(t)^\top \nabla_x \mathbb{E}\big[\exp\big(-\lambda^{-1} \int_t^T f(X_s, s) \, \mathrm{d}s - \lambda^{-1} g(X_T)) \big| X_t = x \big] \rangle \\
&\quad \times \exp\big(-\lambda^{-1} \int_0^t f(X_s, s) \, \mathrm{d}s) \, \mathrm{d}t\big].
\end{aligned}
\tag{22}
$$

It remains to evaluate the conditional expectation $\nabla_x \mathbb{E}\big[\exp\big(-\lambda^{-1} \int_t^T f(X_s, s) \, \mathrm{d}s - \lambda^{-1} g(X_T)) \big| X_t = x \big]$, which we do by a "reparameterization trick" that shifts the dependence on the initial value $x$ into the stochastic processes—here we introduce a free variable $M_t$—and then applying Girsanov theorem. We coin this the *path-wise reparameterization trick*:

**Proposition 1** (Path-wise reparameterization trick for stochastic optimal control). *For each $t \in [0, T]$, let $M_t : [t, T] \to \mathbb{R}^{d \times d}$ be an arbitrary continuously differentiable function matrix-valued function*

*such that $M_t(t) = \mathrm{Id}$. We have that*

$$
\nabla_x \mathbb{E}\big[\exp\big(-\lambda^{-1}\int_t^T f(X_s,s)\,\mathrm{d}s - \lambda^{-1}g(X_T)\big)\big|X_t = x\big]
$$
$$
= \mathbb{E}\big[\big(-\lambda^{-1}\int_t^T M_t(s)\nabla_x f(X_s,s)\,\mathrm{d}s - \lambda^{-1}M_t(T)\nabla g(X_T)
$$
$$
+ \lambda^{-1/2}\int_t^T (M_t(s)\nabla_x b(X_s,s) - \partial_s M_t(s))(\sigma^{-1})^\top(s)\mathrm{d}B_s\big) \tag{23}
$$
$$
\times \exp\big(-\lambda^{-1}\int_t^T f(X_s,s)\,\mathrm{d}s - \lambda^{-1}g(X_T)\big)\big|X_t = x\big].
$$

We prove a more general form of this result (Prop. 4) in Subsec. C.2 and also provide an intuitive derivation in Subsec. C.3. In the proof of Prop. 4, the reparameterization matrices $M_t$ arise as the gradients of a perturbation to the process $X_t$. Similar ideas can potentially be applied to derive losses for generative modeling. If we plug (23) into the right-hand side of (22), and then this back into (21), and we complete the square, we obtain that for some constant $K$ independent of $u$,

$$
\tilde{\mathcal{L}}(u) = \mathbb{E}\big[\tfrac{1}{T}\int_0^T \big\|u(X_t,t) + \sigma(t)\big(\int_t^T M_t(s)\nabla_x f(X_s,s)\,\mathrm{d}s + M_t(T)\nabla g(X_T)
$$
$$
- \lambda^{1/2}\int_t^T (M_t(s)\nabla_x b(X_s,s) - \partial_s M_t(s))(\sigma^{-1})^\top(s)\mathrm{d}B_s\big)\big\|^2\,\mathrm{d}t
$$
$$
\times \exp\big(-\lambda^{-1}\int_0^T f(X_t,t)\,\mathrm{d}t - \lambda^{-1}g(X_T)\big)\big] + K.
$$

If we perform a change of process from $X$ to $X^v$ applying the Girsanov theorem (Cor. 1 in App. C), we obtain the loss $\mathcal{L}_{\mathrm{SOCM}}(u,M)$. $\qquad\square$

The following result clarifies the role of reparameterization matrices, connecting the SOCM and cross-entropy losses.

**Proposition 2** (Bias-variance decomposition of the SOCM loss). *The SOCM loss decomposes into a bias term that only depends on $u$ and a variance term that only depends on $M$:*

$$
\mathcal{L}_{\mathrm{SOCM}}(u,M) = \underbrace{\mathrm{CondVar}(w;M)}_{\substack{\text{Unnormalized expected} \\ \text{conditional variance of } w}} + \underbrace{\mathbb{E}\big[\tfrac{1}{T}\int_0^T \big\|u(X_t^{u^*},t) - u^*(X_t^{u^*},t)\big\|^2\,\mathrm{d}t\, e^{-\lambda^{-1}V(X_0^{u^*},0)}\big]}_{\text{Unnormalized bias of } u}, \tag{24}
$$

*where*

$$
\mathrm{CondVar}(w;M) = \mathbb{E}\big[\tfrac{1}{T}\int_0^T \big\|w(t,v,X^v,B,M_t) - \underbrace{\tfrac{\mathbb{E}[w(t,v,X^v,B,M_t)\alpha(v,X^v,B)|X_t^v,t]}{\mathbb{E}[\alpha(v,X^v,B)|X_t^v,t]}}_{u^*(X_t^v,t)}\big\|^2\,\mathrm{d}t\,\alpha(v,X^v,B)\big].
$$

$$
\tag{25}
$$

Remark that the bias term in equation (24) is equal to the characterization of the cross-entropy loss in (14). In other words, the landscape of $\mathcal{L}_{\mathrm{SOCM}}(u,M)$ with respect to $u$ is the landscape of the cross-entropy loss $\mathcal{L}_{\mathrm{CE}}(u)$. Thus, the SOCM loss can be seen as some form of variance reduction method for the cross-entropy loss, and performs substantially better experimentally (Sec. 4). Yet, the expressions of the SOCM loss and the cross-entropy loss are very different; the former is a least squares loss and is expressed in terms of the gradients of the costs.

---

**Algorithm 2** Stochastic Optimal Control Matching (SOCM)

---

**Input:** State cost $f(x,t)$, terminal cost $g(x)$, diffusion coeff. $\sigma(t)$, base drift $b(x,t)$, noise level $\lambda$, number of iterations $N$, batch size $m$, number of time steps $K$, initial control parameters $\theta_0$, initial matrix parameters $\omega_0$, loss $\mathcal{L}_{\mathrm{SOCM}}$ in (19)

1 **for** $n \in \{0,\dots,N-1\}$ **do**
2    Simulate $m$ trajectories of the process $X^v$ controlled by $v = u_{\theta_n}$, e.g., using Euler-Maruyama updates
3    Detach the $m$ trajectories from the computational graph, so that gradients do not backpropagate
4    Using the $m$ trajectories, compute an $m$-sample Monte-Carlo approximation $\hat{\mathcal{L}}_{\mathrm{SOCM}}(u_{\theta_n}, M_{\omega_n})$ of the loss $\mathcal{L}_{\mathrm{SOCM}}(u_{\theta_n}, M_{\omega_n})$ in (19)
5    Compute the gradients $\nabla_{(\theta,\omega)}\hat{\mathcal{L}}_{\mathrm{SOCM}}(u_{\theta_n}, M_{\omega_n})$ of $\hat{\mathcal{L}}_{\mathrm{SOCM}}(u_{\theta_n}, M_{\omega_n})$ at $(\theta_n, \omega_n)$
6    Obtain $\theta_{n+1}, \omega_{n+1}$ with via an Adam update on $\theta_n, \omega_n$, resp.
7 **end**
**Output:** Learned control $u_{\theta_N}$

---

For good training performance, it is critical that the gradients have high signal-to-noise ratio. Looking at the SOCM loss, a good proxy for low gradient variance is to have low variance for $\tfrac{1}{T}\int_0^T \big\|u(X_t^v,t) - w(t,v,X^v,B,M_t)\big\|^2\,\mathrm{d}t \times \alpha(v,X^v,B)$, and this holds when both $\alpha(v,X^v,B)$ and $w(t,v,X^v,B,M_t)$ have low variance. Next, we present strategies to lower the variance of these two objects.

**Minimizing the variance of the importance weight** $\alpha$   We want to use a vector field $v$ such that $\mathrm{Var}[\alpha(v, X^v, B)]$ is as low as possible. As shown by the following lemma, which is well-known in the literature, setting $v$ to be the optimal control $u^*$ actually achieves variance zero when we condition on the starting point of the controlled process $X^v$. The proof of this result can be found in Hartmann et al. [38], but we include it in Subsec. C.5 for completeness.

**Lemma 1.** *When we set $v = u^*$, the conditional variance $\mathrm{Var}[\alpha(v, X^v, B)|X_0^v = x_{\mathrm{init}}]$ is zero for any $x_{\mathrm{init}} \in \mathbb{R}^d$.*

Of course, we do not have access to the optimal control $u^*$, but it is still a good idea to set $v$ as the closest vector field to $u^*$ that we have access to, which is typically the currently learned control. In some instances, one may benefit from using a warm-started control parameterized as $u_{\mathrm{WS}}(x,t) + u_\theta(x,t)$, where the warm-start $u_{\mathrm{WS}}$ is a reasonably good control obtained via a different strategy (see App. E).

**Minimizing the variance of the matching vector field** $w$   We are interested in finding the family $M = (M_t)_{t\in[0,T]}$ that minimizes the variance of $w(t, v, X^v, B, M_t)$ conditioned on $t$ and $X_t$. Note that this is exactly the term $\mathrm{CondVar}(w; M)$ in the right-hand side of equation (24). Since $\mathrm{CondVar}(w; M)$ does not depend on the specific $v$, the optimal $M$ does not depend on $v$ either. And since the second term in the right-hand side of equation (24) does not depend on $M = (M_t)_{t\in[0,T]}$, minimizing $\mathrm{CondVar}(w; M)$ is equivalent to minimizing $\mathcal{L}(u)$ with respect to $M$.

**Parameterizing the matrices** $M_t$ **vs solving for the optimal matrices**   In practice, we parameterize the matrices $(M_t)_{t\in[0,T]}$ using a function $M_\omega$ with two arguments $(t, s)$. To enforce that $M_\omega(t, t) = \mathrm{Id}$, we set $M_\omega(t, s) = e^{-\gamma(s-t)}\mathrm{Id} + (1 - e^{-\gamma(s-t)})\tilde{M}_{\tilde\omega}(t, s)$, where $\omega = (\gamma, \tilde\omega)$, and $\tilde{M}_{\tilde\omega} : \mathbb{R} \times \mathbb{R} \to \mathbb{R}^{d\times d}$ is an unconstrained neural network. Alternatively, Thm. 4 in App. D shows that the optimal family $M^* = (M_t^*)_{t\in[0,T]}$ can be characterized as the solution of a linear equation in infinite dimensions (a Fredholm equation of the first kind). The discretized linear system has $d^2 K$ equations and variables, $K$ being the number of discretization time points. However, since the optimal $M^*$ does not depend on $v$ (see Remark 1), this is a computation that must be done only once and that may be affordable in some settings. We did not test this approach experimentally.

## 4   Experiments

We consider four experimental settings that we adapt from Nüsken and Richter [59]: QUADRATIC ORNSTEIN UHLENBECK (EASY), QUADRATIC ORNSTEIN UHLENBECK (HARD), LINEAR ORN-STEIN UHLENBECK and DOUBLE WELL. We describe them in detail in App. F. For all of them, we have access to the ground-truth optimal control, which means that we are able to estimate the $L^2$ error incurred by the learned control $u$. In Figure 2 we plot the control $L^2$ error for each IDO algorithm described in Subsec. 2.2, and for the SOCM algorithm (Algorithm 2), for the QUADRATIC OU (EASY) and (HARD) settings. We also include two ablations of SOCM: *(i)* a version of SOCM where the reparameterization matrices $M_t$ are set fixed to the identity $I$, *(ii)* SOCM-Adjoint, where we estimate the conditional expectation in equation (23) using the adjoint method for SDEs instead of the path-wise reparameterization trick (see Subsec. C.4). Code can be found at `https://github.com/facebookresearch/SOC-matching`.

At the end of training, SOCM obtains the lowest $L^2$ error, improving over all existing methods by a factor of around ten. The two SOCM ablations come in second and third by a substantial difference, which underlines the importance of the path-wise reparameterization trick. The best among existing methods is the adjoint method (the relative entropy loss). In Figure 2 (*bottom*) we show the squared norm of the gradient of each loss with respect to the parameters $\theta$ of the control: algorithms with small noise variance have low error values.

In Figure 3, we plot the control $L^2$ error for LINEAR ORNSTEIN UHLENBECK and DOUBLE WELL. For LINEAR OU, the error is around five times smaller for SOCM than for any existing method. For DOUBLE WELL, the SOCM algorithm achieves the third smallest error, slightly behind the variance loss and the adjoint method, but the latter shows instabilities. As we show in Figure 9 in App. F, these instabilities are inherent to the adjoint method and they do not disappear for small learning rates. Both in Figure 2 and Figure 3, we observe that learning the reparameterization matrices is critical to obtain gradient estimates with high signal-to-noise ratio. DOUBLE WELL is a particularly interesting and challenging setting because its solution is highly multimodal: $g$ has 1024 modes. Multimodality

is a feature observed in realistic settings, and is hard to handle because it involves learning the control correctly in each mode.

The costs $f$ and $g$ and the base drift $b$ for QUADRATIC OU (HARD) are five times those of QUADRATIC OU (EASY). Consequently, the factor $\alpha(v, X^v, B)$ initially has a much larger variance for the SOCM methods, and for cross-entropy. As training progresses, $u_{\theta_n}$ gets closer to $u^*$, and consequently the variance of $\alpha(v, X^v, B)$ decreases, which in turn makes learning easier. This explains the initial slow decrease in the control error, followed by a fast drop that places SOCM well below existing algorithms. In App. E, we showcase a control warm-start strategy that can help and speed up convergence.

We also present experimental results on two-mode Gaussian mixture sampling in increasing dimension, using the Path Integral Sampler [84]. We take Gaussians with means that are 2 units apart, and identity variance. Figure 1 shows control objective estimates obtained after running the Adjoint, SOCM, and Cross-entropy algorithms for 40000 iterations, at dimensions $d = 2, 8, 16, 32, 64$, and error bars show standard errors. By Theorem 4 of [84], we know that the optimal value of the control objective is zero; Figure 1 shows the suboptimality gaps incurred by each algorithm. Cross-entropy,

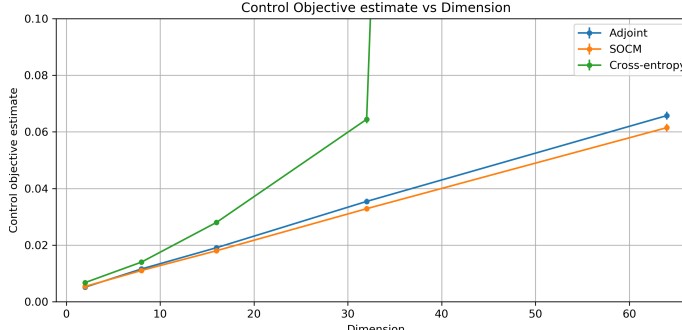

Figure 1: This plot shows the control objective values for different algorithms (Adjoint, SOCM, and Cross-entropy) across multiple dimensions, with error bars indicating the standard deviations. The y-axis is restricted to $[0, 0.1]$ for better visibility of the lower range values; cross-entropy takes value $2.915 \pm 0.008$ at $d = 64$.

which uses the same importance weight as SOCM, performs worse than the other two losses for all dimensions, and its results are particularly poor for dimension 64, because the variance of $\alpha$ is too large for learning to happen. In this case, we see that SOCM has better variance reduction than cross-entropy, despite both using importance weighted objectives for training. We observe that the values for SOCM are slightly below that of Adjoint for most dimensions, which confirms that our method is better for this range of dimensions. If we keep increasing the dimension, SOCM also fails due to higher variance of $\alpha$: for $n = 128$, the control objective estimates for the Adjoint, SOCM, and Cross-Entropy losses are $0.146 \pm 0.001$, $7.49 \pm 0.01$, and $12.61 \pm 0.02$, respectively.

## 5 Conclusion

Our work introduces Stochastic Optimal Control Matching, a novel Iterative Diffusion Optimization technique for stochastic optimal control that stems from the same philosophy as the conditional score matching loss for diffusion models. That is, the control is learned via a least-squares problem by trying to fit a matching vector field. The training loss is optimized with respect to both the control function and a family of reparameterization matrices which appear in the matching vector field. Optimizing the reparameterization matrices reduces the variance of the matching vector field. Experimentally, our algorithm achieves lower error than all existing IDO techniques in four settings.

One of the key ideas for deriving the SOCM algorithm is the path-wise reparameterization trick, a novel technique to obtain low-variance estimates of the gradient of the conditional expectation of a functional of a random process with respect to its initial value. An interesting future direction is to use the path-wise reparameterization trick to decrease the variance of the matching vector field for diffusion models. The main roadblock when we try to apply SOCM to more challenging problems is that the variance of the factor $\alpha(v, X^v, B)$ explodes when $f$ and/or $g$ are large, or when the dimension $d$ is high. The large variance of $\alpha$ is due to the mismatch between the probability measures induced by the learned control and the optimal control, and it decreases as the learned control approaches the optimal control.

The research presented is foundational, but it may serve as the basis of algorithms that improve the quality of generative models.

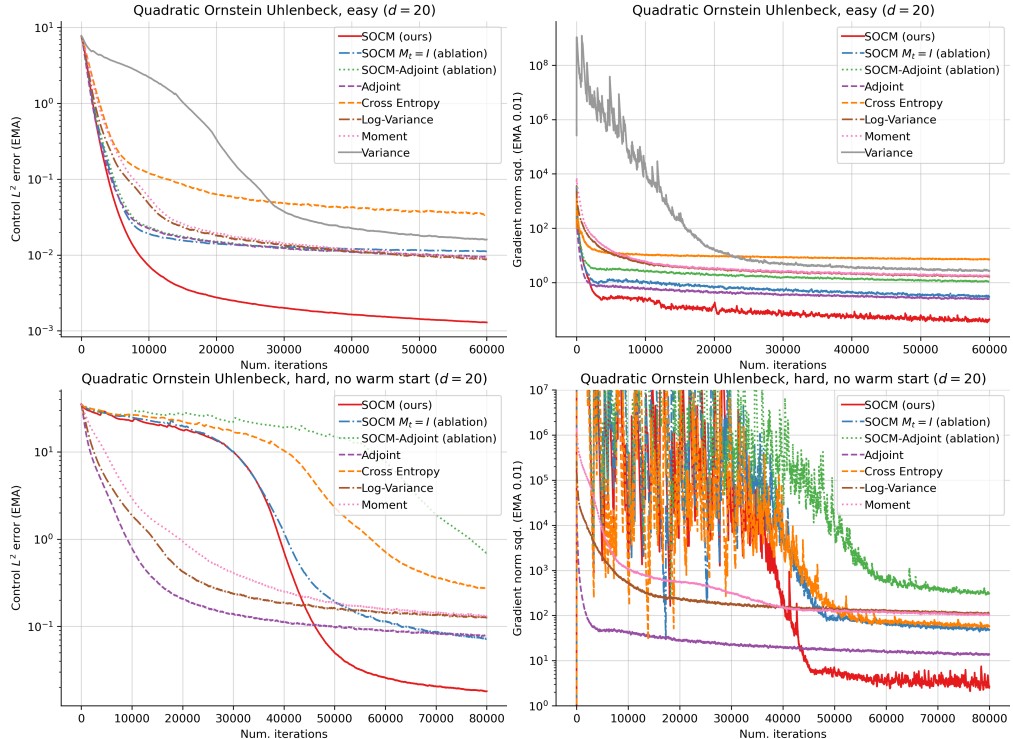

Figure 2: Plots of the $L^2$ error incurred by the learned control (*left*), and the norm squared of the gradient with respect to the parameters $\theta$ of the control (*right*), for the QUADRATIC ORNSTEIN UHLENBECK (EASY) (*top*) and (HARD) (*bottom*) settings and for each IDO loss. Both plots show exponential moving averages.

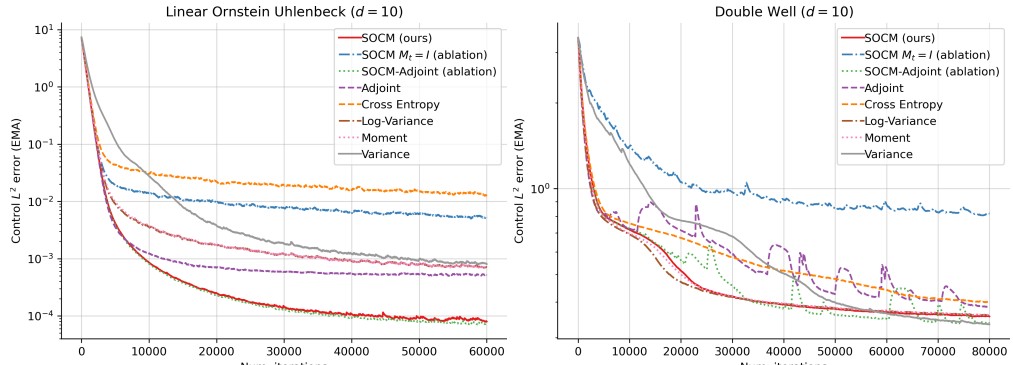

Figure 3: Plots of the $L^2$ error of the learned control for the LINEAR ORNSTEIN UHLENBECK and DOUBLE WELL settings.

**Funding disclosure** Funded by respective author affiliations.

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

# Contents

## A    Technical assumptions

Throughout our work, we make the same assumptions as [59], which are needed for all the objects considered to be well-defined. Namely, we assume that:

(i) The set $\mathcal{U}$ of *admissible controls* is given by

$$\mathcal{U} = \{u \in C^1(\mathbb{R}^d \times [0,T]; \mathbb{R}^d) \mid \exists C > 0, \, \forall (x,s) \in \mathbb{R}^d \times [0,T], \, b(x,s) \leq C(1+|x|)\}.$$

(ii) The coefficients $b$ and $\sigma$ are continuously differentiable, $\sigma$ has bounded first-order spatial derivatives, and $(\sigma\sigma^\top)(x,s)$ is positive definite for all $(x,s) \in \mathbb{R}^d \times [0,T]$. Furthermore, there exist constants $C, c_1, c_2 > 0$ such that

$$\|b(x,s)\| \leq C(1+\|x\|), \qquad \text{(linear growth)}$$
$$c_1\|\xi\|^2 \leq \xi^\top(\sigma\sigma^\top)(x,s)\xi \leq c_2\|\xi\|^2, \qquad \text{(ellipticity)}$$

for all $(x,s) \in \mathbb{R}^d \times [0,T]$ and $\xi \in \mathbb{R}^d$.

## B    Proofs of Sec. 2

**Proof of** (5)    By Itô's lemma, we have that

$$V(X_T^u, T) - V(X_t^u, t) = \int_t^T \left( \partial_s V(X_s^u, s) + \langle b(X_s^u, s) + \sigma(X_s^u, s)u(X_s^u, s), \nabla V(X_s^u, s) \rangle \right.$$
$$\left. + \tfrac{\lambda}{2} \sum_{i,j=1}^d (\sigma\sigma^\top)_{ij}(X_s^u, s) \partial_{x_i} \partial_{x_j} V(X_s^u, s) \right) \mathrm{d}s + S_t^u,$$

where $S_t^u = \sqrt{\lambda} \int_t^T \nabla V(X_s^u, s)^\top \sigma(X_s^u, s) \, \mathrm{d}B_s$. Note that by (4),

$$\partial_s V(X_s^u, s) + \langle b(X_s^u, s) + \sigma(X_s^u, s)u(X_s^u, s), \nabla V(X_s^u, s)\rangle$$
$$+ \tfrac{\lambda}{2} \sum_{i,j=1}^d (\sigma\sigma^\top)_{ij}(X_s^u, s)\partial_{x_i}\partial_{x_j} V(X_s^u, s)$$
$$= \tfrac{1}{2}\|(\sigma^\top \nabla V)(X_s^u, s)\|^2 - f(X_s^u, s) + \langle \sigma(X_s^u, s)u(X_s^u, s), \nabla V(X_s^u, s)\rangle$$
$$= \tfrac{1}{2}\|(\sigma^\top \nabla V)(X_s^u, s) + u(X_s^u, s)\|^2 - \tfrac{1}{2}\|u(X_s^u, s)\|^2 - f(X_s^u, s),$$

and this implies that

$$g(X_T^u) - V(X_t^u, t) = \int_t^T \left( \tfrac{1}{2}\|(\sigma^\top \nabla V)(X_s^u, s) + u(X_s^u, s)\|^2 - \tfrac{1}{2}\|u(X_s^u, s)\|^2 - f(X_s^u, s)\right) \mathrm{d}s + S_t^u \tag{26}$$

Since $\mathbb{E}[S_t^u \mid X_t^u = x] = 0$, rearranging (26) and taking the conditional expectation with respect to $X_t^u$ yields the final result.

**Proof of** (6)-(7)    By Itô's lemma, we have that

$$dV(X_s, s) = \big(\partial_s V(X_s, s) + \langle b(X_s, s), \nabla V(X_s, s)\rangle$$
$$+ \tfrac{\lambda}{2} \sum_{i,j=1}^d (\sigma\sigma^\top)_{ij}(X_s, s)\partial_{x_i}\partial_{x_j} V(X_s, s)\big) \, \mathrm{d}s + \sqrt{\lambda}\nabla V(X_s^u, s)^\top \sigma(X_s^u, s) \, \mathrm{d}B_s, \tag{27}$$

Note that by (4),

$$\partial_s V(X_s, s) + \langle b(X_s, s), \nabla V(X_s, s)\rangle + \tfrac{\lambda}{2} \sum_{i,j=1}^d (\sigma\sigma^\top)_{ij}(X_s, s)\partial_{x_i}\partial_{x_j} V(X_s, s)$$
$$= \tfrac{1}{2}\|(\sigma^\top \nabla V)(X_s, s)\|^2 - f(X_s, s).$$

Plugging this into (27) concludes the proof.

**Proof of** (8)    Since $Y_s = V(X_s, s)$ and $Z_s = \sigma^\top(s)\nabla V(X_s, s) = -u^*(X_s, s)$ satisfy (7), we have that

$$g(X_T) = Y_T = Y_t - \int_t^T (f(X_s, s) - \tfrac{1}{2}\|u^*(X_s, s)\|^2) \, \mathrm{d}s - \sqrt{\lambda} \int_t^T \langle u^*(X_s, s), \mathrm{d}B_s\rangle.$$

Hence, recalling the definition of the work functional in (10), we have that

$$\mathcal{W}(X, t) = Y_t + \tfrac{1}{2} \int_t^T \|u^*(X_s, s)\|^2 \, \mathrm{d}s - \sqrt{\lambda} \int_t^T \langle u^*(X_s, s), \mathrm{d}B_s\rangle. \tag{28}$$

By Novikov's theorem (Thm. 2), we have that

$$\mathbb{E}[\exp(-\lambda^{-1}\mathcal{W}(X, t))|X_t]$$
$$= e^{-\lambda^{-1}Y_t}\mathbb{E}\big[\exp\big(\lambda^{-1/2} \int_t^T \langle u^*(X_s, s), \mathrm{d}B_s\rangle - \tfrac{\lambda^{-1}}{2} \int_t^T \|u^*(X_s, s)\|^2 \, \mathrm{d}s\big)\big|X_t\big] = e^{-\lambda^{-1}Y_t},$$

which concludes the proof of (8).

**Theorem 2** (Novikov's theorem). *Let $\theta_s$ be a locally-$\mathcal{H}_2$ process which is adapted to the natural filtration of the Brownian motion $(B_t)_{t\geq 0}$. Define*

$$Z(t) = \exp\big( \int_0^t \theta_s \, \mathrm{d}B_s - \tfrac{1}{2} \int_0^t \|\theta_s\|^2 \, \mathrm{d}s\big). \tag{29}$$

*If for each $t \geq 0$,*

$$\mathbb{E}\big[\exp\big( \int_0^t \|\theta_s\|^2 \, \mathrm{d}s\big)\big] < +\infty,$$

*then for each $t \geq 0$,*

$$\mathbb{E}[Z(t)] = 1. \tag{30}$$

*Moreover, the process $Z(t)$ is a positive martingale, i.e. if $(\mathcal{F}_t)_{t\geq 0}$ is the filtration associated to the Brownian motion $(B_t)_{t\geq 0}$, then for $t \geq s$, $\mathbb{E}[Z_t|\mathcal{F}_s] = Z_s$.*

**Theorem 3** (Girsanov theorem). *Let $W = (W_t)_{t \in [0,T]}$ be a standard Wiener process, and let $\mathbb{P}$ be its induced probability measure over $C([0,T]; \mathbb{R}^d)$, known as the Wiener measure. Let $Z(t)$ be as defined in* (29) *and suppose that the assumptions of Theorem 2 hold. Let $(\Omega, \mathcal{F})$ be the $\sigma$-algebra associated to $B_T$. For any $F \in \mathcal{F}$, define the measure*

$$\mathbb{Q}(F) = \mathbb{E}_{\mathbb{P}}[Z(T)\mathbf{1}_F].$$

*$\mathbb{Q}$ is a probability measure because of* (30). *Under the probability measure $\mathbb{Q}$, the stochastic process $\{\tilde{W}(t)\}_{0 \le t \le T}$ defined as*

$$\tilde{W}(t) = W(t) - \int_0^t \theta_s \, \mathrm{d}s$$

*is a standard Wiener process. That is, for any $n \ge 0$ and any $0 = t_0 < t_1 < \cdots < t_n$, the increments $\{\tilde{W}(t_{i+1}) - \tilde{W}(t_i)\}_{i=0}^{n-1}$ are independent and $\mathbb{Q}$-Gaussian distributed with mean zero and covariance $(t_{i+1} - t_i)\mathrm{I}$, which means that for any $\alpha \in \mathbb{R}^d$, the moment generating function of $\tilde{W}(t_{i+1}) - \tilde{W}(t_i)$ with respect to $\mathbb{Q}$ is as follows:*

$$\mathbb{E}_{\mathbb{Q}}[\exp(\langle \alpha, \tilde{W}(t_{i+1}) - \tilde{W}(t_i)\rangle)]$$
$$:= \mathbb{E}_{\mathbb{P}}\big[\exp\big(\langle \alpha, W(t_{i+1}) - \int_0^{t_{i+1}} \theta_s \, \mathrm{d}s - W(t_i) + \int_0^{t_i} \theta_s \, \mathrm{d}s\rangle\big)Z(T)\big] = \exp\big(\tfrac{(t_{i+1}-t_i)\|\alpha\|^2}{2}\big).$$

**Corollary 1** (Girsanov theorem for SDEs). *If the two SDEs*

$$\mathrm{d}X_t = b_1(X_t, t) \, \mathrm{d}t + \sigma(X_t, t) \, \mathrm{d}B_t, \qquad X_0 = x_{\mathrm{init}}$$
$$dY_t = (b_1(Y_t, t) + b_2(Y_t, t)) \, \mathrm{d}t + \sigma(Y_t, t) \, \mathrm{d}B_t, \qquad Y_0 = x_{\mathrm{init}}$$

*admit unique strong solutions on $[0, T]$, then for any bounded continuous functional $\Phi$ on $C([0,T])$, we have that*

$$\mathbb{E}[\Phi(X)] = \mathbb{E}\big[\Phi(Y)\exp\big(-\int_0^T \sigma(Y_t, t)^{-1}b_2(Y_t, t) \, \mathrm{d}B_t - \tfrac{1}{2}\int_0^T \|\sigma(Y_t, t)^{-1}b_2(Y_t, t)\|^2 \, \mathrm{d}t\big)\big]$$
$$= \mathbb{E}\big[\Phi(Y)\exp\big(-\int_0^T \sigma(Y_t, t)^{-1}b_2(Y_t, t) \, d\tilde{B}_t + \tfrac{1}{2}\int_0^T \|\sigma(Y_t, t)^{-1}b_2(Y_t, t)\|^2 \, \mathrm{d}t\big)\big],$$

*where $\tilde{B}_t = B_t + \int_0^t \sigma(Y_s, s)^{-1}b_2(Y_s, s) \, \mathrm{d}s$. More generally, $b_1$ and $b_2$ can be random processes that are adapted to filtration of $B$.*

**Lemma 2.** *For an arbitrary $v \in \mathcal{U}$, let $\mathbb{P}^v$ and $\mathbb{P}$ be respectively the laws of the SDEs*

$$\mathrm{d}X_t^v = (b(X_t^v, t) + \sigma(t)v(X_t^v, t)) \, \mathrm{d}t + \sqrt{\lambda}\sigma(t)\mathrm{d}B_t, \qquad X_0^v \sim p_0,$$
$$\mathrm{d}X_t = b(X_t, t) \, \mathrm{d}t + \sqrt{\lambda}\sigma(t)\mathrm{d}B_t, \qquad X_0 \sim p_0.$$

*We have that*

$$\frac{d\mathbb{P}}{d\mathbb{P}^v}(X^v) = \exp\big(-\lambda^{-1/2}\int_0^T \langle v(X_t^v, t), \mathrm{d}B_t^v\rangle + \tfrac{\lambda^{-1}}{2}\int_0^T \|v(X_t^v, t)\|^2 \, \mathrm{d}t\big) \tag{31}$$
$$= \exp\big(-\lambda^{-1/2}\int_0^T \langle v(X_t^v, t), \mathrm{d}B_t\rangle - \tfrac{\lambda^{-1}}{2}\int_0^T \|v(X_t^v, t)\|^2 \, \mathrm{d}t\big),$$
$$\frac{d\mathbb{P}^v}{d\mathbb{P}}(X) = \exp\big(\lambda^{-1/2}\int_0^T \langle v(X_t, t), \mathrm{d}B_t\rangle - \tfrac{\lambda^{-1}}{2}\int_0^T \|v(X_t, t)\|^2 \, \mathrm{d}t\big). \tag{32}$$

*where $B_t^v := B_t + \lambda^{-1/2}\int_0^t v(X_s^v, s) \, \mathrm{d}s$. For the optimal control $u^*$, we have that*

$$\frac{d\mathbb{P}}{d\mathbb{P}^{u^*}}(X^{u^*}) = \exp\big(\lambda^{-1}\big(-V(X_0^{u^*}, 0) + \mathcal{W}(X^{u^*}, 0)\big)\big), \tag{33}$$
$$\frac{d\mathbb{P}^{u^*}}{d\mathbb{P}}(X) = \exp\big(\lambda^{-1}\big(V(X_0, 0) - \mathcal{W}(X, 0)\big)\big), \tag{34}$$

*where the functional $\mathcal{W}$ is defined in* (10).

*Proof.* The proof of (31)-(32) follows directly from Cor. 1. To prove (34), we use that by (28),

$$\mathcal{W}(X, 0) = V(X_0, 0) + \tfrac{1}{2}\int_0^T \|u^*(X_s, s)\|^2 \, \mathrm{d}s - \sqrt{\lambda}\int_0^T \langle u^*(X_s, s), \mathrm{d}B_s\rangle, \tag{35}$$

which implies that

$$\frac{d\mathbb{P}^{u^*}}{d\mathbb{P}}(X) = \exp\big(\lambda^{-1/2}\int_0^T \langle u^*(X_t, t), \mathrm{d}B_t\rangle - \tfrac{\lambda^{-1}}{2}\int_0^T \|u^*(X_t, t)\|^2 \, \mathrm{d}t\big)$$
$$= \exp\big(\lambda^{-1}\big(V(X_0, 0) - \mathcal{W}(X, 0)\big)\big).$$

To prove (33), we use that since $\mathrm{d}X_t^{u^*} = b(X_t^{u^*}, t)\,\mathrm{d}t + \sqrt{\lambda}\sigma(t)\mathrm{d}B_t^{u^*}$, equation (35) holds if we replace $X$ and $B$ by $X^{u^*}$ and $B^{u^*}$, which reads

$$\mathcal{W}(X^{u^*}, 0) = V(X_0^{u^*}, 0) + \tfrac{1}{2}\int_t^T \|u^*(X_s^{u^*}, s)\|^2\,\mathrm{d}s - \sqrt{\lambda}\int_t^T \langle u^*(X_s^{u^*}, s), \mathrm{d}B_s^v\rangle.$$

Hence,

$$\frac{d\mathbb{P}}{d\mathbb{P}^{u^*}}(X^{u^*}) = \exp\big(-\lambda^{-1/2}\int_0^T \langle u^*(X_t^{u^*}, t), \mathrm{d}B_t^{u^*}\rangle + \tfrac{\lambda^{-1}}{2}\int_0^T \|u^*(X_t^{u^*}, t)\|^2\,\mathrm{d}t\big)$$
$$= \exp\big(\lambda^{-1}\big(-V(X_0^{u^*}, 0) + \mathcal{W}(X^{u^*}, 0)\big)\big).$$

$\square$

**Lemma 3.** *The following expression holds:*

$$\mathbb{E}_{\mathbb{P}^u}\big[\log\tfrac{d\mathbb{P}^u}{d\mathbb{P}^{u^*}}\big] = \lambda^{-1}\mathbb{E}\big[\int_0^T \big(\tfrac{1}{2}\|u(X_t^u, t)\|^2 + f(X_t^u, t)\big)\,\mathrm{d}t + g(X_T^u) - V(X_0^u, 0)\big], \quad (36)$$

*Proof.* To prove (36), we write

$$\log\tfrac{d\mathbb{P}^{u^*}}{d\mathbb{P}^u}(X^u) = \log\big(\tfrac{d\mathbb{P}^{u^*}}{d\mathbb{P}}(X^u)\tfrac{d\mathbb{P}}{d\mathbb{P}^u}(X^u)\big) = \log\tfrac{d\mathbb{P}^{u^*}}{d\mathbb{P}}(X^u) + \log\tfrac{d\mathbb{P}}{d\mathbb{P}^u}(X^u)$$
$$= \lambda^{-1}\big(V(X_0^u, 0) - \int_0^T f(X_t^u, t)\,\mathrm{d}t - g(X_T^u)\big)$$
$$- \lambda^{-1/2}\int_0^T \langle u(X_t^u, t), \mathrm{d}B_t\rangle - \tfrac{\lambda^{-1}}{2}\int_0^T \|u(X_t^u, t)\|^2\,\mathrm{d}t .$$

Since $\mathbb{E}_{\mathbb{P}^u}\big[\log\tfrac{d\mathbb{P}^u}{d\mathbb{P}^{u^*}}\big] = -\mathbb{E}_{\mathbb{P}^u}\big[\log\tfrac{d\mathbb{P}^{u^*}}{d\mathbb{P}^u}\big]$, and $\mathbb{E}_{\mathbb{P}^u}\big[\int_0^T \langle u(X_t^u, t), \mathrm{d}B_t\rangle\big] = 0$, the result follows.

$\square$

**Proposition 3.** *(i) The following two expressions hold for arbitrary controls $u, v$ in the class $\mathcal{U}$ of admissible controls:*

$$\tilde{\mathcal{L}}_{\mathrm{CE}}(u) = \mathbb{E}_{\mathbb{P}^{u^*}}\big[\log\tfrac{d\mathbb{P}^{u^*}}{d\mathbb{P}^u}\big] = \mathbb{E}\big[\big(-\lambda^{-1/2}\int_0^T \langle u(X_t^v, t), \mathrm{d}B_t\rangle - \lambda^{-1}\int_0^T \langle u(X_t^v, t), v(X_t^v, t)\rangle\,\mathrm{d}t$$
$$(37)$$
$$+ \tfrac{\lambda^{-1}}{2}\int_0^T \|u(X_t^v, t)\|^2\,\mathrm{d}t + \lambda^{-1}\big(V(X_0^v, 0) - \mathcal{W}(X^v, 0)\big)\big)$$
$$\times \exp\big(\lambda^{-1}\big(V(X_0^v, 0) - \mathcal{W}(X^v, 0)\big)$$
$$- \lambda^{-1/2}\int_0^T \langle v(X_t^v, t), \mathrm{d}B_t\rangle - \tfrac{\lambda^{-1}}{2}\int_0^T \|v(X_t^v, t)\|^2\,\mathrm{d}t\big)\big],$$

$$\tilde{\mathcal{L}}_{\mathrm{CE}}(u) = \tfrac{\lambda^{-1}}{2}\mathbb{E}\big[\int_0^T \|u^*(X_t^{u^*}, t) - u(X_t^{u^*}, t)\|^2\,\mathrm{d}t\big]. \quad (38)$$

*When $p_0$ is concentrated at a single point $x_{\mathrm{init}}$, the terms $V(x_{\mathrm{init}}, 0)$ are constant and can be removed without modifying the landscape. In other words, $\tilde{\mathcal{L}}_{\mathrm{CE}}$ and $\mathcal{L}_{\mathrm{CE}}$ are equal up to constant terms and constant factors.*

*(ii) When $p_0$ is a generic probability measure, $\tilde{\mathcal{L}}_{\mathrm{CE}}$ and $\mathcal{L}_{\mathrm{CE}}$ have different landscapes, and $\mathcal{L}_{\mathrm{CE}}(u) = \mathbb{E}_{\mathbb{P}^{u^*}}\big[\log\tfrac{d\mathbb{P}^{u^*}}{d\mathbb{P}^u}\exp\big(-\lambda^{-1}V(X_0^{u^*}, 0)\big)\big]$. $u^*$ is still the only minimizer of the loss $\mathcal{L}_{\mathrm{CE}}$, and for some constant $K$, we have that*

$$\mathcal{L}_{\mathrm{CE}}(u, 0) = \tfrac{\lambda^{-1}}{2}\mathbb{E}\big[\int_0^T \|u^*(X_t^{u^*}, t) - u(X_t^{u^*}, t)\|^2\,\mathrm{d}t\exp\big(-\lambda^{-1}V(X_0^{u^*}, 0)\big)\big] + K. \quad (39)$$

*Proof.* We begin with the proof of (i), and prove (37) first. Note that by the Girsanov theorem (Thm. 3),

$$\mathbb{E}_{\mathbb{P}^{u^*}}\big[\log\tfrac{d\mathbb{P}^{u^*}}{d\mathbb{P}^u}(X^{u^*})\big] = -\mathbb{E}_{\mathbb{P}^{u^*}}\big[\log\tfrac{d\mathbb{P}^u}{d\mathbb{P}^{u^*}}(X^{u^*})\big] = -\mathbb{E}_{\mathbb{P}^{u^*}}\big[\log\tfrac{d\mathbb{P}^u}{d\mathbb{P}}(X^{u^*}) + \log\tfrac{d\mathbb{P}}{d\mathbb{P}^{u^*}}(X^{u^*})\big]$$
$$(40)$$
$$= -\mathbb{E}_{\mathbb{P}^v}\big[\big(\log\tfrac{d\mathbb{P}^u}{d\mathbb{P}}(X^v) + \log\tfrac{d\mathbb{P}}{d\mathbb{P}^{u^*}}(X^v)\big)\tfrac{d\mathbb{P}^{u^*}}{d\mathbb{P}}(X^v)\tfrac{d\mathbb{P}}{d\mathbb{P}^v}(X^v)\big]$$

Note that by equations (32) and (34),

$$\log\tfrac{d\mathbb{P}^u}{d\mathbb{P}}(X^v) = \lambda^{-1/2}\int_0^T \langle u(X_t^v, t), \mathrm{d}B_t^v\rangle - \tfrac{\lambda^{-1}}{2}\int_0^T \|u(X_t^v, t)\|^2\,\mathrm{d}t,$$
$$= \lambda^{-1/2}\int_0^T \langle u(X_t^v, t), \mathrm{d}B_t\rangle + \lambda^{-1}\int_0^T \langle u(X_t^v, t), v(X_t^v, t)\rangle\,\mathrm{d}t - \tfrac{\lambda^{-1}}{2}\int_0^T \|u(X_t^v, t)\|^2\,\mathrm{d}t,$$
$$\log\tfrac{d\mathbb{P}}{d\mathbb{P}^{u^*}}(X^v) = \lambda^{-1}\big(-V(X_0^v, 0) + \mathcal{W}(X^v, 0)\big).$$

$$(41)$$

where $B_t^v := B_t + \lambda^{-1/2} \int_0^t v(X_s^v, s)\, \mathrm{d}s$. Also,

$$\frac{d\mathbb{P}^{u^*}}{d\mathbb{P}}(X^v) = \exp\left(\lambda^{-1}\left(V(X_0^v, 0) - \mathcal{W}(X^v, 0)\right)\right),$$
$$\frac{d\mathbb{P}}{d\mathbb{P}^v}(X^v) = \exp\left(-\lambda^{-1/2}\int_0^T \langle v(X_t^v, t), \mathrm{d}B_t\rangle - \tfrac{\lambda^{-1}}{2}\int_0^T \|v(X_t^v, t)\|^2\, \mathrm{d}t\right). \tag{42}$$

If we plug (41) and (42) into the right-hand side of (40), we obtain

$$\mathbb{E}_{\mathbb{P}^{u^*}}\left[\log \frac{d\mathbb{P}^{u^*}}{d\mathbb{P}^u}(X^{u^*})\right] = -\mathbb{E}_{\mathbb{P}^{u^*}}\left[(\log \frac{d\mathbb{P}^u}{d\mathbb{P}}(X^v) + \log \frac{d\mathbb{P}}{d\mathbb{P}^{u^*}}(X^v))\frac{d\mathbb{P}^{u^*}}{d\mathbb{P}}(X^v)\frac{d\mathbb{P}}{d\mathbb{P}^u}(X^v)\right]$$
$$= -\mathbb{E}\Big[\big(\lambda^{-1/2}\int_0^T \langle u(X_t^v, t), \mathrm{d}B_t\rangle + \lambda^{-1}\int_0^T \langle u(X_t^v, t), v(X_t^v, t)\rangle\, \mathrm{d}t$$
$$- \tfrac{\lambda^{-1}}{2}\int_0^T \|u(X_t^v, t)\|^2\, \mathrm{d}t + \lambda^{-1}\big(-V(X_0^v, 0) + \mathcal{W}(X^v, 0)\big)\big)$$
$$\times \exp\left(\lambda^{-1}\left(V(X_0^v, 0) - \mathcal{W}(X^v, 0)\right) - \lambda^{-1/2}\int_0^T \langle v(X_t^v, t), \mathrm{d}B_t\rangle - \tfrac{\lambda^{-1}}{2}\int_0^T \|v(X_t^v, t)\|^2\, \mathrm{d}t\right)\Big],$$

which concludes the proof.

To show (38), we use that by Cor. 1,

$$\frac{d\mathbb{P}^u}{d\mathbb{P}^{u^*}}(X^{u^*}) = \exp\left(-\lambda^{-1/2}\int_0^T \langle u^*(X_t^{u^*}, t) - u(X_t^{u^*}, t), \mathrm{d}B_t\rangle - \tfrac{\lambda^{-1}}{2}\int_0^T \|u^*(X_t^{u^*}, t) - u(X_t^{u^*}, t)\|^2\, \mathrm{d}t\right).$$

Hence,

$$\mathbb{E}_{\mathbb{P}^{u^*}}\left[\log \frac{d\mathbb{P}^{u^*}}{d\mathbb{P}^u}\right] = -\mathbb{E}_{\mathbb{P}^{u^*}}\left[\log \frac{d\mathbb{P}^u}{d\mathbb{P}^{u^*}}\right] = \tfrac{\lambda^{-1}}{2}\mathbb{E}\left[\int_0^T \|u^*(X_t^{u^*}, t) - u(X_t^{u^*}, t)\|^2\, \mathrm{d}t\right].$$

Next, we prove (ii). The first instance of $V(X_0^v, 0)$ in (37) can be removed without modifying the landscape of the loss. Hence, we are left with

$$\bar{\mathcal{L}}_{\mathrm{CE}}(u) = \mathbb{E}\Big[\big(-\lambda^{-1/2}\int_0^T \langle u(X_t^v, t), \mathrm{d}B_t^v\rangle - \lambda^{-1}\int_0^T \langle u(X_t^v, t), v(X_t^v, t)\rangle\, \mathrm{d}t \tag{43}$$
$$+ \tfrac{\lambda^{-1}}{2}\int_0^T \|u(X_t^v, t)\|^2\, \mathrm{d}t - \lambda^{-1}\mathcal{W}(X^v, 0)\big)$$
$$\times \exp\left(\lambda^{-1}\left(V(X_0^v, 0) - \mathcal{W}(X^v, 0)\right) - \lambda^{-1/2}\int_0^T \langle v(X_t^v, t), \mathrm{d}B_t\rangle - \tfrac{\lambda^{-1}}{2}\int_0^T \|v(X_t^v, t)\|^2\, \mathrm{d}t\right)\Big]$$

And this can be expressed as

$$\bar{\mathcal{L}}_{\mathrm{CE}}(u) = \mathbb{E}\left[g(u; X_0^v)\exp\left(\lambda^{-1}V(X_0^v, 0)\right)\right],$$

where

$$g(u; x) = \mathbb{E}\Big[\big(-\lambda^{-1/2}\int_0^T \langle u(X_t^v, t), \mathrm{d}B_t^v\rangle - \lambda^{-1}\int_0^T \langle u(X_t^v, t), v(X_t^v, t)\rangle\, \mathrm{d}t$$
$$+ \tfrac{\lambda^{-1}}{2}\int_0^T \|u(X_t^v, t)\|^2\, \mathrm{d}t - \lambda^{-1}\mathcal{W}(X^v, 0)\big)$$
$$\times \exp\left(-\lambda^{-1}\mathcal{W}(X^v, 0) - \lambda^{-1/2}\int_0^T \langle v(X_t^v, t), \mathrm{d}B_t\rangle - \tfrac{\lambda^{-1}}{2}\int_0^T \|v(X_t^v, t)\|^2\, \mathrm{d}t\right)|X_0^v = x\Big].$$

If we consider $g(u; x)$ as a loss function for $u$, note that it is equivalent to the loss $\bar{\mathcal{L}}_{\mathrm{CE}}(u)$ equation in (43) for the choice $p_0 = \delta_x$, i.e., $p_0$ concentrated at $x$. Since the optimal control $u^*$ is independent of the starting distribution $p_0$, we deduce that $u^*$ is the unique minimizer of $g(u; x)$, for all $x \in \mathbb{R}^d$. In consequence, $u^*$ is the unique minimizer of $\mathcal{L}_{\mathrm{CE}}(u) = \mathbb{E}[g(u; X_0^v)]$.

To prove (39), note that up to a constant term, the only difference between $\bar{\mathcal{L}}_{\mathrm{CE}}(u)$ and $\mathcal{L}_{\mathrm{CE}}(u)$ is the expectation is reweighted importance weight $\exp\left(-\lambda^{-1}V(X_0^v, 0)\right)$. $\qquad\square$

**Lemma 4.** *(i) We can rewrite*

$$\tilde{\mathcal{L}}_{\mathrm{Var}_v}(u) = \mathrm{Var}\left(\exp\left(\tilde{Y}_T^{u,v} - \lambda^{-1}g(X_T^v) + \lambda^{-1}V(X_0^v, 0)\right)\right),$$
$$\tilde{\mathcal{L}}_{\mathrm{Var}_v}^{\log}(u) = \mathrm{Var}\left(\tilde{Y}_T^{u,v} - \lambda^{-1}g(X_T^v) + \lambda^{-1}V(X_0^v, 0)\right).$$

*When $p_0$ is concentrated at $x_{\mathrm{init}}$, the terms $V(x_{\mathrm{init}}, 0)$ are constants and can be removed without modifying the landscape. In other words, $\tilde{\mathcal{L}}_{\mathrm{Var}_v}$ and $\tilde{\mathcal{L}}_{\mathrm{Var}_v}^{\log}$ are equal to $\mathcal{L}_{\mathrm{Var}_v}$ and $\mathcal{L}_{\mathrm{Var}_v}^{\log}$ up to a constant term and a constant factor, respectively.*

*(ii) When $p_0$ is general, $\tilde{\mathcal{L}}_{\mathrm{Var}_v}$ and $\mathcal{L}_{\mathrm{Var}_v}$ have a different landscape, and the optimum of $\mathcal{L}_{\mathrm{Var}_v}$ may be different from $u^*$. A related loss that does preserve the optimum is:*

$$\bar{\mathcal{L}}_{\mathrm{Var}_v}(u) = \mathbb{E}[\mathrm{Var}_{\mathbb{P}^v}(\tfrac{d\mathbb{P}^{u^*}}{d\mathbb{P}^u}(X^v)|X_0^v)\exp(-\lambda^{-1}V(X_0^v,0))]$$
$$= \mathbb{E}[\mathrm{Var}(\exp(\tilde{Y}_T^{u,v} - \lambda^{-1}g(X_T^v))|X_0^v)].$$

*In practice, this is implemented by sampling the $m$ trajectories in one batch starting at the same point $X_0^v$.*

*(iii) Also, $\tilde{\mathcal{L}}_{\mathrm{Var}_v}^{\log}$ and $\mathcal{L}_{\mathrm{Var}_v}^{\log}$ have a different landscape, and the optimum of $\mathcal{L}_{\mathrm{Var}_v}^{\log}$ may be different from $u^*$. In particular, $\mathcal{L}_{\mathrm{Var}_v}^{\log}(u) = \mathrm{Var}_{\mathbb{P}^v}(\log \tfrac{d\mathbb{P}^{u^*}}{d\mathbb{P}^u}(X^v)\exp(-\lambda^{-1}V(X_0^v,0)))$. A loss that does preserve the optimum $u^*$ is*

$$\bar{\mathcal{L}}_{\mathrm{Var}_v}^{\log}(u) = \mathbb{E}[\mathrm{Var}_{\mathbb{P}^v}(\log \tfrac{d\mathbb{P}^{u^*}}{d\mathbb{P}^u}(X^v)|X_0^v)\exp(-\lambda^{-1}V(X_0^v,0))]$$
$$= \mathbb{E}[\mathrm{Var}(\tilde{Y}_T^{u,v} - \lambda^{-1}g(X_T^v)|X_0^v)].$$

*Proof.* Using (34) and (31), we have that

$$\tfrac{d\mathbb{P}^{u^*}}{d\mathbb{P}}(X^v) = \exp\big(\lambda^{-1}(V(X_0^v,0) - \mathcal{W}(X^v,0))\big),$$
$$\tfrac{d\mathbb{P}}{d\mathbb{P}^u}(X^v) = \exp\big(-\lambda^{-1/2}\int_0^T\langle u(X_t^v,t),dB_t^v\rangle + \tfrac{\lambda^{-1}}{2}\int_0^T\|u(X_t^v,t)\|^2\,dt\big)$$
$$= \exp\big(-\lambda^{-1/2}\int_0^T\langle u(X_t^v,t),dB_t\rangle - \lambda^{-1}\int_0^T\langle u(X_t^v,t),v(X_t^v,t)\rangle\,dt$$
$$+ \tfrac{\lambda^{-1}}{2}\int_0^T\|u(X_t^v,t)\|^2\,dt\big).$$

Hence,

$$\log\tfrac{d\mathbb{P}^{u^*}}{d\mathbb{P}^u}(X^v) = \log\tfrac{d\mathbb{P}^{u^*}}{d\mathbb{P}}(X^v) + \log\tfrac{d\mathbb{P}}{d\mathbb{P}^u}(X^v) = \tilde{Y}_T^{u,v} - \lambda^{-1}g(X_T^v) + \lambda^{-1}V(X_0^v,0).$$

Since $\tilde{\mathcal{L}}_{\mathrm{Var}_v}(u) = \mathrm{Var}_{\mathbb{P}^v}(\tfrac{d\mathbb{P}^{u^*}}{d\mathbb{P}^u})$ and $\tilde{\mathcal{L}}_{\mathrm{Var}_v}^{\log}(u) = \mathrm{Var}_{\mathbb{P}^v}(\log\tfrac{d\mathbb{P}^{u^*}}{d\mathbb{P}^u})$, this concludes the proof of (i).

To prove (ii), note that for general $p_0$, $V(X_0^v,0)$ is no longer a constant, but it is if we condition on $X_0^v$. The proof of (iii) is analogous. □

## C  Proofs of Sec. 3

### C.1  Proof of Thm. 1 and Prop. 2

We prove Thm. 1 and Prop. 2 at the same time. Recall that by (9), the optimal control is of the form $u^*(x,t) = -\sigma(t)^\top\nabla V(x,t)$. Consider the loss

$$\tilde{\mathcal{L}}(u) = \mathbb{E}\big[\tfrac{1}{T}\int_0^T\big\|u(X_t,t) + \sigma(t)^\top\nabla V(X_t,t)\big\|^2\,dt\,\exp\big(-\lambda^{-1}\int_0^T f(X_t,t)\,dt - \lambda^{-1}g(X_T)\big)\big].$$

Clearly, the unique optimum of $\tilde{\mathcal{L}}$ is $-\sigma(t)^\top\nabla V$. We can rewrite $\tilde{\mathcal{L}}$ as

$$\tilde{\mathcal{L}}(u) = \mathbb{E}\big[\tfrac{1}{T}\int_0^T\big(\|u(X_t,t)\|^2 + 2\langle u(X_t,t),\sigma(t)^\top\nabla V(X_t,t)\rangle + \|\sigma(t)^\top\nabla V(X_t,t)\|^2\big)\,dt \tag{44}$$
$$\times\exp\big(-\lambda^{-1}\int_0^T f(X_t,t)\,dt - \lambda^{-1}g(X_T)\big)\big].$$

Hence, we can express $\tilde{\mathcal{L}}$ as a sum of three terms: one involving $\|u(X_t,t)\|^2$, another involving $\langle u(X_t,t),\sigma(t)^\top V(X_t,t)\rangle$, and a third one, which is constant with respect to $u$, involving $\|\nabla V(X_t,t)\|^2$. The following lemma provides an alternative expression for the cross term:

**Lemma 5.** *The following equality holds:*

$$\mathbb{E}\big[\tfrac{1}{T}\int_0^T\langle u(X_t,t),\sigma(t)^\top\nabla V(X_t,t)\rangle\,dt\,\exp\big(-\lambda^{-1}\int_0^T f(X_t,t)\,dt - \lambda^{-1}g(X_T)\big)\big]$$
$$= -\lambda\mathbb{E}\big[\tfrac{1}{T}\int_0^T\langle u(X_t,t),\sigma(t)^\top\nabla_x\mathbb{E}\big[\exp\big(-\lambda^{-1}\int_t^T f(X_s,s)\,ds - \lambda^{-1}g(X_T)\big)\big|X_t = x\big]\rangle \tag{45}$$
$$\times\exp\big(-\lambda^{-1}\int_0^t f(X_s,s)\,ds\big)\,dt\big].$$

*Proof.* Recall the definition of $\mathcal{W}(X,t)$ in (35), which means that

$$\mathcal{W}(X,0) = \mathcal{W}(X,t) + \int_0^t f(X_s,s)\,\mathrm{d}s. \tag{46}$$

Let $\{\mathcal{F}_t\}_{t \in [0,T]}$ be the filtration generated by the Brownian motion $B$. Then, equation (9) implies that

$$\sigma(t)^\top \nabla V(X_t,t) = -\frac{\lambda\sigma(t)^\top \nabla_x \mathbb{E}\big[\exp\big(-\lambda^{-1}\mathcal{W}(X,t)\big)\big|\mathcal{F}_t\big]}{\mathbb{E}\big[\exp\big(-\lambda^{-1}\mathcal{W}(X,t)\big)\big|\mathcal{F}_t\big]} \tag{47}$$

We proceed as follows:

$$\mathbb{E}\big[\tfrac{1}{T}\int_0^T \langle u(X_t,t), \sigma(t)^\top \nabla V(X_t,t)\rangle\,\mathrm{d}t\,\exp\big(-\lambda^{-1}\mathcal{W}(X,0)\big)\big]$$

$$\overset{(i)}{=} -\lambda\mathbb{E}\big[\tfrac{1}{T}\int_0^T \Big\langle u(X_t,t), \frac{\sigma(t)^\top \nabla_x \mathbb{E}\big[\exp\big(-\lambda^{-1}\mathcal{W}(X,t)\big)\big|\mathcal{F}_t\big]}{\mathbb{E}\big[\exp\big(-\lambda^{-1}\mathcal{W}(X,t)\big)\big|\mathcal{F}_t\big]}\Big\rangle$$

$$\times \mathbb{E}\big[\exp\big(-\lambda^{-1}\mathcal{W}(X,t)\big)\big|\mathcal{F}_t\big]\exp\big(-\lambda^{-1}\int_0^t f(X_s,s)\,\mathrm{d}s\big)\,\mathrm{d}t\big]$$

$$= -\lambda\mathbb{E}\big[\tfrac{1}{T}\int_0^T \big\langle u(X_t,t), \sigma(t)^\top \nabla_x\mathbb{E}\big[\exp\big(-\lambda^{-1}\mathcal{W}(X,t)\big)\big|\mathcal{F}_t\big]\big\rangle\exp\big(-\lambda^{-1}\int_0^t f(X_s,s)\,\mathrm{d}s\big)\,\mathrm{d}t\big]$$

$$\overset{(ii)}{=} -\lambda\mathbb{E}\big[\tfrac{1}{T}\int_0^T \big\langle u(X_t,t), \sigma(t)^\top \nabla_x\mathbb{E}\big[\exp\big(-\lambda^{-1}\mathcal{W}(X,t)\big)\big|X_t=x\big]\big\rangle\exp\big(-\lambda^{-1}\int_0^t f(X_s,s)\,\mathrm{d}s\big)\,\mathrm{d}t\big].$$

Here, (i) holds by equation (47), the law of total expectation and equation (46), and (ii) holds by the Markov property of the solution of an SDE. □

The following proposition, which we prove in Subsec. C.2, provides an alternative expression for $\nabla_x\mathbb{E}\big[\exp\big(-\lambda^{-1}\int_t^T f(X_s,s)\,\mathrm{d}s - \lambda^{-1}g(X_T)\big)\big|X_t=x\big]$. The technique, which is novel and we denote by *Girsanov reparamaterization trick*, is of independent interest and may be applied in other settings, as we discuss in Sec. 5.

**Proposition 1** (Path-wise reparameterization trick for stochastic optimal control). *For each $t \in [0,T]$, let $M_t : [t,T] \to \mathbb{R}^{d\times d}$ be an arbitrary continuously differentiable function matrix-valued function such that $M_t(t) = \mathrm{Id}$. We have that*

$$\nabla_x\mathbb{E}\big[\exp\big(-\lambda^{-1}\int_t^T f(X_s,s)\,\mathrm{d}s - \lambda^{-1}g(X_T)\big)\big|X_t=x\big]$$

$$= \mathbb{E}\big[\big(-\lambda^{-1}\int_t^T M_t(s)\nabla_x f(X_s,s)\,\mathrm{d}s - \lambda^{-1}M_t(T)\nabla g(X_T)$$

$$+ \lambda^{-1/2}\int_t^T (M_t(s)\nabla_x b(X_s,s) - \partial_s M_t(s))(\sigma^{-1})^\top(s)\mathrm{d}B_s\big)$$

$$\times \exp\big(-\lambda^{-1}\int_t^T f(X_s,s)\,\mathrm{d}s - \lambda^{-1}g(X_T)\big)\big|X_t=x\big]. \tag{23}$$

Plugging (23) into the right-hand side of (45), we obtain that

$$\mathbb{E}\big[\tfrac{1}{T}\int_0^T \langle u(X_t,t), \sigma(t)^\top \nabla V(X_t,t)\rangle\,\mathrm{d}t\,\exp\big(-\lambda^{-1}\int_0^T f(X_t,t)\,\mathrm{d}t - \lambda^{-1}g(X_T)\big)\big]$$

$$= \mathbb{E}\big[\tfrac{1}{T}\int_0^T \big\langle u(X_t,t), \sigma(t)^\top\big(\int_t^T M_t(s)\nabla_x f(X_s,s)\,\mathrm{d}s + M_t(T)\nabla g(X_T)$$

$$- \lambda^{1/2}\int_t^T (M_t(s)\nabla_x b(X_s,s) - \partial_s M_t(s))(\sigma^{-1})^\top(s)\mathrm{d}B_s\big)\big\rangle\,\mathrm{d}t$$

$$\times \exp\big(-\lambda^{-1}\int_0^T f(X_t,t)\,\mathrm{d}t - \lambda^{-1}g(X_T)\big)\big].$$

If we plug this into the right-hand side of (44) and complete the squared norm, we get that

$$\tilde{\mathcal{L}}(u) = \mathbb{E}\big[\tfrac{1}{T}\int_0^T\big(\big\|u(X_t,t) - \tilde{w}(t,X,B,M_t)\big\|^2 - \big\|\tilde{w}(t,X,B,M_t)\big\|^2$$

$$+ \big\|u^*(X_t,t)\big\|^2\big)\,\mathrm{d}t\,\exp\big(-\lambda^{-1}\mathcal{W}(X,0)\big)\big]$$

where $\tilde{w}$ is defined as:

$$\tilde{w}(t,X,B,M_t) = \sigma(t)^\top\big(-\int_t^T M_t(s)\nabla_x f(X_s,s)\,\mathrm{d}s - M_t(T)\nabla g(X_T)$$

$$+ \lambda^{1/2}\int_t^T (M_t(s)\nabla_x b(X_s,s) - \partial_s M_t(s))(\sigma^{-1})^\top(s)\mathrm{d}B_s\big).$$

We also define $\Phi(u;X,B)$ as

$$\Phi(u;X,B) = \tfrac{1}{T}\int_0^T\big(\big\|u(X_t,t) - \tilde{w}(t,X,B,M_t)\big\|^2\big)\,\mathrm{d}t.$$

Now, by the Girsanov theorem (Thm. 3), we have that for an arbitrary control $v \in \mathcal{U}$,

$$\mathbb{E}[\Phi(u; X, B) \exp\left(-\lambda^{-1}\mathcal{W}(X, 0)\right)]$$
$$= \mathbb{E}\left[\Phi(u; X^v, B^v) \exp\left(-\lambda^{-1}\mathcal{W}(X^v, 0) - \lambda^{-1/2}\int_0^T \langle v(X_t^v, t), \mathrm{d}B_t^v\rangle + \frac{\lambda^{-1}}{2}\int_0^T \|v(X_t^v, t)\|^2 \, \mathrm{d}t\right)\right]$$
$$= \mathbb{E}\left[\Phi(u; X^v, B^v) \exp\left(-\lambda^{-1}\mathcal{W}(X^v, 0) - \lambda^{-1/2}\int_0^T \langle v(X_t^v, t), \mathrm{d}B_t\rangle - \frac{\lambda^{-1}}{2}\int_0^T \|v(X_t^v, t)\|^2 \, \mathrm{d}t\right)\right],$$

where $B_t^v := B_t + \lambda^{-1/2}\int_0^t v(X_s^v, s)\, \mathrm{d}s$. Reexpressing $B^v$ in terms of $B$, we can rewrite $\Phi(u; X^v, B^v)$ and $\tilde{w}(t, X^v, B^v, M_t)$ as follows:

$$\Phi(u; X^v, B^v) = \frac{1}{T}\int_0^T \left\| u(X_t^v, t) - \tilde{w}(t, X^v, B^v, M_t) \right\|^2 \mathrm{d}t,$$
$$\tilde{w}(t, X^v, B^v, M_t) = \sigma(t)^\top \Big( -\int_t^T M_t(s)\nabla_x f(X_s^v, s)\, \mathrm{d}s - M_t(T)\nabla g(X_T^v)$$
$$+ \lambda^{1/2}\int_t^T (M_t(s)\nabla_x b(X_s^v, s) - \partial_s M_t(s))(\sigma^{-1})^\top(X_s^v, s)\mathrm{d}B_s$$
$$+ \int_t^T (M_t(s)\nabla_x b(X_s^v, s) - \partial_s M_t(s))(\sigma^{-1})^\top(X_s^v, s)v(X_s^v, s)\mathrm{d}s \Big).$$

Putting everything together, we obtain that

$$\tilde{\mathcal{L}}(u) = \mathcal{L}_{\mathrm{SOCM}}(u, M) - K,$$

where $\mathcal{L}(u, M)$ is the loss defined in (19) (note that $w(t, v, X^v, B, M_t) := \tilde{w}(t, X^v, B^v, M_t)$), and

$$K = \mathbb{E}\left[\frac{1}{T}\int_0^T (\left\|\tilde{w}(t, X, B, M_t)\right\|^2 - \left\|u^*(X_t, t)\right\|^2)\, \mathrm{d}t \, \exp\left(-\lambda^{-1}\mathcal{W}(X, 0)\right)\right]$$

To complete the proof of equation (24), remark that $\tilde{\mathcal{L}}(u)$ can be rewritten as

$$\tilde{\mathcal{L}}(u) = \mathbb{E}\left[\frac{1}{T}\int_0^T \left\|u(X_t, t) - u^*(X_t, t)\right\|^2 \mathrm{d}t \, \exp\left(-\lambda^{-1}\mathcal{W}(X, 0)\right)\right]$$
$$= \mathbb{E}\left[\frac{1}{T}\int_0^T \left\|u(X_t, t) - u^*(X_t, t)\right\|^2 \mathrm{d}t \, \frac{d\mathbb{P}^{u^*}}{d\mathbb{P}}(X)\exp(-\lambda^{-1}V(X_0, 0))\right]$$
$$= \mathbb{E}\left[\frac{1}{T}\int_0^T \left\|u(X_t^{u^*}, t) - u^*(X_t^{u^*}, t)\right\|^2 \mathrm{d}t \, \exp(-\lambda^{-1}V(X_0^{u^*}, 0))\right].$$

It only remains to reexpress $K$. Note that by Prop. 1, we have that

$$u^*(X_t, t) = \frac{\mathbb{E}\left[\tilde{w}(t, X, B, M_t)\exp\left(-\lambda^{-1}\mathcal{W}(X, 0)\right)|\mathcal{F}_t\right]}{\mathbb{E}\left[\exp\left(-\lambda^{-1}\mathcal{W}(X, 0)\right)|\mathcal{F}_t\right]}$$
$$= \frac{\mathbb{E}\left[\tilde{w}(t, X, B, M_t)\frac{d\mathbb{P}^{u^*}}{d\mathbb{P}}(X)|\mathcal{F}_t\right]\exp(-\lambda^{-1}V(X_0, 0))}{\mathbb{E}\left[\frac{d\mathbb{P}^{u^*}}{d\mathbb{P}}(X)|\mathcal{F}_t\right]\exp(-\lambda^{-1}V(X_0, 0))} = \frac{\mathbb{E}\left[\tilde{w}(t, X, B, M_t)\frac{d\mathbb{P}^{u^*}}{d\mathbb{P}}(X)|\mathcal{F}_t\right]}{\mathbb{E}\left[\frac{d\mathbb{P}^{u^*}}{d\mathbb{P}}(X)|\mathcal{F}_t\right]}$$
$$= \mathbb{E}\left[\tilde{w}(t, X^{u^*}, B^{u^*}, M_t)|X_t^{u^*} = X_t\right]$$

Hence, using the Girsanov theorem (Thm. 3) several times, we have that

$$K = \mathbb{E}\left[\frac{1}{T}\int_0^T \left\|\tilde{w}(t, X^{u^*}, B^{u^*}, M_t)\right\|^2 - \left\|\mathbb{E}\left[\tilde{w}(t, X^{u^*}, B^{u^*}, M_t)|X_t^{u^*}\right]\right\|^2 \mathrm{d}t \, \exp(-\lambda^{-1}V(X_0^{u^*}, 0))\right]$$
$$\tag{48}$$
$$= \mathbb{E}\left[\frac{1}{T}\int_0^T \left\|\tilde{w}(t, X^{u^*}, B^{u^*}, M_t) - \mathbb{E}\left[\tilde{w}(t, X^{u^*}, B^{u^*}, M_t)|X_t^{u^*}\right]\right\|^2 \mathrm{d}t \, \exp(-\lambda^{-1}V(X_0^{u^*}, 0))\right]$$
$$= \mathbb{E}\left[\frac{1}{T}\int_0^T \left\|\tilde{w}(t, X, B, M_t) - \frac{\mathbb{E}[\tilde{w}(t, X, B, M_t)\exp(-\lambda^{-1}\mathcal{W}(X, 0))|X_t]}{\mathbb{E}[\exp(-\lambda^{-1}\mathcal{W}(X, 0))|X_t]}\right\|^2 \mathrm{d}t \, \exp(-\lambda^{-1}\mathcal{W}(X, 0))\right]$$
$$= \mathbb{E}\left[\frac{1}{T}\int_0^T \left\|w(t, v, X^v, B, M_t) - \frac{\mathbb{E}[w(t, v, X^v, B, M_t)\alpha(v, X^v, B)|X_t^v]}{\mathbb{E}[\alpha(v, X^v, B)|X_t^v]}\right\|^2 \mathrm{d}t \, \alpha(v, X^v, B)\right],$$

which concludes the proof, noticing that $K = \mathrm{Var}(w; M)$.

**Remark 1** (The optimal $M_t$ is the same for all $v$). *Looking at equation* (48)*, we observe that* $\mathrm{Var}(w; M)$ *does not depend on the base control $v$. Since minimizing $\mathcal{L}_{\mathrm{SOCM}}(u, M)$ with respect to $M$ is equivalent to minimizing $\mathrm{Var}(w; M)$, we deduce that the optimal $M$ does not depend on the vector field $v$.*

## C.2 Proof of the path-wise reparameterization trick (Prop. 1)

We prove a more general statement (Prop. 4), and show that Prop. 1 is a particular case of it.

**Proposition 4** (Path-wise reparameterization trick). *Let $(\Omega, \mathcal{F}, \mathbb{P})$ be a probability space, and $B : \Omega \times [0, T] \to \mathbb{R}^d$ be a Brownian motion. Let $X : \Omega \times [0, T] \to \mathbb{R}^d$ be the uncontrolled process given by (6), and let $\psi : \Omega \times \mathbb{R}^d \times [0, T] \to \mathbb{R}^d$ be an arbitrary random process such that:*

- *For all $z \in \mathbb{R}^d$, the process $\psi(\cdot, z, \cdot) : \Omega \times [0, T] \to \mathbb{R}^d$ is adapted to the filtration $(\mathcal{F}_s)_{s \in [0,T]}$ of the Brownian motion $B$.*

- *For all $\omega \in \Omega$, $\psi(\omega, \cdot, \cdot) : \mathbb{R}^d \times [0, T] \to \mathbb{R}^d$ is a twice-continuously differentiable function such that $\psi(\omega, z, 0) = z$ for all $z \in \mathbb{R}^d$, and $\psi(\omega, 0, s) = 0$ for all $s \in [0, T]$.*

*Let $F : C([0, T]; \mathbb{R}^d) \to \mathbb{R}$ be a Fréchet-differentiable functional. We use the notation $X + \psi(z, \cdot) = (X_s(\omega) + \psi(\omega, z, s))_{s \in [0,T]}$ to denote the shifted process, and we will omit the dependency of $\psi$ on $\omega$ in the proof. Then,*

$$\nabla_x \mathbb{E}\big[ \exp\big( - F(X) \big) \big| X_0 = x \big] \tag{49}$$
$$= \mathbb{E}\Big[\Big( - \nabla_z F(X + \psi(z, \cdot))\big|_{z=0} + \lambda^{-1/2} \int_0^T (\nabla_z \psi(0, s) \nabla_x b(X_s, s) - \nabla_z \partial_s \psi(0, s))(\sigma^{-1})^\top (s) \mathrm{d}B_s\Big)$$

$$\times \exp\big( - F(X) \big) \big| X_0 = x \Big]$$

*Proof of Prop. 1.* Given a family of functions $(M_t)_{t \in [0,T]}$ satisfying the conditions in Prop. 1, we can define a family $(\psi_t)_{t \in [0,T]}$ of functions $\psi_t : \mathbb{R}^d \times [t, T] \to \mathbb{R}^d$ as $\psi_t(z, s) = M_t(s)^\top z$. Note that $\psi_t(z, t) = z$ for all $z \in \mathbb{R}^d$ and $\psi_t(0, s) = 0$ for all $s \in [t, T]$, and that $\nabla_z \psi_t(z, s) = M_t(s)$. Hence, $\psi_t$ can be seen as a random process which is constant with respect to $\omega \in \Omega$, and which fulfills the conditions in Prop. 4 up to a trivial time change of variable from $[t, T]$ to $[0, T]$.

We also define the family $(F_t)_{t \in [0,T]}$ of functionals $F_t : C([t, T]; \mathbb{R}^d) \to \mathbb{R}$ as $F_t(X) = \lambda^{-1} \int_t^T f(X_s, s) \, \mathrm{d}s + \lambda^{-1} g(X_T)$. We have that

$$\nabla_z F_t(X + \psi_t(z, \cdot))$$
$$= \nabla_z \big(\lambda^{-1} \int_t^T f(X_s + \psi_t(z, s), s) \, \mathrm{d}s + \lambda^{-1} g(X_T + \psi_t(z, T))\big)$$
$$\overset{(i)}{=} \lambda^{-1} \int_t^T \nabla_z \psi_t(z, s) \nabla f(X_s + \psi_t(z, s), s) \, \mathrm{d}s + \lambda^{-1} \nabla_z \psi_t(z, T) \nabla g(X_T + \psi_t(z, T))$$
$$= \lambda^{-1} \int_t^T M_t(s) \nabla f(X_s + \psi_t(z, s), s) \, \mathrm{d}s + \lambda^{-1} M_t(T) \nabla g(X_T + \psi_t(z, T)),$$

where equality (i) holds by the Leibniz rule. Using that $\psi_t(0, s) = 0$, we obtain that:

$$\nabla_z F_t(X + \psi_t(z, \cdot))\big|_{z=0} = \lambda^{-1} \int_t^T \nabla_z \psi_t(0, s) \nabla f(X_s, s) \, \mathrm{d}s + \lambda^{-1} \nabla_z \psi_t(T, 0) \nabla g(X_T),$$

Up to a trivial time change of variable from $[t, T]$ to $[0, T]$, Prop. 1 follows from plugging these choices into equation (49).

**Remark 2.** *We can use matrices $M_t(s)$ that depend on the process $X$ up to time $s$, since the resulting processes $\psi_t(\cdot, z, \cdot)$ are adapted to the filtration of the Brownian motion $B$. More specifically, if we let $M_t : \mathbb{R}^d \times [t, T] \to \mathbb{R}^{d \times d}$ be an arbitrary continuously differentiable function matrix-valued function such that $M_t(x, t) = \mathrm{Id}$ for all $x \in \mathbb{R}^d$, and we define the exponential moving average of $X$ as the process $X^{(v)}$ given by*

$$X_t^{(v)} = v \int_0^t e^{-v(t-s)} X_s \, \mathrm{d}s,$$

*we have that*

$$\frac{\mathrm{d}}{\mathrm{d}s} M_t(X_s^{(v)}, s) = \langle \nabla M_t(X_s^{(v)}, s), \frac{\mathrm{d}X_s^{(v)}}{\mathrm{d}s} \rangle + \partial_s M_t(X_s^{(v)}, s) = \lambda \langle \nabla_x M_t(X_s^{(v)}, s), X_s - X_s^{(v)} \rangle + \partial_s M_t(X_s^{(v)}, s),$$

*and we can write*

$$
\begin{aligned}
&\nabla_x \mathbb{E}\big[\exp\big(-\lambda^{-1}\textstyle\int_t^T f(X_s,s)\,\mathrm{d}s - \lambda^{-1}g(X_T)\big)\big|X_t = x\big] \\
&= \mathbb{E}\big[\big(-\lambda^{-1}\textstyle\int_t^T M_t(X_s^{(v)},s)\nabla_x f(X_s,s)\,\mathrm{d}s - \lambda^{-1}M_t(X_T^{(v)},T)\nabla g(X_T) \\
&\qquad + \lambda^{-1/2}\textstyle\int_t^T (M_t(X_s^{(v)},s)\nabla_x b(X_s,s) - \tfrac{\mathrm{d}}{\mathrm{d}s}M_t(X_s^{(v)},s))(\sigma^{-1})^\top(s)\mathrm{d}B_s\big) \\
&\qquad \times \exp\big(-\lambda^{-1}\textstyle\int_t^T f(X_s,s)\,\mathrm{d}s - \lambda^{-1}g(X_T)\big)\big|X_t = x\big].
\end{aligned}
$$

*Plugging this into the proof of Thm. 1, we would obtain a variant of SOCM (Alg. 2) where the matrix-valued neural network $M_\omega$ takes inputs $(t,s,x)$ instead of $(t,s)$. Since the optimization class is larger, from the bias-variance in Prop. 2 we deduce that this variant would yield a lower variance of the vector field $w$, and likely an algorithm with lower error. This is at the expense of an increased number of function evaluations (NFE) of $M_\omega$; one would need $\frac{K(K+1)m}{2}$ NFE per batch instead of only $\frac{K(K+1)}{2}$, which may be too expensive if the architecture of $M_\omega$ is large. A way to speed up the computation per batch is to parameterize $M_t$ using cubic splines.*

$\square$

*Proof of Prop. 4.* Recall that

$$
\mathrm{d}X_s = b(X_s,s)\,\mathrm{d}s + \sqrt{\lambda}\sigma(s)\,\mathrm{d}B_s, \qquad X_0 \sim p_0,
$$

is the SDE for the uncontrolled process. For arbitrary $x,z \in \mathbb{R}^d$, we consider the following SDEs conditioned on the initial points:

$$
\mathrm{d}X_s^{(x+z)} = b(X_s^{(x+z)},s)\,\mathrm{d}s + \sqrt{\lambda}\sigma(s)\,\mathrm{d}B_s, \qquad X_0^{(x+z)} = x+z, \tag{50}
$$

$$
\mathrm{d}X_s^{(x)} = b(X_s^{(x)},s)\,\mathrm{d}s + \sqrt{\lambda}\sigma(s)\,\mathrm{d}B_s, \qquad X_0^{(x)} = x. \tag{51}
$$

Suppose that $\psi : \mathbb{R}^d \times [0,T] \to \mathbb{R}^d$ satisfies the properties in the statement of Prop. 4. If $\tilde{X}^{(x)}$ is a solution of

$$
\mathrm{d}\tilde{X}_s^{(x)} = (b(\tilde{X}_s^{(x)} + \psi(z,s),s) - \partial_s\psi(z,s))\,\mathrm{d}s + \sqrt{\lambda}\sigma(s)\,\mathrm{d}B_s, \qquad \tilde{X}_0^{(x)} = x,
$$

then $X^{(x+z)} = \tilde{X}^{(x)} + \psi(z,\cdot)$ is a solution of (50). This is because $X_0^{(x+z)} = \tilde{X}_0^{(x)} + \psi(z,0) = \tilde{X}_0^{(x)} + z = x + z$, and

$$
\begin{aligned}
\mathrm{d}X_s^{(x+z)} &= \mathrm{d}\tilde{X}_s^{(x)} + \partial_s\psi(z,s)\,\mathrm{d}s \\
&= (b(\tilde{X}_s^{(x)} + \psi(z,s),s) - \partial_s\psi(z,s))\,\mathrm{d}s + \sqrt{\lambda}\sigma(s)\,\mathrm{d}B_s + \partial_s\psi(z,s)\,\mathrm{d}s \\
&= b(X_s^{(x+z)},s)\,\mathrm{d}s + \sqrt{\lambda}\sigma(s)\,\mathrm{d}B_s,
\end{aligned}
$$

Note that we may rewrite (51) as

$$
\begin{aligned}
\mathrm{d}X_s^{(x)} &= (b(X_s^{(x)} + \psi(z,s),s) - \partial_s\psi(z,s))\,\mathrm{d}s \\
&\quad + (b(X_s^{(x)},s) - b(X_s^{(x)} + \psi(z,s),s) + \partial_s\psi(z,s))\,\mathrm{d}s + \sqrt{\lambda}\sigma(s)\,\mathrm{d}B_s, \qquad X_t^{(x)} \sim p_0.
\end{aligned}
$$

Hence, since $\psi(z,s)$ is a random process adapted to the filtration of $B$, we can apply the Girsanov theorem for SDEs (Corollary 1) on $\tilde{X}^{(x)}$ and $X^{(x)}$, and we have that for any bounded continuous functional $\Phi$,

$$
\begin{aligned}
&\mathbb{E}[\Phi(\tilde{X}^{(x)})] \\
&= \mathbb{E}\big[\Phi(X^{(x)})\exp\big(\textstyle\int_0^T \lambda^{-1/2}\sigma(s)^{-1}(b(X_s^{(x)} + \psi(z,s),s) - b(X_s^{(x)},s) - \partial_s\psi(z,s))\,\mathrm{d}B_s \\
&\qquad\qquad - \tfrac{1}{2}\textstyle\int_0^T \|\lambda^{-1/2}\sigma(s)^{-1}(b(X_s^{(x)} + \psi(z,s),s) - b(X_s^{(x)},s) - \partial_s\psi(z,s))\|^2\,\mathrm{d}s\big)\big].
\end{aligned}
\tag{52}
$$

We can write

$$\mathbb{E}\big[\exp\big(-F(X)\big)\big|X_0 = x + z\big] \overset{\text{(i)}}{=} \mathbb{E}\big[\exp\big(-F(X^{(x+z)})\big)\big] \overset{\text{(ii)}}{=} \mathbb{E}\big[\exp\big(-F(\tilde{X}^{(x)} + \psi(z, \cdot))\big)\big]$$

$$\overset{\text{(iii)}}{=} \mathbb{E}\big[\exp\big(-F(X^{(x)} + \psi(z, \cdot))\big)$$

$$\times \exp\big(\int_0^T \lambda^{-1/2}\sigma(s)^{-1}(b(X_s^{(x)} + \psi(z, s), s) - b(X_s^{(x)}, s) - \partial_s\psi(z, s))\,\mathrm{d}B_s$$

$$-\tfrac{1}{2}\int_0^T \|\lambda^{-1/2}\sigma(s)^{-1}(b(X_s^{(x)} + \psi(z, s), s) - b(X_s^{(x)}, s) - \partial_s\psi(z, s))\|^2\,\mathrm{d}s\big)\big]$$

$$\overset{\text{(iv)}}{=} \mathbb{E}\big[\exp\big(-F(X + \psi(z, \cdot)) + \int_0^T \lambda^{-1/2}\sigma(s)^{-1}(b(X_s + \psi(z, s), s) - b(X_s, s) - \partial_s\psi(z, s))\,\mathrm{d}B_s$$

$$-\tfrac{1}{2}\int_0^T \|\lambda^{-1/2}\sigma(s)^{-1}(b(X_s + \psi(z, s), s) - b(X_s, s) - \partial_s\psi(z, s))\|^2\,\mathrm{d}s\big)\big|X_0 = x\big]$$
$$(53)$$

Equality (i) holds by the definition of $X^{(x+z)}$, equality (ii) holds by the fact $X_s^{(x+z)} = \tilde{X}_s^{(x)} + \psi(z, s)$, equality (iii) holds by equation (52), and equality (iv) holds by the definition of $X_s^{(x)}$. We conclude the proof by differentiating the right-hand side of (53) with respect to $z$. Namely,

$$\nabla_x\mathbb{E}\big[\exp\big(-F(X)\big)\big|X_0 = x\big] = \nabla_z\mathbb{E}\big[\exp\big(-F(X)\big)\big|X_0 = x + z\big]\big|_{z=0}$$

$$\overset{\text{(i)}}{=} \mathbb{E}\big[\big(-\nabla_z F(X + \psi(z, \cdot)) + \lambda^{-1/2}\int_0^T (\nabla_z\psi(0, s)\nabla_x b(X_s, s) - \nabla_z\partial_s\psi(0, s))(\sigma^{-1})^\top(s)\mathrm{d}B_s\big)$$

$$\times \exp\big(-F(X)\big)\big|X_0 = x\big]$$

In equality (i) we used (53), and that:

- by the Leibniz rule,

$$\nabla_z\int_0^T \|\sigma(s)^{-1}(b(X_s + \psi(z, s), s) - b(X_s, s) - \partial_s\psi(z, s))\|^2\,\mathrm{d}s\big|_{z=0}$$

$$= \int_0^T \nabla_z\|\sigma(s)^{-1}(b(X_s + \psi(z, s), s) - b(X_s, s) - \partial_s\psi(z, s))\|^2\big|_{z=0}\,\mathrm{d}s = 0.$$

- and by the Leibniz rule for stochastic integrals (see [42]),

$$\nabla_z\big(\int_0^T \sigma(s)^{-1}(b(X_s + \psi(z, s), s) - b(X_s, s) - \partial_s\psi(z, s))\,\mathrm{d}B_s\big)\big|_{z=0}$$

$$= \int_0^T (\nabla_z\psi(0, s)\nabla_x b(X_s, s) - \nabla_z\partial_s\psi(0, s))(\sigma^{-1})^\top(s)\,\mathrm{d}B_s.$$

$\square$

## C.3  Informal derivation of the path-wise reparameterization trick

In this subsection, we provide an informal, intuitive derivation of the path-wise reparameterization trick as stated in Prop. 4. For simplicity, we particularize the functional $F$ to $F(X) = \lambda^{-1}\int_0^T f(X_s, s)\,\mathrm{d}s + \lambda^{-1}g(X_T)$. Consider the Euler-Maruyama discretization of the uncontrolled process $X$ defined in (6), with $K + 1$ time steps (let $\delta = T/K$ be the step size). This is a family of random variables $\hat{X} = (\hat{X}_k)_{k=0:K}$ defined as

$$\hat{X}_0 \sim p_0, \qquad \hat{X}_{k+1} = \hat{X}_k + \delta b(\hat{X}_k, k\delta) + \sqrt{\delta\lambda}\sigma(k\delta)\varepsilon_k, \qquad \varepsilon_k \sim N(0, I).$$

Note that we can approximate

$$\mathbb{E}\big[\exp\big(-\lambda^{-1}\int_0^T f(X_s, s)\,\mathrm{d}s - \lambda^{-1}g(X_T)\big)\big|X_0 = x\big]$$

$$\approx \mathbb{E}\big[\exp\big(-\lambda^{-1}\delta\sum_{k=0}^{K-1} f(\hat{X}_k, s) - \lambda^{-1}g(\hat{X}_K)\big)\big|\hat{X}_0 = x\big],$$

and that this is an equality in the limit $K \to \infty$, as the interpolation of the Euler-Maruyama discretization $\hat{X}^{(x)}$ converges to the process $X^{(x)}$. Now, remark that for $k \in \{0, \ldots, K - 1\}$, $\hat{X}_{k+1}|\hat{X}_k \sim N(\hat{X}_k + \delta b(\hat{X}_k, k\delta), \delta\lambda(\sigma\sigma^\top)(k\delta))$. Hence,

$$\mathbb{E}\big[\exp\big(-\lambda^{-1}\delta\sum_{k=0}^{K-1} f(\hat{X}_k, s) - \lambda^{-1}g(\hat{X}_K)\big)\big|\hat{X}_0 = x\big]$$

$$= C^{-1}\iint_{(\mathbb{R}^d)^K} \exp\big(-\lambda^{-1}\delta\sum_{k=0}^{K-1} f(\hat{x}_k, s) - \lambda^{-1}g(\hat{x}_K)$$

$$-\tfrac{1}{2\delta\lambda}\sum_{k=1}^{K-1} \|\sigma^{-1}(k\delta)(\hat{x}_{k+1} - \hat{x}_k - \delta b(\hat{x}_k, k\delta))\|^2$$

$$-\tfrac{1}{2\delta\lambda}\|\sigma^{-1}(0)(\hat{x}_1 - x - \delta b(x, 0))\|^2\big)\,\mathrm{d}\hat{x}_1\cdots\mathrm{d}\hat{x}_K,$$

where $C = \sqrt{(2\pi\delta\lambda)^K \prod_{k=0}^{K-1} \det((\sigma\sigma^\top)(k\delta))}$. Now, let $\psi : \mathbb{R}^d \times [0,T] \to \mathbb{R}^d$ be an arbitrary twice differentiable function such that $\psi(z,0) = z$ for all $z \in \mathbb{R}^d$, and $\psi(0,s) = 0$ for all $s \in [0,T]$. We can write

$$\nabla_x \mathbb{E}\big[\exp\big(-\lambda^{-1}\delta \textstyle\sum_{k=0}^{K-1} f(\hat{X}_k, s) - \lambda^{-1} g(\hat{X}_K))|\hat{X}_0 = x\big]$$
$$= \nabla_z \mathbb{E}\big[\exp\big(-\lambda^{-1}\delta \textstyle\sum_{k=0}^{K-1} f(\hat{X}_k, s) - \lambda^{-1} g(\hat{X}_K))|\hat{X}_0 = x + z\big]|_{z=0}$$
$$= C^{-1}\nabla_z\big(\textstyle\iint_{(\mathbb{R}^d)^K} \exp\big(-\lambda^{-1}\delta \textstyle\sum_{k=0}^{K-1} f(\hat{x}_k, s) - \lambda^{-1} g(\hat{x}_K)$$
$$- \tfrac{1}{2\delta\lambda} \textstyle\sum_{k=1}^{K-1} \|\sigma^{-1}(k\delta)(\hat{x}_{k+1} - \hat{x}_k - \delta b(\hat{x}_k, k\delta))\|^2$$
$$- \tfrac{1}{2\delta\lambda}\|\sigma^{-1}(0)(\hat{x}_1 - (x+z) - \delta b(x+z, 0))\|^2\big)\, \mathrm{d}\hat{x}_1 \cdots \mathrm{d}\hat{x}_K)|_{z=0}$$
$$= C^{-1}\nabla_z\big(\textstyle\iint_{(\mathbb{R}^d)^K} \exp\big(-\lambda^{-1}\delta \textstyle\sum_{k=0}^{K-1} f(\hat{x}_k + \psi(z, k\delta), s) - \lambda^{-1} g(\hat{x}_K + \psi(z, K\delta))$$
$$- \tfrac{1}{2\delta\lambda}\textstyle\sum_{k=1}^{K-1} \|\sigma^{-1}(k\delta)(\hat{x}_{k+1} + \psi(z, (k+1)\delta) - \hat{x}_k - \psi(z, k\delta) - \delta b(\hat{x}_k + \psi(z, k\delta), k\delta))\|^2$$
$$- \tfrac{1}{2\delta\lambda}\|\sigma^{-1}(0)(\hat{x}_1 + \psi(z,\delta) - (x + \psi(z,0)) - \delta b(x + \psi(z,0), 0))\|^2\big)\, \mathrm{d}\hat{x}_1 \cdots \mathrm{d}\hat{x}_K)|_{z=0},$$

$$(54)$$

In the last equality, we used that for $k \in \{1, \ldots, K\}$, the variables $\hat{x}_k$ are integrated over $\mathbb{R}^d$, which means that adding an offset $\psi(z, k\delta)$ does not change the value of the integral. We also used that $\psi(z, 0) = z$. Now, for fixed values of $\hat{x} = (\hat{x}_1, \ldots, \hat{x}_K)$, and letting $\hat{x}_0 = x$, we define

$$G_{\hat{x}}(z) = \lambda^{-1}\delta \textstyle\sum_{k=0}^{K-1} f(\hat{x}_k + \psi(z, k\delta), s) + \lambda^{-1} g(\hat{x}_K + \psi(z, K\delta))$$
$$+ \tfrac{1}{2\delta\lambda} \textstyle\sum_{k=0}^{K-1} \|\sigma^{-1}(k\delta)(\hat{x}_{k+1} + \psi(z, (k+1)\delta) - \hat{x}_k - \psi(z, k\delta) - \delta b(\hat{x}_k + \psi(z, k\delta), k\delta))\|^2.$$

Using that $\psi(0, s) = 0$ for all $s \in [0, T]$, we have that:

$$G_{\hat{x}}(0) = \lambda^{-1}\delta \textstyle\sum_{k=0}^{K-1} f(\hat{x}_k, s) + \lambda^{-1} g(\hat{x}_K) + \tfrac{1}{2\delta\lambda} \textstyle\sum_{k=0}^{K-1} \|\sigma^{-1}(k\delta)(\hat{x}_{k+1} - \hat{x}_k - \delta b(\hat{x}_k, k\delta))\|^2.$$

$$\nabla G_{\hat{x}}(z)|_{z=0} = \lambda^{-1}\delta \textstyle\sum_{k=0}^{K-1} \nabla\psi(0, k\delta)\nabla f(\hat{x}_k, s) + \lambda^{-1}\nabla\psi(0, K\delta)\nabla g(\hat{x}_K)$$
$$+ \tfrac{1}{\delta\lambda} \textstyle\sum_{k=0}^{K-1} (\nabla_z\psi(0, (k+1)\delta) - \nabla_z\psi(0, k\delta) - \delta\nabla\psi(0, k\delta)\nabla b(\hat{x}_k, k\delta))$$
$$\times ((\sigma^{-1})^\top \sigma^{-1})(k\delta)(\hat{x}_{k+1} - \hat{x}_k - \delta b(\hat{x}_k, k\delta))$$

And we can express the right-hand side of (54) in terms of $G_{\hat{x}}(0)$ and $\nabla G_{\hat{x}}(z)|_{z=0}$:

$$\nabla_z\big(C^{-1}\textstyle\iint_{(\mathbb{R}^d)^K} \exp\big(-G_{\hat{x}}(z)\big)\, \mathrm{d}y_1 \cdots \mathrm{d}y_K\big)$$
$$= -C^{-1}\textstyle\iint_{(\mathbb{R}^d)^K} \nabla G_{\hat{x}}(z)|_{z=0} \exp\big(-G_{\hat{x}}(0)\big)\, \mathrm{d}y_1 \cdots \mathrm{d}y_K$$

We define $\epsilon_k = \tfrac{1}{\sqrt{\delta\lambda}}\sigma^{-1}(k\delta)(\hat{x}_{k+1} - \hat{x}_k - \delta b(\hat{x}_k, k\delta))$, and then, we are able to write

$$\hat{x}_{k+1} = \hat{x}_k + \delta b(\hat{x}_k, k\delta) + \sqrt{\delta\lambda}\sigma(k\delta)\epsilon_k, \qquad \hat{x}_0 = x \qquad (55)$$
$$G_{\hat{x}}(0) = \lambda^{-1}\delta \textstyle\sum_{k=0}^{K-1} f(\hat{x}_k, s) + \lambda^{-1} g(\hat{x}_K) + \tfrac{1}{2}\textstyle\sum_{k=0}^{K-1} \|\epsilon_k\|^2,$$
$$\nabla G_{\hat{x}}(z)|_{z=0} = \lambda^{-1}\delta \textstyle\sum_{k=0}^{K-1} \nabla\psi(0, k\delta)\nabla f(\hat{x}_k, s) + \lambda^{-1}\nabla\psi(0, K\delta)\nabla g(\hat{x}_K)$$
$$+ \sqrt{\delta\lambda^{-1}}\textstyle\sum_{k=0}^{K-1} (\partial_s\nabla_z\psi(0, k\delta) + O(\delta) - \nabla\psi(0, k\delta)\nabla b(\hat{x}_k, k\delta))(\sigma^{-1})^\top(k\delta)\epsilon_k.$$

$$(56)$$

Then, taking the limit $K \to \infty$ (i.e. $\delta \to 0$), we recognize (55) as Euler-Maruyama discretization of the uncontrolled process $X$ in equation (6) conditioned on $X_0 = x$, and the last term in (56) as the Euler-Maruyama discretization of the stochastic integral $\lambda^{-1/2}\int_0^T (\partial_s\nabla_z\psi(0, s) - \nabla\psi(0, s)\nabla b(X_s^{(x)}, s))(\sigma^{-1})^\top(s)\, \mathrm{d}B_s$. Thus,

$$\lim_{K\to\infty} \nabla_x \mathbb{E}\big[\exp\big(-\lambda^{-1}\delta \textstyle\sum_{k=0}^{K-1} f(\hat{X}_k, s) - \lambda^{-1} g(\hat{X}_K))\big]$$
$$= \mathbb{E}\big[\big(-\lambda^{-1}\textstyle\int_0^T \nabla\psi(0, s)\nabla_x f(X_s, s)\, \mathrm{d}s - \lambda^{-1}\nabla\psi(0, T)\nabla g(X_T)$$
$$+ \lambda^{-1/2}\textstyle\int_0^T (\nabla\psi(0, s)\nabla_x b(X_s, s) - \partial_s\nabla\psi(0, s))(\sigma^{-1})^\top(s)\, \mathrm{d}B_s\big)$$
$$\times \exp\big(-\lambda^{-1}\textstyle\int_0^T f(X_s, s)\, \mathrm{d}s - \lambda^{-1} g(X_T))|X_0 = x\big],$$

which concludes the derivation.

## C.4 SOCM-Adjoint: replacing the path-wise reparameterization trick with the adjoint method

**Proposition 5.** *Let* $\mathcal{L}_{\mathrm{SOCM-Adj}} : L^2(\mathbb{R}^d \times [0, T]; \mathbb{R}^d) \to \mathbb{R}$ *be the loss function defined as*

$$\mathcal{L}_{\mathrm{SOCM-Adj}}(u) := \mathbb{E}\Big[\tfrac{1}{T} \int_0^T \big\| u(X_t^v, t) + \sigma(t)^\top a(t, X^v) \big\|^2 \, dt \times \alpha(v, X^v, B)\Big] \, ,$$

*where* $X^v$ *is the process controlled by* $v$ *(i.e.,* $dX_t = (b(X_t, t) + \sigma(t)v(X_t, t)) \, dt + \sqrt{\lambda}\sigma(X_t, t) \, dB_t$ *and* $X_0 \sim p_0$*),* $\alpha(v, X^v, B)$ *is the importance weight defined in* (20)*, and* $a(t, X^v)$ *is the solution of the ODE*

$$\frac{da(t)}{dt} = -\nabla_x b(X_t^v, t)a(t) - \nabla_x f(X_t^v, t),$$
$$a(T) = \nabla g(X_T^v),$$

$\mathcal{L}_{\mathrm{SOCM-Adj}}$ *has a unique optimum, which is the optimal control* $u^*$.

*Proof.* The proof follows the same structure as that of Thm. 1. Instead of plugging the path-wise reparameterization trick (Prop. 1) in the right-hand side of (22), we make use of Lemma 6 to evaluate $\nabla_x \mathbb{E}\big[ \exp\big( -\lambda^{-1} \int_0^T f(X_t, t) \, dt - \lambda^{-1} g(X_T) \big) | X_0 = x \big]$. Particular cases of the result in Lemma 6 have been used in previous works such as [54, 51]. We present a more general form that covers state costs $f$, as well as stochastic integrals. We also present a simpler proof of the result based on Lagrange multipliers. □

**Lemma 6** (Adjoint method for SDEs). *[54, 51] Let* $X : \Omega \times [0, T] \to \mathbb{R}^d$ *be the uncontrolled process defined in* (6)*, with initial condition* $X_0 = x$*. We define the random process* $a : \Omega \times [0, T] \to \mathbb{R}^d$ *such that for all* $\omega \in \Omega$*, using the short-hand* $a(t) := a(\omega, t)$*,*

$$da_t(\omega) = \big( -\nabla_x b(X_t(\omega), t)a_t(\omega) - \nabla_x f(X_t(\omega), t) \big) \, dt - \nabla_x h(X_t(\omega), t) \, dB_t,$$
$$a_T(\omega) = \nabla_x g(X_T(\omega)),$$

*we have that*

$$\nabla_x \mathbb{E}\big[ \int_0^T f(X_t(\omega), t) \, dt + \int_0^T \langle h(X_t(\omega), t), \, dB_t \rangle + g(X_T(\omega)) | X_0(\omega) = x \big] = \mathbb{E}\big[ a_0(\omega) \big],$$

$$\nabla_x \mathbb{E}\big[ \exp\big( -\int_0^T f(X_t(\omega), t) \, dt - \int_0^T \langle h(X_t(\omega), t), \, dB_t \rangle - g(X_T(\omega)) \big) | X_0(\omega) = x \big]$$
$$= -\mathbb{E}\big[ a_0(\omega) \exp\big( -\int_0^T f(X_t(\omega), t) \, dt - \int_0^T \langle h(X_t(\omega), t), \, dB_t \rangle - g(X_T(\omega)) \big) | X_0(\omega) = x \big].$$

*Proof.* We will use an approach based on Lagrange multipliers. Define a process $a : \Omega \times [0, T] \to \mathbb{R}^d$ such that for any $\omega \in \Omega$, $a(\omega, \cdot)$ is differentiable. For a given $\omega \in \Omega$, we can write

$$\int_0^T f(X_t(\omega), t) \, dt + \int_0^T \langle h(X_t(\omega), t), \, dB_t \rangle + g(X_T(\omega))$$
$$= \int_0^T f(X_t(\omega), t) \, dt + \int_0^T \langle h(X_t(\omega), t), \, dB_t \rangle + g(X_T(\omega))$$
$$- \int_0^T \langle a_t(\omega), (dX_t(\omega) - b(X_t(\omega), t) \, dt - \sigma(t) \, dB_t) \rangle.$$

By Lemma 7, we have that

$$\int_0^T \langle a_t(\omega), dX_t(\omega) \rangle = \langle a_T(\omega), X_T(\omega) \rangle - \langle a_0(\omega), X_0(\omega) \rangle - \int_0^T \langle X_t(\omega), \tfrac{da_t}{dt}(\omega) \rangle \, dt.$$

Hence,

$$\nabla_x \big( \int_0^T f(X_t(\omega), t) \, dt + \int_0^T \langle h(X_t(\omega), t), \, dB_t \rangle + g(X_T(\omega)) \big)$$
$$= \nabla_x \big( \int_0^T f(X_t(\omega), t) \, dt + \int_0^T \langle h(X_t(\omega), t), \, dB_t \rangle + g(X_T(\omega))$$
$$- \langle a_T(\omega), X_T(\omega) \rangle + \langle a_0(\omega), X_0(\omega) \rangle + \int_0^T \big( \langle a_t(\omega), b(X_t(\omega), t) \rangle + \langle \tfrac{da_t}{dt}(\omega), X_t(\omega) \rangle \big) \, dt$$
$$+ \int_0^T \langle a_t(\omega), \sigma(t) \, dB_t \rangle \big)$$
$$= \int_0^T \nabla_x X_t(\omega) \nabla_x f(X_t(\omega), t) \, dt + \int_0^T \nabla_x X_t(\omega) \nabla_x h(X_t(\omega), t) \, dB_t + \nabla_x X_T(\omega) \nabla_x g(X_T(\omega))$$
$$- \nabla_x X_T(\omega) a_T(\omega) + \nabla_x X_0(\omega) a_0(\omega)$$
$$+ \int_0^T \big( \nabla_x X_t(\omega) \nabla_x b(X_t(\omega), t) a_t(\omega) + \nabla_x X_t(\omega) \tfrac{da_t}{dt}(\omega) \big) \, dt$$
$$= \int_0^T \nabla_x X_t(\omega) \big( \nabla_x f(X_t(\omega), t) + \nabla_x b(X_t(\omega), t) a_t(\omega) + \tfrac{da_t}{dt}(\omega) \big) \, dt$$
$$+ \nabla_x X_T(\omega) \big( \nabla_x g(X_T(\omega)) - a_T(\omega) \big) + a_0(\omega) + \int_0^T \nabla_x X_t(\omega) \nabla_x h(X_t(\omega), t) \, dB_t.$$

In the last line we used that $\nabla_x X_0(\omega) = \nabla_x x = I$. If choose $a$ such that

$$da_t(\omega) = \big( - \nabla_x b(X_t(\omega), t) a_t(\omega) - \nabla_x f(X_t(\omega), t) \big) \, dt - \nabla_x h(X_t(\omega), t) \, dB_t,$$
$$a_T(\omega) = \nabla_x g(X_T(\omega)),$$

then we obtain that

$$\nabla_x \big( \int_0^T f(X_t(\omega), t) \, dt + \int_0^T \langle h(X_t(\omega), t), \, dB_t \rangle + g(X_T(\omega)) \big) = a_0(\omega),$$

and by the Leibniz rule,

$$\nabla_x \mathbb{E} \big[ \int_0^T f(X_t(\omega), t) \, dt + \int_0^T \langle h(X_t(\omega), t), \, dB_t \rangle + g(X_T(\omega)) \big]$$
$$= \mathbb{E} \big[ \nabla_x \big( \int_0^T f(X_t(\omega), t) \, dt + \int_0^T \langle h(X_t(\omega), t), \, dB_t \rangle + g(X_T(\omega)) \big) \big] = \mathbb{E} \big[ a_0(\omega) \big],$$

and

$$\nabla_x \mathbb{E} \big[ \exp \big( - \int_0^T f(X_t(\omega), t) \, dt - \int_0^T \langle h(X_t(\omega), t), \, dB_t \rangle - g(X_T(\omega)) \big) \big]$$
$$= -\mathbb{E} \big[ \nabla_x \big( \int_0^T f(X_t(\omega), t) \, dt + \int_0^T \langle h(X_t(\omega), t), \, dB_t \rangle + g(X_T(\omega)) \big)$$
$$\times \exp \big( - \int_0^T f(X_t(\omega), t) \, dt - \int_0^T \langle h(X_t(\omega), t), \, dB_t \rangle - g(X_T(\omega)) \big) \big]$$
$$= -\mathbb{E} \big[ a_0(\omega) \exp \big( - \int_0^T f(X_t(\omega), t) \, dt - \int_0^T \langle h(X_t(\omega), t), \, dB_t \rangle - g(X_T(\omega)) \big) \big].$$

$\square$

**Lemma 7** (Stochastic integration by parts, [60]). *Let*

$$dX_t = a_t \, dt + b_t \, dW_t^1,$$
$$dY_t = f_t \, dt + g_t \, dW_t^2.$$

*where $a_t$, $b_t$, $f_t$, $g_t$ are continuous square integrable processes adapted to a filtration $(\mathcal{F}_t)_{t \in [0, T]}$, and $W^1$, $W^2$ are Brownian motions adapted to the same filtration. Then,*

$$X_t Y_t - X_0 Y_0 = \int_0^t X_s \, dY_s + \int_0^t Y_s \, dX_s + \int_0^t \langle dX_s, dY_s \rangle$$
$$= \int_0^t X_s \, dY_s + \int_0^t Y_s \, dX_s + \int_0^t \langle b_s \, dW_s^1, g_s \, dW_s^2 \rangle.$$

**Remark 3** (Related work to the path-wise reparameterization trick: sensitivity analysis). *As shown above, the adjoint method for SDEs is an alternative to the path-wise reparameterization trick. Prior to [54], an array of works developed methods to compute derivatives of functionals of stochastic processes with respect to generic parameters $\alpha$ that appear either in the drift or diffusion coefficients [52]. This area is known as sensitivity analysis, and has been developed largely with financial applications in mind (more specifically, to compute the "Greeks"). In low dimensions, dynamic programming [3] or finite differences [26, 53] work well, but they scale poorly to high dimensions. In high dimensions, several approaches have been proposed (see the section 1 of [29] for a comprehensive although dated overview):*

- *The* path-wise method *(which we refer to as adjoint method) involves taking the gradient $\nabla_\alpha \mathbb{E}[f(X_t)]$ inside of the expectation as $\mathbb{E}[\nabla_\alpha f(X_t)]$ and was first described by [82].*

- *The* likelihood method *or* score method *[27, 69] consists in rewriting $\nabla_\alpha \mathbb{E}[f(X_T)]$ as $\mathbb{E}[f(X_T) H]$, where $H$ is a random variable which is equal to $\nabla_\alpha \log p(\alpha, X_T)$, $p(\alpha, \cdot)$ being the density of the law of $X_T$ with respect to the Lebesgue measure. [82] provide explicit weights $H$, under the restrictions that $\alpha$ appears only in the drift of the SDE (and not in the diffusion coefficient) and that the diffusion coefficient is elliptic, using the Girsanov theorem. [29] provide an expression for $H$ in the case where $H$ also appears in the diffusion coefficient, using Malliavin calculus.*

*The estimator of the path-wise reparameterization trick is formally similar to the likelihood method estimator, but it is different in that $\alpha$ is the initial condition of the process, and does not appear either in the drift nor the diffusion coefficient.*

## C.5 Proof of Lemma 1

*Proof.* Since the equality (28) holds almost surely for the pair $(X, B)$, it must also hold almost surely for $(X^v, B^v)$, which satisfy the same SDE. That is

$$\mathcal{W}(X^v, 0) = V(X_0^v, 0) + \tfrac{1}{2} \int_0^T \|u^*(X_s^v, s)\|^2 \, \mathrm{d}s - \sqrt{\lambda} \int_0^T \langle u^*(X_s^v, s), \mathrm{d}B_s^v \rangle,$$

Thus, we obtain that

$$\alpha(v, X^v, B) = \exp\big( -\lambda^{-1} \mathcal{W}(X^v, 0) - \lambda^{-1/2} \int_0^T \langle v(X_t^v, t), \mathrm{d}B_t^v \rangle + \tfrac{\lambda^{-1}}{2} \int_0^T \|v(X_t^v, t)\|^2 \, \mathrm{d}t \big)$$

$$= \exp\big( -\lambda^{-1} V(X_0^v, 0) - \tfrac{\lambda^{-1}}{2} \int_0^T \|u^*(X_s^v, s)\|^2 \, \mathrm{d}s + \lambda^{-1/2} \int_0^T \langle u^*(X_s^v, s), \mathrm{d}B_s^v \rangle$$

$$- \lambda^{-1/2} \int_0^T \langle v(X_t^v, t), \mathrm{d}B_t^v \rangle + \tfrac{\lambda^{-1}}{2} \int_0^T \|v(X_t^v, t)\|^2 \, \mathrm{d}t \big),$$

and this is equal to $\exp\big( -V(X_0^v, 0) \big)$ when $v = u^*$. Since we condition on $X_0^v = x_{\mathrm{init}}$, we have obtained that the random variable takes constant value $\exp\big( -V(x_{\mathrm{init}}, 0) \big)$ almost surely, which means that its variance is zero. $\qquad\square$

# D   Optimal reparameterization matrices

**Theorem 4** (Optimal reparameterization matrices). *Let $v$ be an arbitrary control in $\mathcal{U}$. Define the integral operator $\mathcal{T}_t : L^2([t, T]; \mathbb{R}^{d \times d}) \to L^2([t, T]; \mathbb{R}^{d \times d})$ as*

$$[\mathcal{T}_t(\dot{M}_t)](s) = \int_t^T \dot{M}_t(s') \mathbb{E}\big[ \chi(s', X^v, B) \chi(s, X^v, B)^\top \times \alpha(v, X^v, B) \big] \, \mathrm{d}s',$$

*where*

$$\chi(t, X^v, B) := \int_t^T \nabla_x f(X_s^v, s) \, \mathrm{d}s + \nabla g(X_T^v) + (\sigma_s^{-1})^\top(t) v(X_t^v, t)$$

$$- \int_t^T \nabla_x b(X_s^v, s)(\sigma_s^{-1})^\top(s) v(X_t^v, t) \, \mathrm{d}s - \int_t^T \nabla_x b(X_s^v, s)(\sigma_s^{-1})^\top(s) \, \mathrm{d}B_s.$$

*If we define $N_t(s) = -\mathbb{E}\big[ \big(\nabla g(X_T^v) + \int_t^T \nabla_x f(X_{s'}^v, s') \, \mathrm{d}s'\big) \chi(t, X^v, B)^\top \times \alpha(v, X^v, B) \big]$, the optimal $M^* = (M_t^*)_{t \in [0, T]}$ is of the form $M_t^*(s) = I + \int_t^s \dot{M}_t^*(s') \, \mathrm{d}s'$, where $\dot{M}_t^*$ is the unique solution of the following Fredholm equation of the first kind:*

$$\mathcal{T}_t(\dot{M}_t) = N_t.$$

The proof of (25) shows that minimizing $\mathrm{Var}(w; M)$ is equivalent to minimizing

$$\mathbb{E}\big[ \tfrac{1}{T} \int_0^T \big\| w(t, v, X^v, B, M_t) \big\|^2 \, \mathrm{d}t \, \alpha(v, X^v, B) \big]. \tag{57}$$

To optimize with respect to $M$, it is convenient to reexpress it in terms of $\dot{M} = (\dot{M}_t)_{t \in [0, T]}$ as $M_t(s) = I + \int_t^s \dot{M}_t(s') \, \mathrm{d}s'$. By Fubini's theorem, we have that

$$\int_t^T M_t(s) \nabla_x f(X_s^v, s) \, \mathrm{d}s = \int_t^T \big( I + \int_t^s \dot{M}_t(s') \, \mathrm{d}s' \big) \nabla_x f(X_s^v, s) \, \mathrm{d}s$$

$$= \int_t^T \nabla_x f(X_s^v, s) \, \mathrm{d}s + \int_t^T \dot{M}_t(s) \int_s^T \nabla_x f(X_{s'}^v, s') \, \mathrm{d}s' \, \mathrm{d}s,$$

$$- \int_t^T (M_t(s) \nabla_x b(X_s^v, s) - \dot{M}_t(s))(\sigma^{-1})^\top(s) v(X_s^v, s) \, \mathrm{d}s$$

$$= \int_t^T \dot{M}_t(s)(\sigma^{-1})^\top(s) v(X_s^v, s) \, \mathrm{d}s - \int_t^T \dot{M}_t(s) \int_s^T \nabla_x b(X_{s'}^v, s')(\sigma_{s'}^{-1})^\top(s') v(X_s^v, s) \, \mathrm{d}s' \, \mathrm{d}s,$$

$$- \lambda^{1/2} \int_t^T (M_t(s) \nabla_x b(X_s^v, s) - \dot{M}_t(s))(\sigma^{-1})^\top(s) \, \mathrm{d}B_s$$

$$= \lambda^{1/2} \big( \int_t^T \dot{M}_t(s)(\sigma^{-1})^\top(s) v(X_s^v, s) \, \mathrm{d}s - \int_t^T \dot{M}_t(s) \int_s^T \nabla_x b(X_{s'}^v, s')(\sigma_{s'}^{-1})^\top(s') \, \mathrm{d}B_{s'} \, \mathrm{d}s \big).$$

Hence, we can rewrite (57) as

$$\mathcal{G}(\dot{M}) = \mathbb{E}\big[ \tfrac{1}{T} \int_0^T \big\| \sigma(t)^\top \big( \int_t^T \nabla_x f(X_s^v, s) \, \mathrm{d}s + \nabla g(X_T^v)$$

$$+ \int_t^T \dot{M}_t(s) \big( \int_s^T \nabla_x f(X_{s'}^v, s') \, \mathrm{d}s' + \nabla g(X_T^v) + (\sigma^{-1})^\top(s) v(X_s^v, s)$$

$$- \int_s^T \nabla_x b(X_{s'}^v, s')(\sigma_{s'}^{-1})^\top(s') v(X_s^v, s) \, \mathrm{d}s' - \int_s^T \nabla_x b(X_{s'}^v, s')(\sigma_{s'}^{-1})^\top(s') \, \mathrm{d}B_{s'} \big) \, \mathrm{d}s \big) \big\|^2 \, \mathrm{d}t$$

$$\times \alpha(v, X^v, B) \big]$$

The first variation $\frac{\delta \mathcal{G}}{\delta M}(\dot{M})$ of $\mathcal{G}$ at $\dot{M}$ is defined as the family $Q = (Q_t)_{t \in [0,T]}$ of matrix-valued functions such that for any collection of matrix-valued functions $P = (P_t)_{t \in [0,T]}$,

$$\partial_\epsilon \mathcal{V}(\dot{M} + \epsilon P)|_{\epsilon=0} = \lim_{\epsilon \to 0} \frac{\mathcal{V}(\dot{M}+\epsilon P) - \mathcal{V}(M)}{\epsilon} = \langle P, Q \rangle := \int_0^T \int_t^T \langle P_t(s), Q_t(s) \rangle_F \, ds \, dt,$$

where $\dot{M} + \epsilon P := (\dot{M}_t + \epsilon P_t)_{t \in [0,T]}$. Now, note that

$$
\begin{aligned}
\partial_\epsilon \mathcal{V}(\dot{M} + \epsilon P)|_{\epsilon=0} &= \partial_\epsilon \mathbb{E}\Big[\frac{1}{T} \int_0^T \Big\|\sigma(t)^\top \Big( \int_t^T \nabla_x f(X_s^v, s) \, ds + \nabla g(X_T^v) \\
&\quad + \int_t^T (\dot{M}_t(s) + \epsilon P_t(s)) \Big( \int_s^T \nabla_x f(X_{s'}^v, s') \, ds' + \nabla g(X_T^v) + (\sigma^{-1})^\top(s) v(X_s^v, s) \\
&\quad - \int_s^T \nabla_x b(X_{s'}^v, s')(\sigma_{s'}^{-1})^\top(s') v(X_s^v, s) \, ds' - \int_s^T \nabla_x b(X_{s'}^v, s')(\sigma_{s'}^{-1})^\top(s') \, dB_{s'} \Big) \, ds \Big) \Big\|^2 \, dt \\
&\quad \times \alpha(v, X^v, B) \Big]\Big|_{\epsilon=0} \\
&= \mathbb{E}\Big[\frac{2}{T} \int_0^T \Big\langle \sigma(t)\sigma(t)^\top \Big( \int_t^T \nabla_x f(X_s^v, s) \, ds + \nabla g(X_T^v) \\
&\quad + \int_t^T \dot{M}_t(s) \Big( \int_s^T \nabla_x f(X_{s'}^v, s') \, ds' + \nabla g(X_T^v) + (\sigma^{-1})^\top(s) v(X_s^v, s) \\
&\quad - \int_s^T \nabla_x b(X_{s'}^v, s')(\sigma_{s'}^{-1})^\top(s') v(X_s^v, s) \, ds' - \int_s^T \nabla_x b(X_{s'}^v, s')(\sigma_{s'}^{-1})^\top(s') \, dB_{s'} \Big) \, ds \Big), \\
&\quad \int_t^T P_t(s) \Big( \int_s^T \nabla_x f(X_{s'}^v, s') \, ds' + \nabla g(X_T^v) + (\sigma^{-1})^\top(s) v(X_s^v, s) \\
&\quad - \int_s^T \nabla_x b(X_{s'}^v, s')(\sigma_{s'}^{-1})^\top(s') v(X_s^v, s) \, ds' - \int_s^T \nabla_x b(X_{s'}^v, s')(\sigma_{s'}^{-1})^\top(s') \, dB_{s'} \Big) \, ds \Big\rangle \, dt \\
&\quad \times \alpha(v, X^v, B) \Big].
\end{aligned}
$$
(58)

If we define
$$
\begin{aligned}
\chi(s, X^v, B) &:= \int_s^T \nabla_x f(X_{s'}^v, s') \, ds' + \nabla g(X_T^v) + (\sigma^{-1})^\top(s) v(X_s^v, s) \\
&\quad - \int_s^T \nabla_x b(X_{s'}^v, s')(\sigma_{s'}^{-1})^\top(s') v(X_s^v, s) \, ds' - \int_s^T \nabla_x b(X_{s'}^v, s')(\sigma_{s'}^{-1})^\top(s') \, dB_{s'},
\end{aligned}
$$

we can rewrite (58) as
$$\partial_\epsilon \mathcal{V}(\dot{M}+\epsilon P)|_{\epsilon=0} = \mathbb{E}\Big[\frac{1}{T} \int_0^T \Big\langle \sigma(t)\sigma(t)^\top \Big( \int_t^T \nabla_x f(X_s^v, s) \, ds + \nabla g(X_T^v) + \int_t^T \dot{M}_t(s) \chi(s, X^v, B) \, ds \Big),$$
(59)
$$\int_t^T P_t(s) \chi(s, X^v, B) \, ds \Big\rangle \, ds \times \alpha(v, X^v, B) \Big]$$

Now let us reexpress equation (59) as:
$$
\begin{aligned}
&\mathbb{E}\Big[\frac{1}{T} \int_0^T \Big\langle \sigma\sigma^\top(t) \Big( \nabla g(X_T^v) + \int_t^T \big( \nabla_x f(X_s^v, s) + \dot{M}_t(s) \chi(s, X^v, B) \big) \, ds \Big), \\
&\qquad \int_t^T P_t(s) \chi(s, X^v, B) \, ds \Big\rangle \, dt \times \alpha(v, X^v, B) \Big] \\
&\overset{(i)}{=} \mathbb{E}\Big[\frac{1}{T} \int_0^T \int_0^s \Big\langle P_t(s) \chi(s, X^v, B), \\
&\qquad \sigma\sigma^\top(t) \big( \nabla g(X_T^v) + \int_t^T (\nabla_x f(X_{s'}^v, s') + \dot{M}_t(s')\chi(s', X^v, B)) \, ds' \big) \Big\rangle \, dt \, ds \times \alpha(v, X^v, B) \Big] \\
&\overset{(ii)}{=} \mathbb{E}\Big[\frac{1}{T} \int_0^T \int_0^s \Big\langle \sigma\sigma^\top(t) \big( \nabla g(X_T^v) + \int_t^T (\nabla_x f(X_{s'}^v, s') + \dot{M}_t(s')\chi(s', X^v, B)) \, ds' \big) \chi(X^v, s, B)^\top, \\
&\qquad P_t(s) \Big\rangle_F \, dt \, ds \times \alpha(v, X^v, B) \Big] \\
&= \int_0^T \int_0^s \Big\langle \frac{1}{T} \sigma\sigma^\top(t) \mathbb{E}\big[ \big( \nabla g(X_T^v) + \int_t^T (\nabla_x f(X_{s'}^v, s') + \dot{M}_t(s')\chi(X^v, s', B)) \, ds' \big) \chi(X^v, s, B)^\top \alpha(v, X^v, B) \big], \\
&\qquad P_t(s) \Big\rangle_F \, dt \, ds.
\end{aligned}
$$
(60)

Here, equality (i) holds by Lemma 8 with the choices $\alpha(t, s) = P_t(s)\chi(X^v, s, B)$, $\gamma(t) = \sigma\sigma^\top(t)\big(\nabla g(X_T^v) + \int_t^T \big(\nabla_x f(X_s^v, s) + \dot{M}_t(s)\chi(X^v, s, B)\big) \, ds$. Equality (ii) follows from the fact that for any matrix $A$ and vectors $b, c$, $\langle Ab, c \rangle = c^\top Ab = \text{Tr}(c^\top Ab) = \text{Tr}(Abc^\top) = \langle B, cb^\top \rangle_F$, where $\langle \cdot, \cdot \rangle_F$ denotes the Frobenius inner product.

The first-order necessary condition for optimality states that at the optimal $\dot{M}^*$, the first variation $\frac{\delta \mathcal{G}}{\delta M}(\dot{M}^*)$ is zero. In other words, $\partial_\epsilon \mathcal{V}(\dot{M} + \epsilon P)|_{\epsilon=0}$ is zero for any $P$. Hence, the right-hand side

of (60) must be zero for any $P$, which implies that almost everywhere with respect to $t \in [0, T]$, $s \in [s, T]$,

$$\mathbb{E}\big[\big(\nabla g(X_T^v) + \int_t^T (\nabla_x f(X_{s'}^v, s') + \dot{M}_t(s')\chi(X^v, s', B))\, \mathrm{d}s'\big)\chi(X^v, s, B)^\top \alpha(v, X^v, B)\big] = 0.$$

To derive this, we also used that $\sigma(t)$ is invertible by assumption.

Define the integral operator $\mathcal{T}_t : L^2([t, T]; \mathbb{R}^{d \times d}) \to L^2([t, T]; \mathbb{R}^{d \times d})$ as

$$[\mathcal{T}_t(\dot{M}_t)](s) = \int_t^T \dot{M}_t(s') \mathbb{E}\big[\chi(X^v, s', B)\chi(X^v, s, B)^\top \times \alpha(v, X^v, B)\big]\, \mathrm{d}s'$$

If we define $N_t(s) = -\mathbb{E}\big[\big(\nabla g(X_T^v) + \int_t^T \nabla_x f(X_{s'}^v, s')\, \mathrm{d}s'\big)\chi(X^v, s, B)^\top \times \alpha(v, X^v, B)\big]$, the problem that we need to solve to find the optimal $\dot{M}_t$ is

$$\mathcal{T}_t(\dot{M}_t) = N_t.$$

This is a Fredholm equation of the first kind.

**Lemma 8.** *If $\alpha, \beta : [0, T] \times [0, T] \to \mathbb{R}^d$, $\gamma : [0, T] \to \mathbb{R}^d$, $\delta : [0, T] \to \mathbb{R}^{d \times d}$ are arbitrary integrable functions, we have that*

$$\int_0^T \big\langle \int_t^T \alpha(t, s)\, \mathrm{d}s, \gamma(t) \big\rangle\, \mathrm{d}t = \int_0^T \int_0^s \big\langle \alpha(t, s), \gamma(t) \big\rangle\, \mathrm{d}t\, \mathrm{d}s,$$

*Proof.* We have that:

$$\int_0^T \int_t^T \big\langle \alpha(t, s), \gamma(t) \big\rangle\, \mathrm{d}s\, \mathrm{d}t \overset{(i)}{=} \int_0^T \int_0^{T-t} \big\langle \alpha(t, T - s), \gamma(t) \big\rangle\, \mathrm{d}s\, \mathrm{d}t$$
$$\overset{(ii)}{=} \int_0^T \int_0^t \big\langle \alpha(T - t, T - s), \gamma(T - t) \big\rangle\, \mathrm{d}s\, \mathrm{d}t \overset{(iii)}{=} \int_0^T \int_s^T \big\langle \alpha(T - t, T - s), \gamma(T - t) \big\rangle\, \mathrm{d}t\, \mathrm{d}s$$
$$\overset{(iv)}{=} \int_0^T \int_{T-s}^T \big\langle \alpha(T - t, s), \gamma(T - t) \big\rangle\, \mathrm{d}t\, \mathrm{d}s \overset{(v)}{=} \int_0^T \int_0^s \big\langle \alpha(t, s), \gamma(t) \big\rangle\, \mathrm{d}t\, \mathrm{d}s$$

Here, in equalities (i), (ii), (iv) and (v) we make changes of variables of the form $t \mapsto T - t$, $s \mapsto T - s$, $s' \mapsto T - s'$. In equality (iii) we use Fubini's theorem. $\qquad\square$

# E   Control warm-starting

We introduce the *Gaussian warm-start*, a control warm-start strategy that we adapt from [56], and that we use in our experiments in Figure 7. Their work tackles generalized Schrödinger bridge problems, which are different from the control setting in that the final distribution is known and there is no terminal cost. The following proposition, that provides an analytic expression of the control needed for the density of the process to be Gaussian at all times, is the foundation of our method.

**Proposition 6.** *Given $Z \sim N(0, I)$ define the random process $Y$ as*

$$Y_t = \mu(t) + \tilde{\Gamma}(t)Z, \qquad \text{where } \mu(t) \in \mathbb{R}^d, \ \tilde{\Gamma}(t) = \sqrt{t}\Gamma(t) \in \mathbb{R}^{d \times d}. \tag{61}$$

*Define the control $u : \mathbb{R}^d \times [0, T] \to \mathbb{R}^d$ as*

$$u(x, t) = \sigma(t)^{-1}\big(\partial_t \mu(t) + \big((\partial_t \Gamma(t))\Gamma(t)^{-1} + \tfrac{I - (\sigma\sigma^\top)(t)(\Sigma\Sigma^\top)^{-1}(t)}{2t}\big)(x - \mu(t)) - b(x, t)\big) \tag{62}$$

*Then, if $\Gamma_0 = \sigma(0)$, the controlled process $X^u$ defined in equation (2) has the same marginals as $Y$. That is, for all $t \in [0, T]$, $\mathrm{Law}(Y_t) = \mathrm{Law}(X_t^u)$.*

*Proof.* Following [56], we have that

$$\partial_t X_t = \partial_t \mu_t + \partial_t \tilde{\Gamma}(t)Z = \partial_t \mu(t) + (\partial_t \tilde{\Gamma}(t))\tilde{\Gamma}(t)^{-1}(X_t - \mu(t)),$$
$$\nabla \log p_t(x) = -\tilde{\Sigma}(t)^{-1}(x - \mu(t)), \qquad \tilde{\Sigma}(t) = \tilde{\Gamma}(t)\tilde{\Gamma}(t)^\top.$$

Now, $p_t$ satisfies the continuity equation equation

$$\partial_t p_t = -\nabla \cdot ((\partial_t \mu(t) + (\partial_t \tilde{\Gamma}(t))\tilde{\Gamma}(t)^{-1}(x - \mu(t)))p_t) \tag{63}$$

Let $D(t) = \frac{1}{2}\sigma(t)\sigma(t)^\top$. We want to reexpress (63) as a Fokker-Planck equation of the form

$$\partial_t p_t = -\nabla \cdot (v(x,t)p_t) + \sum_{i=1}^{d}\sum_{j=1}^{d}\partial_i\partial_j(D_{ij}(t)p_t) = -\nabla \cdot (v(x,t)p_t) + \sum_{i=1}^{d}\partial_i\sum_{j=1}^{d}(D_{ij}(t)\partial_j p_t)$$

$$= -\nabla \cdot (v(x,t)p_t) + \nabla \cdot (D(t)\nabla p_t) = -\nabla \cdot (v(x,t)p_t) + \nabla \cdot (D(t)\nabla\log p_t(x)p_t)$$
$$= -\nabla \cdot ((v(x,t) - D(t)\nabla\log p_t(x))p_t).$$

Hence, we need that

$$v(x,t) - D(t)\nabla\log p_t = \partial_t\mu(t) + (\partial_t\tilde{\Gamma}(t))\tilde{\Gamma}(t)^{-1}(x - \mu(t)),$$
$$\implies v_t(x) = \partial_t\mu(t) + ((\partial_t\tilde{\Gamma}(t))\tilde{\Gamma}(t)^{-1}(x - \mu(t)) + \frac{(\sigma\sigma^\top)(t)}{2}\nabla\log p_t(x)$$
$$= \partial_t\mu(t) + (\partial_t\tilde{\Gamma}(t))\tilde{\Gamma}(t)^{-1}(x - \mu(t)) - \frac{(\sigma\sigma^\top)(t)}{2}\Sigma(t)^{-1}(x - \mu(t)).$$

If we let $\tilde{\Gamma}(t) = \Gamma(t)\sqrt{t}$, then $\tilde{\Sigma}(t) = t\Gamma(t)\Gamma(t)^\top = t\Sigma(t)$ and $\partial_t\tilde{\Gamma}(t) = \partial_t\Gamma(t)\sqrt{t} + \frac{\Gamma(t)}{2\sqrt{t}}$. That is,

$$v(x,t) = \partial_t\mu(t) + \left(\partial_t\Gamma(t)\sqrt{t} + \frac{\Gamma(t)}{2\sqrt{t}}\right)\frac{\Gamma(t)^{-1}}{\sqrt{t}}(x - \mu(t)) - \frac{(\sigma\sigma^\top)(t)}{2}\frac{\Sigma(t)^{-1}}{t}(x - \mu(t))$$

$$= \partial_t\mu(t) + (\partial_t\Gamma(t))\Gamma(t)^{-1}(x - \mu(t)) + \frac{1}{2t}(x - \mu(t)) - \frac{(\sigma\sigma^\top)(t)\Sigma(t)^{-1}}{2t}(x - \mu(t))$$

For $v$ to be finite at $t = 0$, we need that $(\sigma\sigma^\top)(0)\Sigma(0)^{-1} = I$, which holds, for example, if $\Gamma(0) = \sigma(0)$. Also, to match the form of (2), we need that

$$v(x,t) = b(x,t) + \sigma(t)u(x,t),$$
$$\implies u(x,t) = \sigma(t)^{-1}\left(\partial_t\mu_t + \left((\partial_t\Gamma(t))\Gamma(t)^{-1} + \frac{I - (\sigma\sigma^\top)(t)\Sigma(t)^{-1}}{2t}\right)(x - \mu_t) - b(x,t)\right).$$
$\qquad\qquad\qquad\qquad\qquad\qquad\qquad\qquad\qquad\qquad\qquad\qquad\qquad\qquad\qquad\qquad\qquad\qquad \square$

The warm-start control is computed as the solution of a *Restricted Gaussian Stochastic Optimal Control* problem, where we constrain the space of controls to those that induce Gaussian paths as described in Prop. 6. In practice, we learn a linear spline $\mu = (\mu^{(b)})_{b=0}^{\mathcal{B}}$, where $\mu^{(b)} \in \mathbb{R}^d$, and a linear spline $\Gamma = (\Gamma^{(b)})_{b=0}^{\mathcal{B}}$, where $\Gamma^{(b)} \in \mathbb{R}^{d \times d}$. These linear splines take the role of $\mu(t)$ and $\Sigma(t)$ in (61). Given splines $\mu$ and $\Gamma$, we obtain the warm-start control using (62); for a given $t \in [0,T)$, if we let $b_- = \lfloor \mathcal{B}t/T \rfloor$, $b_+ = b_- + 1$, $\Delta = T/\mathcal{B}$, we have that

$$\widehat{\mu}(t) = \frac{(t - b_-\Delta)\mu^{(b_+)} + (b_+\Delta - t)\mu^{(b_-)}}{\Delta}, \qquad \widehat{\partial_t\mu}(t) = \frac{\mu^{(b_+)} - \mu^{(b_-)}}{\Delta}, \qquad (64)$$

$$\widehat{\Gamma}(t) = \frac{(t - b_-\Delta)\Gamma^{(b_+)} + (b_+\Delta - t)\Gamma^{(b_-)}}{\Delta}, \qquad \widehat{\partial_t\Gamma}(t) = \frac{\Gamma^{(b_+)} - \Gamma^{(b_-)}}{\Delta}, \qquad (65)$$

$$\hat{u}(x,t) = \sigma(t)^{-1}\left(\widehat{\partial_t\mu}(t) + \left(\widehat{\partial_t\Gamma}(t)\widehat{\Gamma}(t)^{-1} + \frac{I - (\sigma\sigma^\top)(t)(\widehat{\Sigma}\widehat{\Sigma}^\top)^{-1}(t)}{2t}\right)(x - \widehat{\mu}(t)) - b(x,t)\right). (66)$$

Algorithm 3 provides a method to learn the splines $\mu$, $\Gamma$. It is a stochastic optimization algorithms in which the spline parameters are updated by sampling $Y_t$ in (61) at different times, computing the control cost relying on (66), and taking its gradient.

---

**Algorithm 3** Restricted Gaussian Stochastic Optimal Control

---

**Input:** State cost $f(x,t)$, terminal cost $g(x)$, diffusion coeff. $\sigma(t)$, base drift $b(x,t)$, noise level $\lambda$, number of iterations $N$, batch size $m$, number of time steps $K$, number of spline knots $\mathcal{B}$, initial mean spline knots $\mu_0 = (\mu_0^{(b)})_{b=0}^{\mathcal{B}}$, initial noise spline knots $\Gamma_0 = (\Gamma_0^{(b)})_{b=0}^{\mathcal{B}}$.

1 **for** $n = 0 : (N - 1)\}$ **do**
2 $\quad$ Sample $m$ i.i.d. variables $(Z_i)_{i=1}^{n} \sim N(0,I)$ and $m$ times $(t_i)_{i=1}^{n} \sim \text{Unif}([0,T])$.
3 $\quad$ **for** $j = 0 : K$ **do**
4 $\quad\quad$ Set $t_j = jT/K$, and compute $\widehat{\mu}_n(t_j)$, $\widehat{\partial_t\mu}_n(t_j)$, $\widehat{\Gamma}_n(t_j)$, $\widehat{\partial_t\Gamma}_n(t_j)$ according to (64), (65) using $\mu_n$, $\Gamma_n$
5 $\quad\quad$ **for** $i = 1 : m$ **do** compute $Y_{ij} = \hat{\mu}(t_j) + \sqrt{t_j}\widehat{\Gamma}(t_j)Z_i$ and $\hat{u}_n(Y_{ij}, t_j)$ using (66);
6 $\quad$ **end**
7 $\quad$ Compute $\hat{\mathcal{L}}_{\text{RGSOC}}(\mu_n, \Gamma_n) = \frac{1}{m}\sum_{i=1}^{m}\left(\frac{T}{K}\sum_{j=0}^{K-1}\left(\frac{1}{2}\|\hat{u}(Y_{ij}, t_j)\|^2 + f(Y_{ij}, t_j)\right) + g(Y_{iK})\right)$
8 $\quad$ Compute the gradient of $\hat{\mathcal{L}}_{\text{RGSOC}}(\mu_n, \Gamma_n)$ with respect to the spline parameters $(\mu_n, \Gamma_n)$.
9 $\quad$ Obtain $\mu_{n+1}$, $\Gamma_{n+1}$ with via an Adam update on $\mu_n$, $\Gamma_n$ resp. (or another stochastic algorithm)
10 **end**
**Output:** Learned splines $\mu_N$, $\Gamma_N$, control $\hat{u}_N$

---

Once we have access to the restricted control $\hat{u}_N$, we can warm-start the control in Algorithms 1 and 2 by introducing $\hat{u}_N$ as an offset. That is, we parameterize the control as $u_\theta = \hat{u}_N + \tilde{u}_\theta$.

# F  Experimental details and additional plots

## F.1  Experimental details

The control $L^2$ error curves show the following quantity:

$$\mathbb{E}_{t,\mathbb{P}^{u^*}}[\|u^*(X_t^{u^*},t) - u(X_t^{u^*},t)\|^2 e^{-\lambda^{-1}V(X_0^{u^*},0)}]/\mathbb{E}_{t,\mathbb{P}^{u^*}}[e^{-\lambda^{-1}V(X_0^{u^*},0)}]$$

That is, we sample trajectories using the optimal control, and compute the error using a Monte Carlo estimate. In all our experiments, the distribution $X_0^{u^*}$ is a delta, which means that we do not need to compute $V(X_0^{u^*},0)$. We keep an exponential moving average (EMA) estimate of the control $L^2$ error, which we show in the plots. To compute it, we sample ten batches of optimally controlled trajectories every 10 training iterations, and we update the quantity with the average of the ten batches, using EMA coefficient 0.02. All other quantities shown in the plots are also smoothed out using EMA with coefficient 0.01, except for control objective values, which are computed as the average of 65536 samples, every 5000 training steps.

For all losses and all settings, we train the control using Adam with learning rate $1 \times 10^{-4}$. For SOCM, we train the reparametrization matrices using Adam with learning rate $1 \times 10^{-2}$. We use batch size $m = 128$ unless otherwise specified. When used, we run the warm-start algorithm (Algorithm 3) with $\mathcal{B} = 20$ knots, $K = 200$ time steps, and batch size $m = 512$, and we use Adam with learning rate $3 \times 10^{-4}$ for $N = 60000$ iterations.

**QUADRATIC ORNSTEIN-UHLENBECK**  The choices for the functions of the control problem are:

$$b(x,t) = Ax, \quad f(x,t) = x^\top P x, \quad g(x) = x^\top Q x, \quad \sigma(t) = \sigma_0.$$

where $Q$ is a positive definite matrix. Control problems of this form are better known as linear quadratic regulator (LQR) and they admit a closed form solution [78, Thm. 6.5.1]. The optimal control is given by:

$$u_t^*(x) = -2\sigma_0^\top F_t x,$$

where $F_t$ is the solution of the Ricatti equation

$$\frac{dF_t}{dt} + A^\top F_t + F_t A - 2F_t \sigma_0 \sigma_0^\top F_t + P = 0$$

with the final condition $F_T = Q$. Within the QUADRATIC OU class, we consider two settings:

- Easy: We set $d = 20$, $A = 0.2I$, $P = 0.2I$, $Q = 0.1I$, $\sigma_0 = I$, $\lambda = 1$, $T = 1$, $x_{\text{init}} = 0.5N(0,I)$. We do not use warm-start for any algorithm. We take $K = 50$ time discretization steps, and we use random seed 0.
- Hard: We set $d = 20$, $A = I$, $P = I$, $Q = 0.5I$, $\sigma_0 = I$, $\lambda = 1$, $T = 1$, $x_{\text{init}} = 0.5N(0,I)$. We use the *Gaussian warm-start* (App. E). We take batch size $m = 64$ and $K = 150$ time discretization steps, and we use random seed 0.

**LINEAR ORNSTEIN-UHLENBECK**  The functions of the control problem are chosen as follows:

$$b(x,t) = Ax, \quad f(x,t) = 0, \quad g(x) = \langle \gamma, x \rangle, \quad \sigma(t) = \sigma_0.$$

The optimal control for this class of problems is given by [59, Sec. A.4]:

$$u_t^*(x) = -\sigma_0^\top e^{A^\top(T-t)}\gamma.$$

We use exactly the same functions as [59]: we sample $(\xi_{ij})_{1 \le i,j \le d}$ once at the beginning of the simulation, and set:

$$d = 10, \quad A = -I + (\xi_{ij})_{1 \le i,j \le d}, \quad \gamma = \mathbb{1}, \quad \sigma_0 = I + (\xi_{ij})_{1 \le i,j \le d},$$
$$T = 1, \quad \lambda = 1, \quad x_{\text{init}} = 0.5N(0,I).$$

We take $K = 100$ time discretization steps, and we use random seed 0.

**DOUBLE WELL** We also use exactly the same functions as [59], which are the following:

$$b(x,t) = -\nabla\Psi(x), \quad \Psi(x) = \sum_{i=1}^{d}\kappa_i(x_i^2 - 1)^2, \quad f(x) = 0, \quad g(x) = \sum_{i=1}^{d}\nu_i(x_i^2 - 1)^2, \quad \sigma_0 = I,$$

where $d = 10$, and $\kappa_i = 5$, $\nu_i = 3$ for $i \in \{1, 2, 3\}$ and $\kappa_i = 1$, $\nu_i = 1$ for $i \in \{4, \ldots, 10\}$. We set $T = 1$, $\lambda = 1$ and $x_{\text{init}} = 0$. We take $K = 200$ time discretization steps, and we use random seed 0. The Double Well problem is actually highly non-trivial, and is multimodal. The only reason we can produce a "ground truth" control to compare to in this setting is that we use significant knowledge of the problem; we analytically reduce it to 1D problems by decoupling each dimension and apply numerical methods to solve the Hamilton-Jacobi-Bellman equation for these 1D problems. It is not a problem where we actually have the ground truth control in closed form.

**PATH INTEGRAL SAMPLER ON MIXTURE OF GAUSSIANS** We set

$$b(x,t) = 0, \qquad f(x,t) = 0, \qquad g(x) = \log(\mu^0(x)/\mu(x)) = -\frac{\|x\|^2}{2} - \frac{d}{2}\log(2\pi) - \log\mu(x),$$

where $T = 1$, and $\mu$ is the density of a mixture of two Gaussians with means $\pm e_1$, where $e_1 = (1, 0, \ldots, 0)$, and variance Id. Note that we take $\mu$ to be normalized, i.e. $\int \mu(x)\,\mathrm{d}x = 1$, or equivalently, $\log Z = \log\left(\int \mu(x)\,\mathrm{d}x\right) = 0$. In Figure 1, we use the following Monte Carlo estimator of the control objective at the control $u$:

$$\hat{S}^u(X) = \int_0^T \left(\frac{1}{2}\|u(X_t^u, t)\|^2 + f(X_t^u, t)\right)\mathrm{d}t + g(X_T^u) + \int\langle u(X_t^u, t), \mathrm{d}B_t\rangle.$$

Note that this estimator is unbiased because $\mathbb{E}[\int\langle u(X_t^u, t), \mathrm{d}B_t\rangle] = 0$. This is known as the Sticking the Landing estimator, as it has zero variance when $u$ is the optimal control [72]. The fact that $\mathbb{E}[-\hat{S}^u(X)] \leq \log Z = 0$ with equality when $u = u^*$ is stated as [84, Thm. 4].

## F.2 Model architectures

As a general guideline, the control function can be thought of as the analog of the score function in diffusion models; hence, a natural choice for the architecture can be U-Nets or diffusion transformers if the control task is on images, audio or video. Other domains may require different architectures. In the experiments we report, we used the architecture implemented in the class `FullyConnectedUNet` within the file `SOC_matching/models.py`. It is a simplified version of the U-Net architecture where both the down-sampling and up-sampling layers are fully connected with ReLU activations, and the horizontal layers are linear transformations. We use three down-sampling and up-sampling steps, with widths 256, 128 and 64 (hence, the first down-sampling step is actually an up-sampling, because the data dimensions in our experiments range from 10 to 20).

The reparameterization matrices have an unusual trait, which is that their input dimension is small (two) while their output dimension is large ($d^2$). Hence, the kind of functions that they need to learn are low dimensional and hence easy. In our case, we used the architecture implemented in the class `SigmoidMLP` within the file `SOC_matching/models.py`, which is essentially a three layer multilayer perceptron with ReLU activations and output dimension $d^2$, whose output is averaged with the identity matrix using sigmoid weights, in order to enforce that $M_t(t)$ be the identity matrix.

## F.3 Additional tables and plots

Table 1 shows the average times per iteration for each algorithm. Each algorithm was run using a 16GB V100 GPU.

| SOCM | SOCM $M_t = I$ | SOCM adj. | Adj. | Cross entropy | Log-variance | Moment | Variance |
|------|------|------|------|------|------|------|------|
| 0.222 | 0.090 | 0.099 | 0.169 | 0.086 | 0.117 | 0.087 | 0.086 |

Table 1: Time per iteration (exponential moving average) for various algorithms in seconds per iteration, for the QUADRATIC OU (EASY) experiments (Figure 2).

Figure 4 shows the control objective (1) for the four settings. The error bars for the control objective plots show the confidence intervals for $\pm$ one standard deviation, computed via a Monte Carlo

estimate using 65536 trajectories per data point. They show the standard error of the mean. As expected, SOCM also obtains the lowest values for the control objective, up to the estimation error.

Figure 5 shows the normalized standard deviation of the importance weight for the learned control $u$: $\sqrt{\text{Var}[\alpha(u, X^u, B)]}/\mathbb{E}[\alpha(u, X^u, B)]$. By Lemma 1, when $X_0^u = x_{\text{init}}$ for an arbitrary $x_{\text{init}}$ (which is the case for all our experiments), this quantity is zero for the optimal control $u^*$. Hence, the normalized standard deviation of $\alpha$ is an alternative metric to measure the optimality of the learned control.

Figure 6 shows an exponential moving average of the norm squared of the gradient for LINEAR OU and DOUBLE WELL. For LINEAR OU, the minimum gradient norm is achieved by the adjoint method, while for DOUBLE WELL it is achieved by the cross entropy loss. The training instabilities of the adjoint method become apparent as well. Interestingly, in both settings the algorithms with smallest gradients are not SOCM, which is the algorithm with smallest error as shown in Figure 3. Understanding this phenomenon is outside of the scope of this paper.

Figure 7 shows plots of the control $L^2$ error, the norm squared of the gradient, and the control objective for the QUADRATIC OU (HARD) setting, using a warm-start strategy detailed in App. E. Figure 7 shows that SOCM is once again the algorithm that achieves the lowest error and the smallest gradients. Remark that the warm-start control is a reasonable approximation of the optimal control, as the initial control $L^2$ error is much lower than in the other figures.

Figure 8 shows the value of the training loss for SOCM and its two ablations: SOCM with constant $M_t = I$, and SOCM-Adjoint. For all such algorithms, the training loss is the sum of the $L^2$ error of the learned control $u$, and the expected conditional variance of the matching vector field $w$. Thus, the difference between the training loss plots and the $L^2$ error plots is the expected conditional variance of $w$. We observe that the expected conditional variance in the QUADRATIC OU setting is orders of magnitude smaller for SOCM than for its two ablations. For LINEAR OU, SOCM and SOCM-adjoint have similar expected conditional variance, and a possible explanation is that the LINEAR OU setting is very simple. In the DOUBLE WELL setting, the SOCM-adjoint training loss curve has spikes that are probably caused by instabilities of the adjoint method. These spikes can be attributed mostly to the expected conditional variance term, since the corresponding $L^2$ error curve in Figure 3 does not present them.

Figure 9 shows that the instabilities of the adjoint method are inherent to the loss, because they also appear at small learning rates: $3 \times 10^{-5}$ is smaller than the learning rates typically used for Adam, which hover from $1 \times 10^{-4}$ to $1 \times 10^{-3}$.

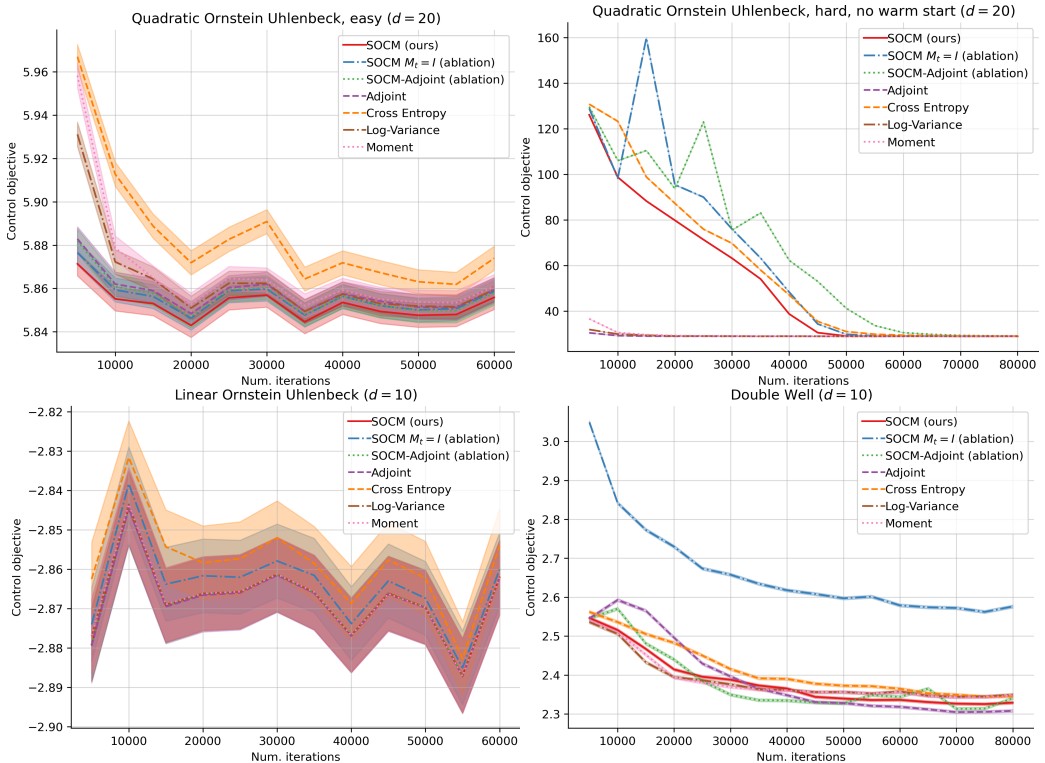

Figure 4: Plots of the control objective for the four settings.

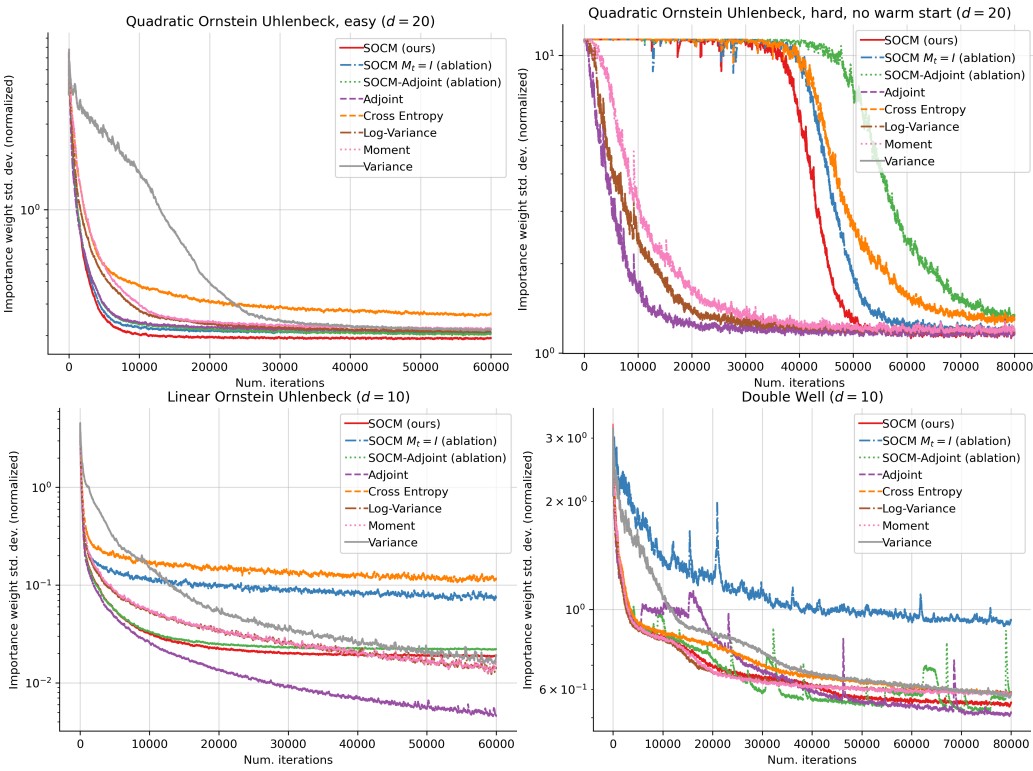

Figure 5: Plots of the normalized standard deviation of the importance weights: $\sqrt{\mathrm{Var}[\alpha(u, X^u, B)]}/\mathbb{E}[\alpha(u, X^u, B)]$.

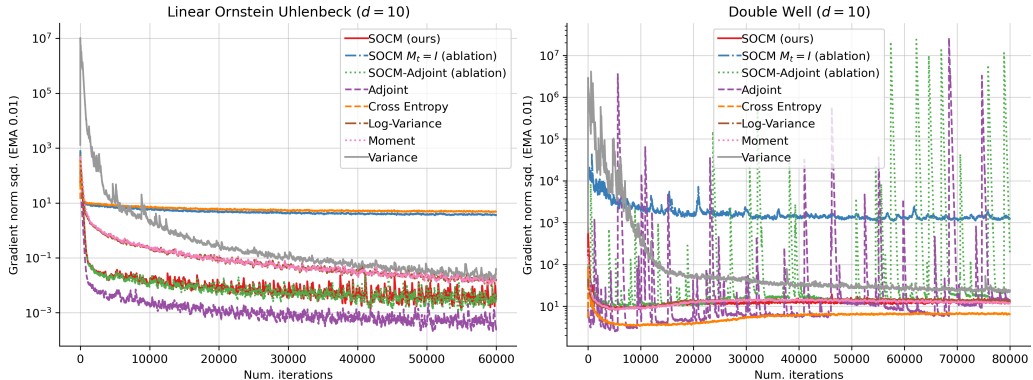

Figure 6: Plots of the norm squared of the gradient for the LINEAR ORNSTEIN UHLENBECK and DOUBLE WELL settings.

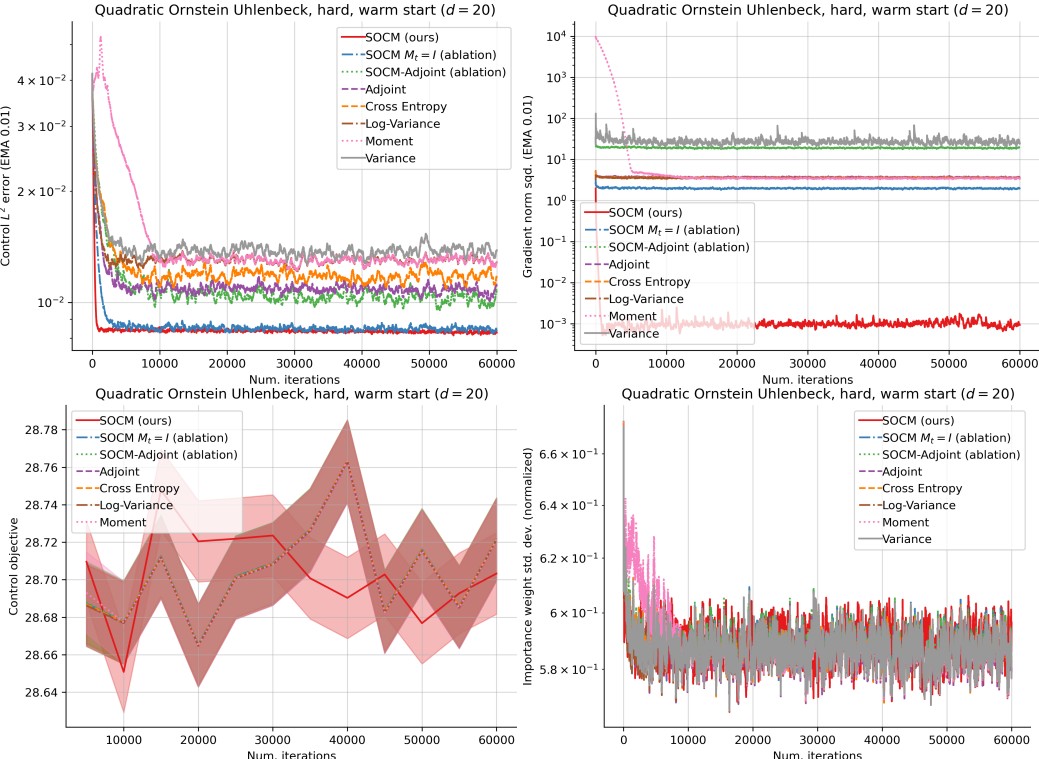

Figure 7: Plots of the control $L^2$ error, the norm squared of the gradient, and the control objective for the QUADRATIC ORNSTEIN-UHLENBECK (HARD) setting, without using warm-start.

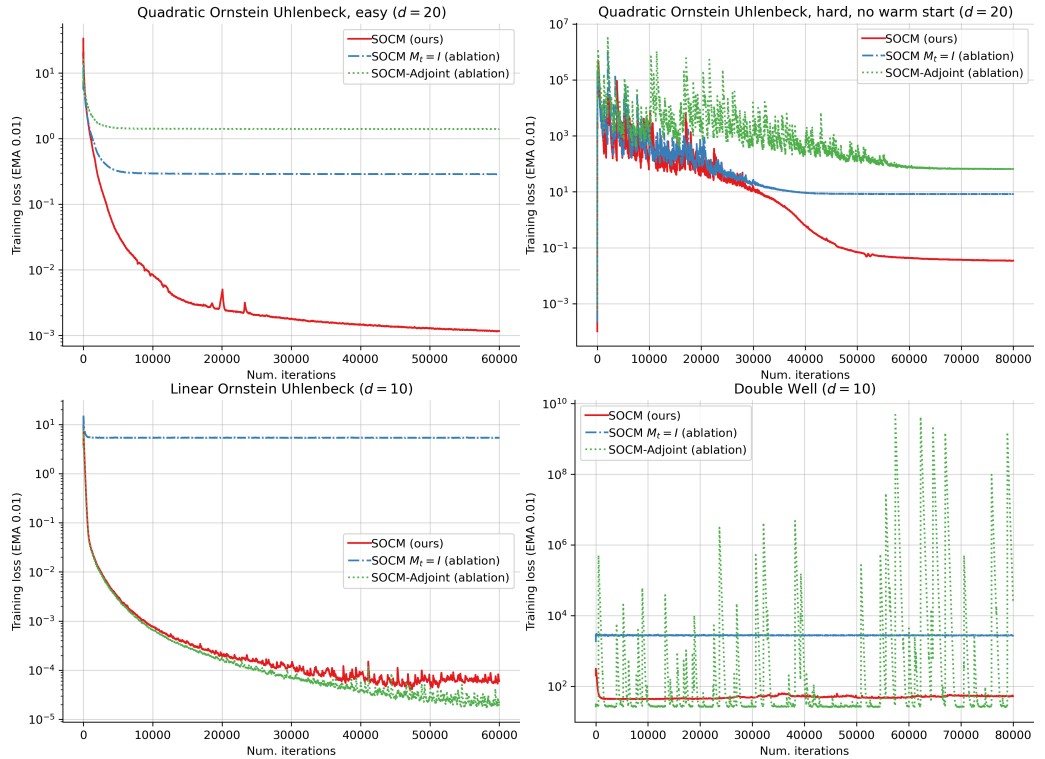

Figure 8: Plots of the training loss for SOCM and its two ablations: SOCM with constant $M_t = I$, and SOCM-Adjoint.

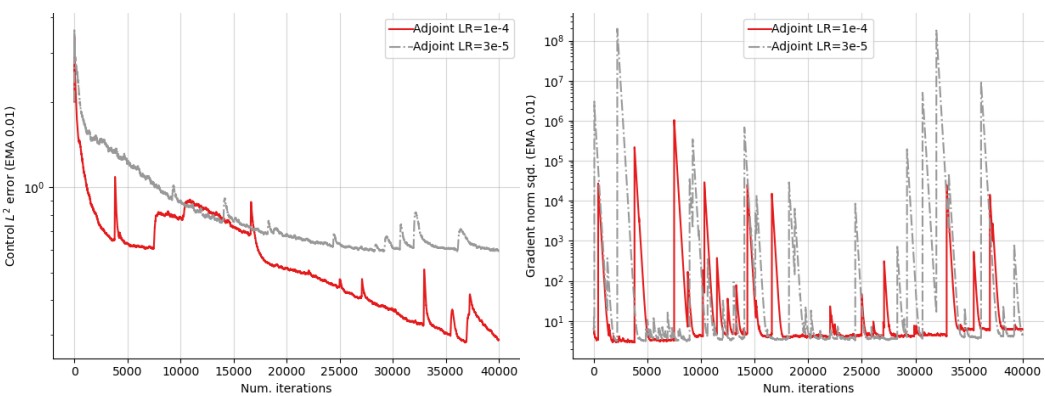

Figure 9: Plots of the control $L^2$ error and the norm squared of the gradient for the adjoint method on DOUBLE WELL, for two different values of the Adam learning rate. The instabilities of the adjoint method persist for small learning rates, signaling an inherent issue with the loss.

