# OpenReview forum: "Stochastic Optimal Control Matching"
_NeurIPS.cc/2024/Conference — NeurIPS 2024 poster_

### Official Review · Reviewer_Uos4 · 2024-07-10

**Soundness:** 4
**Presentation:** 3
**Contribution:** 4
**Rating:** 6
**Confidence:** 4

**Summary:**

In this paper, the authors propose a novel learning algorithm, Stochastic Optimal Control Matching (SOCM), to numerically solve general formulations of Stochastic Optimal Control (SOC) problems involving affined-controlled diffusion processes. They build upon the Iterative Diffusion Optimization (IDO) [1] framework, which consists in iteratively refining a parametric controlled diffusion process by minimizing at each iteration (with stochastic gradient descent) a specific objective function with respect to the parametric control. Previous works had for instance considered the relative entropy loss, the cross-entropy loss, the log-variance loss or the moment loss as the objective function. In SOCM, it is a least-squares regression loss (with multiplicative importance weight) which aims at fitting the parametric control to a vector field which depends on a family of *reparameterization matrices* (also optimized). The design of this objective function relies on standard tools from SOC theory, as well as an original contribution, the *path-wise reparameterization trick*, to compute gradients of conditional expectations of a functional applied on a random process. The authors show that the SOCM loss can be decomposed as the sum of a bias term, that is linked to the cross-entropy loss, and variance term, which is only affected by the *reparameterization matrices*. Hence, these extra parameters can be seen as a way to reduce the variance of the SOCM loss, which motivates their introduction. Moreover, this loss has the computational advantage to avoid computing gradients of the control along the path measure (which is the main drawback of the relative-entropy loss). Finally, the authors conduct numerical experiments to compare their approach to existing designs of IDO losses. They consider four different settings ($d\in \{10,20\}$) with access to the ground-truth control, which allows them to compute the $L^2$ error on the control. Using this metric, their results indicate better performance on most of the settings while maintaining a certain training stability.

[1] Solving high-dimensional Hamilton–Jacobi–Bellman pdes using neural networks: perspectives from the theory of controlled diffusions and measures on path space. Nüsken et al. 2021

**Strengths:**

- The paper is very well-written, it is a pleasure to read it. In particular, the authors pay attention to define their notation, introduce with clarity the SOC framework, state clear mathematical statements, recall (and prove) standard results from SOC theory, provide intuition on theoretical results (with details on the proof and meaningful comments). This is really good work.
- The relation to prior work is clearly well established: in particular, the comparison between the SOCM loss and the previous IDO losses (presented in Section 2.2) is well highlighted.
- This papers introduces an interesting contribution that may be applied beyond this framework (in particular, in the generative community where terms involving gradients of expectations often appear) : this is the path-wise reparameterization trick, which is proved to be decisive in the numerics.

**Weaknesses:**

- In my opinion, the major weakness of this paper is the lack of an additional numerical experiment, which represents a "realistic" setting (for instance, where the expression of the control is not known). For instance (as mentioned by the authors), a significant line of recent research has considered the sampling problem via a SOC perspective, see for example [1,2,3]. I am convinced that the SOCM contribution would have more impact with additional sampling numerics comparing SOCM, relative entropy [1,2] and log-variance [3] losses, for challenging distributions (namely, multi-modal distributions in relatively high dimension). In this case, the quality of the methods would be assessed with sampling metrics. This weakness explains my current score (although I really appreciate the paper).
- I find that the complexity/accuracy tradeoff of the IDO losses (including SOCM) is not well highlighted to me. The table provided by the authors only considers one setting. To have a full picture, it should be given for all settings.

[1] Path Integral Sampler. Zhang et al. 2022.

[2] Denoising Diffusion Sampler. Vargas et al. 2023

[3] Improving sampling via learned diffusions. Richter et al. 2023

**Questions:**

- Have you tried to restrict the optimization of the reparameterization matrices to scalar matrices ? I have the feeling that this choice may align a low computational budget with a good expressivity.
- Have you tried another parameterization of these matrices ? In particular, have you considered simple form such as $M_{w}(t,s)=I_d + \gamma(s-t)\tilde{M}_{\tilde{w}}(s,t)$ where $w=(\gamma, \tilde{w})$ and $\gamma(0)=0$ ? Otherwise, why the choice of the sigmoid ?
- I find that the warm-start strategy for the optimization of the control is a very good idea, as it benefits from the tractability of the stochastic interpolants with the lightness of the spline formulation. However, I have the feeling that this strategy works in the presented settings since they are "kind of Gaussian". Do you think it may still be of interest for general SOC problems ?
- Could you explain why the second Ornstein-Uhlenbeck setting is called 'hard' ?
- Could you provide the results on $L^2$ error of the control without EMA, as it is presented in [1] ?
- I am quite surprised of the relatively computational budget induced by the use of the log-variance loss, could you comment on this ?

[1] Solving high-dimensional Hamilton–Jacobi–Bellman pdes using neural networks: perspectives from the theory of controlled diffusions and measures on path space. Nüsken et al. 2021

**Limitations:**

The main limitation of the approach is discussed in Section 5: it is the variance of the importance weight in the SOCM loss, which may blow up.

---

> ### Author Rebuttal · Authors · 2024-08-06
>
> We thank the reviewer for providing an extremely succinct description of our paper. We completely agree with the reviewer’s concerns regarding limitations, but we just want to highlight our perspective that this paper proposes a rather novel way of thinking about constructing methods for SOC problems which we believe will have a big influence on future methods that do scale well to practical applications. That is, we put together this paper as it paints a complete picture of the new proposed approach, even if it is not yet a “silver bullet” that can solve all SOC problems, and chose to further develop scalable variants in a separate follow-up work. Below, we provide detailed responses to the reviewer’s questions.
>
> **“[...] the major weakness of this paper is the lack of an additional numerical experiment, which represents a "realistic" setting [...] I am convinced that the SOCM contribution would have more impact with additional sampling numerics [...]”**
>
> We benchmark SOCM against existing methods on control problems that have been studied before in the literature (Nüsken & Richter, 2021), which allows us to clearly focus our analysis on the loss function instead of other components of the problem. These are control problems where we have access to the optimal control, which allows us to track the control $L^2$ error. Alternatively, we would only be able to compute the control objective, which as shown in Figure 3 (pg. 35), is a less sensitive metric due to numerical errors.
>
> We also want to clarify that, as noted in the global response, the Double Well problem is far from a toy problem and is actually highly non-trivial due to its multimodal nature. We chose this problem because it is a representative challenging problem that appears in SOC interpretations of sampling problems, such as the path integral sampler. By exploiting the specific decoupling nature of the potential, we can reduce it to a 1D problem for reference solutions. We do not test SOCM on additional realistic control problems because it often has gradient variance issues due to the variance of the importance weight $\alpha$ (see Section 3 in the paper). We believe that the strength of our paper is that our framework is completely novel, and it will be the basis for the development of more algorithms that do scale well to realistic problems (ongoing work).
>
> **“I find that the complexity/accuracy tradeoff of the IDO losses (including SOCM) is not well highlighted to me. The table provided by the authors only considers one setting. To have a full picture, it should be given for all settings.”**
>
> In Table 1 we show the time per iteration for each loss for Quadratic OU (easy). For all losses and settings, almost all of the runtime is spent evaluating and backpropagating through the control neural network. Since the number of control computations per timestep for a given loss is the same across settings, and we are using the same neural network architecture for all settings, the time per iteration for all losses in a single setting is actually an accurate reflection of all the settings we’ve considered: the time per iteration for other settings is roughly proportional to the ratio of the number of timesteps that we are using. For SOCM, there is the additional cost of computing and backpropagating through the M function, but that also depends on the specific architecture that is used and can also be reduced by using sparse parameterizations.
>
> **“Have you tried to restrict the optimization of the reparameterization matrices to scalar matrices ? I have the feeling that this choice may align a low computational budget with a good expressivity.”**
>
> We have not tried to restrict the optimization of the reparameterization matrices to scalar or diagonal matrices, but we agree that this would allow us to trade off fast computation and low variance. We will add an experiment where we use scalar and diagonal reparameterization matrices, and we expect that it will simply be somewhere between full M and the M=I ablation result in terms of performance.
>
> **“Have you tried another parameterization of these matrices ? In particular, have you considered simple form such as $M_{\omega}(t,s) = I + \gamma(s-t) \tilde{M}_{\tilde{\omega}}(s,t)$ where $\omega = (\gamma, \tilde{\omega})$ and $\gamma(0)=0$? Otherwise, why the choice of the sigmoid ?”**
>
> We would like to clarify that in line 263, $\gamma$ is a scalar, not a one-dimensional function. Hence, arguably our form for $M_{\omega}$ is simpler than the one proposed by the reviewer. Still, the expression proposed by the reviewer makes sense and is a viable alternative.
>
> **“I find that the warm-start strategy for the optimization of the control is a very good idea [...] this strategy works in the presented settings since they are "kind of Gaussian". Do you think it may still be of interest for general SOC problems ?”**
>
> We agree with the reviewer's comment. Warm-starting the control is a strategy that makes sense when the following conditions hold simultaneously:
>
> (i) An arbitrary initialization for control neural network causes the importance weight $\alpha$ to have high variance. If the variance of $\alpha$ is already low for an arbitrary initialization, there is no need for warm-start. This is the reason that we do not use warm-start for Quadratic OU (easy), Linear OU and Double Well. For Quadratic OU (hard), we try both no warm-start (in the main text) and warm-start (in the appendix): not warm-starting causes the learning to happen much slower for algorithms that use importance weights, although SOCM is still the best-performing loss by the end of training.
>
> (ii) The warm-started control is close enough to the optimal control, such that the importance weight $\alpha$ has low variance. As the reviewer points out, this holds when the control problem is “kind of Gaussian” (unimodal), but it does not work when multimodality is present, as in the Double Well setting.

---

> ### Author Response · Authors · 2024-08-06
> **Rebuttal (2/2)**
>
> **“Could you explain why the second Ornstein-Uhlenbeck setting is called 'hard' ?”**
>
> We refer to the second OU setting as hard because for losses that rely on importance weights (SOCM, SOCM-Adjoint, Cross-Entropy), the importance weight $\alpha$ has a high variance due to the magnitude of the running and terminal costs being larger. When the importance weight $\alpha$ has a high variance, the signal-to-noise ratio for the gradients becomes small, making learning hard (at least initially). For the OU hard setting, SOCM still manages to outperform the other methods at advanced stages of training because as the learned control approaches the optimal control, the variance of $\alpha$ drops. We will show plots for an OU setting with even larger matrices for the costs of the problem (see answer to Reviewer gdJ8), and we will see that learning becomes impossible for SOCM.
>
> **“Could you provide the results on  error of the control without EMA, as it is presented in [1] ?”**
>
> We believe that showing EMA plots is more informative because their variance is lower. At later stages of training, the EMA value depends mostly on the previous 100 values (the EMA coefficient is 0.02), which means that the EMA value is very close to the actual value.
>
> **“I am quite surprised of the relatively computational budget induced by the use of the log-variance loss, could you comment on this ?”**
>
> In Table 1, we report that the variance loss takes 0.086 seconds per iteration, and that the log-variance loss takes 0.117 seconds per iteration. The computation for the two losses is very similar. We attribute the discrepancy to the speed fluctuation of the GPUs. We will run both loss training again and report an updated number.

---

> > ### Comment · Reviewer_Uos4 · 2024-08-08
> > **Answer to the rebuttal**
> >
> > First, I would like to thank the authors for providing precise answers to my questions and my comments, this is much appreciated. Second, I would like to re-emphasize that I acknowledge the high quality of their work, even if it is not a 'silver bullet' as they call it; I am still convinced that it can be the basis of future impactful works.
> >
> > Although the authors have addressed all the points I have raised, I would like to re-insist on the first one: the extension of the SOC formulation to sampling tasks (following the PIS/DDS framework), which has become very important in the sampling community. The authors explain that they **did not test SOCM on additional realistic control problems because it often has gradient variance issues due to the variance of the importance weight**. Without being as ambitious as considering "realistic" settings, have you tried to apply your method to sample from a Gaussian mixture with two modes in increasing dimension ? Does the "variance" issue appear in this synthetic setting ?

---

> > > ### Author Response · Authors · 2024-08-14
> > >
> > > We would like to thank the reviewer for their helpful comments. We have taken their suggestion into consideration, and present experimental results on two-mode Gaussian mixture sampling in increasing dimension, using the Path Integral Sampler [81]. Namely, we set $p_0 = \delta_{x_0}$, $b(x,t)=0$, $f(x,t)=0$, $T=1$, and $g(x)=\log (\mu^0(x)/\mu(x)) = - \|x\|^2/2 - d/2 \log (2\pi) - \log \mu(x)$, where $\mu$ is the density of a mixture of two Gaussians with means $\pm 1$ and variance $1$.
> > > Note that we take $\mu$ to be normalized, i.e. $\int \mu(x) dx = 1$, or equivalently, $\log Z := \log (\int \mu(x) dx) = 0$.
> > >
> > > For context, if we let $\hat{S}^u(X) = \int_0^T \frac{1}{2}\|u(X_t,t)\|^2 \, dt + \int_0^T \langle u(X_t,t), dB_t  \rangle + \log (\mu^0(x)/\mu(x))$, Theorem 4 of the path integral sampler paper states that $- \mathbb{E}[\hat{S}^u(X^u)] \leq \log Z = 0$, for any control $u$, and that equality holds when $u = u^*$. Hence, the quantity $\mathbb{E}[\hat{S}^u(X^u)]$, which we report in the “- ELBO” column, allows us to benchmark different SOC algorithms: the smaller the better. A perfect SOC algorithm would yield zero in this setting. We also track the regular Control Objective, which is equal in expectation to the negative ELBO, because $\mathbb{E}[ \int_0^T \langle u(X_t,t), dB_t  \rangle] = 0$ by the martingale property of stochastic integrals.
> > > We show the standard error for each quantity. We ran the experiments using the architectures described in the paper, for a total of 40000 iterations.
> > >
> > > | Algorithm | Dimension | Control Objective | - ELBO |
> > > |----------|----------|----------|----------|
> > > | Adjoint             | 2    | 0.00282 +/- 0.00321 | 0.00515 +/- 0.00041 |
> > > | SOCM             | 2    | 0.00317 +/- 0.00320 | 0.00541 +/- 0.00042 |
> > > | Cross-entropy | 2    | 0.00450 +/- 0.00320 | 0.00677 +/- 0.00046 |
> > > | Adjoint             | 8    | 0.01560 +/- 0.00435 | 0.01157 +/- 0.00058 |
> > > | SOCM             | 8    | 0.01495 +/- 0.00434 | 0.01104 +/- 0.00057 |
> > > | Cross-entropy | 8    | 0.01817 +/- 0.00433 | 0.01400 +/- 0.00064 |
> > > | Adjoint             | 16  | 0.02356 +/- 0.00548 | 0.01909 +/- 0.00075 |
> > > | SOCM             | 16  | 0.02242 +/- 0.00548 | 0.01802 +/- 0.00073 |
> > > | Cross-entropy | 16  | 0.03288 +/- 0.00544 | 0.02803 +/- 0.00091 |
> > > | Adjoint             | 32  | 0.04271 +/- 0.00726 | 0.03544 +/- 0.00102 |
> > > | SOCM             | 32  | 0.04013 +/- 0.00726 | 0.03287 +/- 0.00098 |
> > > | Cross-entropy | 32  | 0.07167 +/- 0.00718 | 0.06445 +/- 0.00138 |
> > > | Adjoint             | 64  | 0.07669 +/- 0.00991 | 0.06576 +/- 0.00141 |
> > > | SOCM             | 64  | 0.07143 +/- 0.00992 | 0.06150 +/- 0.00136 |
> > > | Cross-entropy | 64  | 2.90879 +/- 0.00816 | 2.91517 +/- 0.00846 |
> > >
> > > Cross-entropy, which uses the same importance weight $\alpha$ as SOCM, performs worse than the other two losses for all dimensions, and its results are particularly poor for dimension 64. This is because the variance of $\alpha$ is too large for learning to happen. In this case, we see that SOCM has better variance reduction than cross-entropy, despite both using importance weighted objectives for training. Note that $\alpha = \exp(...)$ where the exponent scales linearly with dimension (can be seen from Eq 20).
> > >
> > > We observe that the -ELBO for SOCM is slightly below that of Adjoint for most dimensions, which confirms that our method is better for this range of dimensions, but if we were to keep increasing the dimension, SOCM should eventually also fail due to higher variance of $\alpha$. We are still running the higher dimensions, and will also include a case where Adjoint overtakes SOCM.

---

### Official Review · Reviewer_cXHg · 2024-07-11

**Soundness:** 2
**Presentation:** 3
**Contribution:** 3
**Rating:** 5
**Confidence:** 3

**Summary:**

This paper proposes a novel algorithm for approximating the solution to the Hamilton-Jacobi-Bellman (HJB) equation with a neural network control policy. Rather than backpropagating through rollouts of the dynamics, the authors develop a least-squares objective which resembles the score-matching loss used in diffusion models. However, this requires computing gradients of a diffusion process with respect to its initial condition. To address this, the authors develop a novel path-wise reparameterization trick which relies on a family of reparameterization matrices. They show how to optimize these matrices to reduce the variance of the objective estimate. They demonstrate that their method obtains a lower error with respect to the ground-truth control on toy problems, sometimes by an order of magnitude.

**Strengths:**

- The proposed objective function acts as a form of variance reduction for the cross-entropy loss when solving stochastic optimal control problems.
- The novel reparameterization trick for estimating gradients of diffusion processes with respect to its initial condition may be more broadly applicable.
- On toy problems, their method appears to generally outperform other approaches in solving the HJB equation for the optimal controls.
- The paper is well organized and overall written well. It provides a thorough related work section and does a good job explaining the novelty and results.

**Weaknesses:**

- The evaluations only consider simple toy problems. Moreover, they only plot the L2 error with respect to the optimal control. However, this does not necessarily tell us about the actual task performance due to compounding errors.
- On the Double Well system, there is not a clear advantage compared to the variance loss and adjoint method. However, the authors do discuss how their method appears more stable than the adjoint-based ablation.

**Questions:**

- How do all the methods compare in terms of actual task performance?
- How do these methods perform on more realistic control problems?
- Why does the proposed method not work as well on the Double Well system compared to the variance baseline?

**Limitations:**

The authors discuss limitations to scaling the approach up to more challenging problems due to the variance of the importance weight.

---

> ### Author Rebuttal · Authors · 2024-08-06
>
> We thank the reviewer for their comments.
>
> **“The evaluations only consider simple toy problems. Moreover, they only plot the L2 error with respect to the optimal control. However, this does not necessarily tell us about the actual task performance due to compounding errors.” “How do all the methods compare in terms of actual task performance?”**
>
> We would appreciate it if the reviewer clarified what they mean by “compounding errors”. Regarding task performance, we would also like to note that in Figure 3 in Appendix F.3, we plot the control objective (the quantity in the right-hand side of eq. 12). In Figure 3 we see that the control objective is a much less sensitive metric than the control $L^2$ error: it is harder to benchmark the losses using the control objective. From the perspective of measures over processes, the control $L^2$ error and the control objective are two sides of the same coin: the control $L^2$ error can be seen as the KL divergence between the probability measure $\mathbb{P}^{u^*}$ of the optimally controlled process and the probability measure $\mathbb{P}^{u}$ of the process controlled by $u$. And up to a constant term, the control objective is the reversed KL divergence between the same pair of measures.
>
> We would also like to clarify that, as noted in the global response, the Double Well problem is far from a toy problem and is actually highly non-trivial due to its multimodal nature. We chose this problem because it is a representative challenging problem that appears in SOC interpretations of sampling problems, such as the path integral sampler. By exploiting the specific decoupling nature of the potential, we can reduce it to a 1D problem for reference solutions. However, our SOCM method does not utilize this strong prior knowledge and solves it as a generic multimodal high-dimensional problem.
>
> **“On the Double Well system, there is not a clear advantage compared to the variance loss and adjoint method. However, the authors do discuss how their method appears more stable than the adjoint-based ablation. [...] Why does the proposed method not work as well on the Double Well system compared to the variance baseline?”**
>
> The Double Well (Figure 2 right-side) setting is different from other ones in that the terminal cost has 1024 modes. Hence, in order to obtain a small $L^2$ error, it is necessary to learn the control well in all or almost all the modes. Each trajectory that we sample will visit at most a few modes. Assuming for simplicity that each trajectory visits a single mode, and grouping trajectories into batches, by the end of the 80000 iterations, the “effective” number of batches per mode is 80000/1024=78.125, which is too small to get errors close to zero. Yet, this setting is interesting because it shows the behavior of the adjoint loss under multimodality: its $L^2$ error has a decreasing tendency but it wavers substantially, which is undesirable. We attribute this poor behavior of adjoint to the lack of convexity of the problem and SOCM has a clear stability advantage.
>
> Regarding the comparison to the variance loss, note that the SOCM method converges significantly faster and the we find variance objective is a poor choice most of the time with this benchmark being the only exception. We believe SOCM initially learns very fast on this problem due to the training of M (can be seen if compared to the M=I ablation) but may find sub-optimal local minima whereas the variance method has higher variance but may have a chance of finding better local minima given enough training time.
>
> **“How do these methods perform on more realistic control problems?”**
>
> We benchmark SOCM against existing methods on control problems that have been studied before in the literature (Nüsken & Richter, 2021), which allows us to clearly focus our analysis on the loss function instead of other components of the problem.
>
> We want to clarify that, as noted in the global response, the Double Well problem is far from a toy problem and is actually highly non-trivial due to its multimodal nature. We do not test SOCM on more realistic control problems because it often has gradient variance issues due to the variance of the importance weight $\alpha$ (see Section 3 in the paper). We believe that the strength of our paper is that our framework is completely novel, and it will be the basis for the development of more algorithms that do scale well to realistic problems (ongoing work).

---

### Official Review · Reviewer_gdJ8 · 2024-07-12

**Soundness:** 3
**Presentation:** 4
**Contribution:** 4
**Rating:** 7
**Confidence:** 1

**Summary:**

This paper presents stochastic optimal control matching (SOCM), which is an iterative diffusion optimization for optimal control aiming to fit a matching vector field. The authors introduce a new loss function and address the analysis and design of a learning-based control method.

**Strengths:**

The work is nicely motivated in Introduction, showing the drawbacks of traditional works.  The proposed control method is supported by the uniqueness analysis of the control logic (Theorem 1) and the sophisticated design methods (Propositions 1 and 2). In the reviewer's understanding, they are technically correct.

**Weaknesses:**

As stated in Algorithm 2 below, reducing noise in the gradient is crucial for the presented algorithm.  This weakness is addressed by Lemma 1 and extensions.

**Questions:**

As stated in Introduction, the work is motivated by stabilizing the unstable training of conventional IDO, which comes from the non-convexity of the loss.  Could the authors comment and/or perform some motivating experiments to show the stability of the training by SOCM?  They can emphasize the contribution of this paper.

**Limitations:**

No limitations in this work.  This work is devoted to the theoretical analysis of control system design, and it does not directly bring a negative social impact.

---

> ### Author Rebuttal · Authors · 2024-08-06
>
> We thank the reviewers for their encouraging rating and for providing us the opportunity to clarify the key importance of the proposed method below.
>
> **“Could the authors comment and/or perform some motivating experiments to show the stability of the training by SOCM? They can emphasize the contribution of this paper.”**
>
> Firstly, SOCM is stable by construction because its loss is convex in function space, unlike the adjoint method, which is not. In particular, we see that in the Double Well setting (Figure 2 right), the adjoint-based methods are quite unstable and have large ups-and-downs in the control error during training.
> Secondly, we also introduce the free parameter $M$ in our novel path-wise reparameterization gradient which can significantly reduce variance, allowing us to easily outperform related importance-weighted methods such as the cross entropy method.
> We also ablated both of these two claims through our ablation experiments (correspondingly, these are the “SOCM-Adjoint” which uses adjoint method instead of our path-wise reparameterization gradient, and the $M=I$ ablation, which are shown in all experimental settings to lead to worse results).

---

> > ### Comment · Reviewer_gdJ8 · 2024-08-13
> >
> > Thank you for your response.

---

### Official Review · Reviewer_Su1g · 2024-07-12

**Soundness:** 3
**Presentation:** 3
**Contribution:** 3
**Rating:** 5
**Confidence:** 3

**Summary:**

**Summary**

This paper introduces Stochastic Optimal Control Matching (SOCM), a novel algorithm for solving stochastic optimal control problems. Key contributions include:

1. SOCM algorithm, adapting ideas from conditional score matching in diffusion models
2. A new "path-wise reparameterization trick" for gradient estimation
3. Theoretical analysis including a bias-variance decomposition
4. Empirical evaluation showing superior performance on 3 out of 4 benchmarks

SOCM learns a control function by fitting a matching vector field via least squares, while optimizing reparameterization matrices to reduce variance. The method is currently limited to linear Gaussian models and requires knowledge of certain parameters. Experiments demonstrate SOCM's effectiveness on theoretical benchmarks, outperforming existing methods in most cases. The paper provides a solid theoretical foundation but lacks exploration of real-world applications or non-linear systems.

**Strengths:**

The paper introduces Stochastic Optimal Control Matching (SOCM), a novel algorithm for solving stochastic optimal control problems. Its originality lies in adapting ideas from conditional score matching in diffusion models to the domain of optimal control. This creative combination represents an interesting cross-pollination between two active areas of research.

The quality of the theoretical work is notable. The authors provide a comprehensive mathematical foundation for their method, including detailed proofs and a novel "path-wise reparameterization trick". This theoretical rigor is a significant strength of the paper.

In terms of clarity, the paper is well-structured and clearly written. The authors effectively guide the reader from the problem formulation through the theoretical development to the empirical results. The use of illustrative examples and detailed appendices aids in understanding the complex mathematical concepts presented.

The significance of this work lies in its potential to improve the efficiency of solving stochastic optimal control problems. The empirical results, showing improved performance over existing methods on multiple benchmarks, underscore the practical impact of this approach. However, the significance is somewhat limited by the current restrictions to linear Gaussian models.

**Weaknesses:**

The primary weakness of this paper is its limited scope and applicability. The method is currently restricted to linear Gaussian models and requires knowledge of certain model parameters. This significantly narrows its potential impact on the broader field of stochastic optimal control. The authors should discuss potential approaches to extend SOCM to more general settings, such as nonlinear or non-Gaussian systems.

While the empirical results are promising, they are limited to theoretical benchmarks. The paper would be strengthened by including experiments on real-world problems or more complex simulated environments. This would help demonstrate the method's practical utility and potential for broader impact.

The scalability of the method is not thoroughly addressed. As the dimensionality of the problem increases, how does the computational complexity of SOCM compare to existing methods? A more detailed analysis of computational requirements and scaling properties would be valuable.

The comparison with existing methods, while showing SOCM's superior performance, could be more comprehensive. Including comparisons with the most recent state-of-the-art methods would provide a clearer picture of SOCM's relative performance in the current landscape of stochastic optimal control algorithms.

**Questions:**

1. How might SOCM be extended to handle nonlinear or non-Gaussian systems? Are there specific challenges you foresee in this extension?

2. The paper focuses on theoretical benchmarks. Have you considered applying SOCM to any real-world stochastic optimal control problems? If so, what challenges did you encounter or do you anticipate?

3. How does the computational complexity of SOCM scale with the dimensionality of the problem? Could you provide a more detailed comparison of computational requirements with existing methods?

4. The path-wise reparameterization trick is an interesting contribution. Could you elaborate on potential applications of this technique outside of stochastic optimal control?

5. The paper mentions that SOCM requires knowledge of certain model parameters. In practical scenarios where these parameters might not be known precisely, how sensitive is SOCM to parameter misspecification?

6. Have you explored the performance of SOCM in settings with sparse or noisy rewards, which are common challenges in reinforcement learning?

---

> ### Author Rebuttal · Authors · 2024-08-06
>
> We thank the reviewer for an accurate list of our contributions; however, we think there may be a misunderstanding regarding the scope of applications. We detail our responses to the reviewer’s concerns and questions below:
>
> **“The method is currently restricted to linear Gaussian models and requires knowledge of certain model parameters. [...] The paper mentions that SOCM requires knowledge of certain model parameters. In practical scenarios where these parameters might not be known precisely, how sensitive is SOCM to parameter misspecification?”**
>
> We are unsure about this question and we think there may be a misunderstanding. We would appreciate it if the reviewer could point us to the comment that they refer to.
>
> SOCM does not actually require any explicit assumptions on the model parameters or a linear-Gaussian model. To clarify our interpretation of this concern, the linear-Gaussian model assumption usually refers to state transitions $p(x_{t+h} | x_t, u_t)$ being Gaussian distributed with linear dependence on the state $x_t$ and independent control variables $u_t$. In the continuous-time formulation, the updates under a linear-Gaussian model would be of the form $dX_t = Ax_t + Bu_t + \sigma(t) dW_t$. However, in our formulation we use neural networks to model a control velocity field, resulting in non-linear updates. That is, the updates we have are of the form $dX_t = u_t(x_t) + \sigma(t) dW_t$, which includes the linear-Gaussian model but also generalizes to nonlinear dependencies.
> The only assumption SOCM actually makes is that the objective functional depends on the control velocity field only through a quadratic form (equation 1). This is required for the path-integral representation in Theorem 1 but this is a typical assumption made in nearly all stochastic optimal control methods. SOCM requires the same amount of knowledge as the existing methods: we just need to know the base SDE, the running cost and the terminal cost.
>
> **“How might SOCM be extended to handle nonlinear or non-Gaussian systems? Are there specific challenges you foresee in this extension?”**
>
> As we detail in the proof sketch of Thm. 1, SOCM relies on the path-integral representation of the optimal control (eq. 8). As far as we know, such path-integral representations hold for control systems with arbitrary base drift and linear dependency of the drift on the control, and it is a typical assumption made by works in the literature (e.g. Nüsken & Richter, 2021). If path-integral representations exist or are developed for more general control problems, we believe that our technique may be adapted to handle those.
>
> **“The paper focuses on theoretical benchmarks. Have you considered applying SOCM to any real-world stochastic optimal control problems? If so, what challenges did you encounter or do you anticipate?”**
>
> SOCM performs well when the importance weight $\alpha$ defined in eq. 20 has low variance, which holds when its exponent has low variance. In our experimental section, we show that when $\alpha$ has low variance, SOCM outperforms existing methods. However, when $\alpha$ has high variance, the signal-to-noise ratio of the stochastic gradient is low, and the performance of the algorithm is compromised (much like it happens for the existing cross-entropy method, which also contains the factor $\alpha$). We do not regard SOCM as the solution to solve all stochastic optimal control problems, but rather as a new perspective which can lead to the development of more algorithms (ongoing work).
>
> **“How does the computational complexity of SOCM scale with the dimensionality of the problem? Could you provide a more detailed comparison of computational requirements with existing methods?”**
>
> Table 1 in Section F.3 shows the time per iteration for each of the loss functions that we consider: SOCM takes 0.22 seconds, while the adjoint loss takes 0.169 seconds, and all other methods are around 0.1 seconds. The reason that SOCM takes longer is that there is an additional neural network that is trained: the M function. The second paragraph in Section F.2 describes the neural network we use for the M function. In our setup, the total cost of M evaluations per iteration is $O(d^2 K^2)$, where $d$ is the dimension of the control system and $K$ is the number of discretization steps. The dependency $O(d^2)$ stems from the fact that we parameterize the whole matrix $M$; an alternative would be to parameterize $M$ as a diagonal matrix, which would result in a cost $O(d)$ at the expense of higher gradient variance. To further reduce the variance while using more computational resources, it is also possible to choose a function $M$ that depends on the iterate $X_t$ (see Remark 2). We will provide more clarity regarding the computation cost in the paper.
>
> **“The path-wise reparameterization trick is an interesting contribution. Could you elaborate on potential applications of this technique outside of stochastic optimal control?”**
>
> The path-wise reparameterization trick can be used as a drop-in replacement of the adjoint method when computing gradients of functionals on ODE/SDE trajectories, either with respect to the starting point or with respect to neural network parameters. The path-wise reparameterization trick provides a natural way to enable variance reduction, thanks to the arbitrary function M. One can potentially apply this trick to any setting where neural ODEs/SDEs are used, and also to estimate scores of SDEs with arbitrary drifts.

---

> ### Author Response · Authors · 2024-08-06
> **Rebuttal (2/2)**
>
> **“Have you explored the performance of SOCM in settings with sparse or noisy rewards, which are common challenges in reinforcement learning?”**
>
> In our benchmark problem Linear Ornstein Uhlenbeck, we only assume there is a terminal cost/reward and no intermediate state cost/rewards. This is a type of sparse reward (only one per episode) setting that reinforcement learning typically studies. However, we think the main difference to typical reinforcement learning applications is that control problems often assume the state costs are differentiable, which allows the use of gradient-based / adjoint methods for solving control problems, which can significantly reduce the effect of the sparse or noisy reward problem.

---

> > ### Comment · Reviewer_Su1g · 2024-08-08
> > **Thanks and raise score**
> >
> > Thank you for your response to my comments. I have read your rebuttal and am happy to raise my score.

---

### Author Rebuttal · Authors · 2024-08-06

We thank the reviewers for their helpful comments. We would like to clarify and reemphasize our contributions, and to provide a global response to some issues that have been raised by multiple reviewers.

We acknowledge that our method has limitations due to the use of importance weighting and is not yet a "silver bullet" that can handle non-trivial SOC problems. However, it provides an unconventional view of SOC methods and provides a new direction to explore SOC methods as least squares objectives. That is, we believe this paper paints a complete picture in deriving a new framework for least squares objective, and scalable variants are better covered in a separate follow-up.

Our paper makes two key contributions:

(i) The formulation of least squares objectives that directly regress onto the optimal control, leading to the proposed SOCM objective. We have empirically found that SOCM easily outperforms existing importance-weighted objectives such as the popular cross entropy method (either in faster training, better final result, or both). Compared to adjoint-based methods, we find that SOCM exhibits more stable training (as can be seen on the Double Well) and often much lower control errors (as in OU Quadratic and OU Linear).

(ii) Our proposed path-wise reparameterization gradient is orthogonal to the SOCM objective and is a general method for computing gradients of cost functionals. In particular, the path-wise reparameterization gradient has a built-in variance reduction option in the form of the matrix $M$, which we see significantly improves convergence speed and performance. Both of these claims are closely ablated in our experiments (as “SOCM-Adjoint” and “M=I” in our experimental results).

A common point among reviewers is that the paper only considers toy control problems. The main reasons we chose such problems are that (i) they were used by (Nüsken & Richter, 2021) as a benchmarking suite and (ii) because we are able to compute (or closely approximate) ground truth solutions for the control function, and thus we can assess the $L^2$ error incurred when learning the control. The alternative metric is the control objective functional itself, which is much less informative due to numerical errors (see Fig. 3 in App. F.3).

Furthermore, there is (understandably) some misunderstanding about the difficulty of some of the benchmark problems as it is not emphasized in the paper. In particular, the Double Well problem is actually highly non-trivial, is multimodal, and is also closely related to the SOC interpretations of sampling problems such as path integral sampler (Ref. 81: Zhang & Chen, 2022). The only reason we can produce a "ground truth" control to compare to in this setting is that we use significant knowledge of the problem; we analytically reduce it to a 1D problem and apply numerical methods to solve this 1D problem. It is not a problem where we actually have the ground truth control in closed form.

We hope that these points help clarify the main concerns raised by reviewers. We also provide detailed responses to each reviewer separately.

---

### Decision · Program_Chairs · 2024-09-25

**Decision:**

Accept (poster)

**Comment:**

All reviewers agree that this paper presents two key contributions that are of significant value: (1) the formulation of least-squares objectives to perform regression onto the optimal control of a stochastic optimal control problem (inspired in conditional score matching form diffusion models) and (2) a novel reparameterization trick for gradient estimation, which has a built-in variance reduction component that can be more broadly applicable beyond SOC. The clarity, presentation and quality of this work is excellent, and as pointed out by several reviewers.

During the rebuttal, some initial clarification was needed about which specific class of SOC problems is considered in the work. This was clarified quickly: the work considers a special case of path-integral control problems (non-linear dynamics with additive cost) where state and controls have the same dimensionality.

The only weakness shared by reviewers is that experimental evaluation did not include a real-world realistic scenario. All reviewers agree that this is not a not a reason for rejecting the paper and that overall the strengths outweigh this weakness. I fully agree and thus recommend acceptance.

I do suggest authors to consider the following points to improve the manuscript for its final revised version:

- Emphasize the complexity of the double well task. This was done during the rebuttal and should be incorporated in the revised paper.
- Incorporate as much as possible the additional GMM experiment discussed with reviewer Uos4, including certain details to ensure fair comparison such as adding results using log-variance and another sampling metric [2]. Even if it is not a realistic real-world application, it definitely helps to better understand the variance reduction behavior of the proposed algorithm in relation to other methods.
- Although the paper provides a thorough related work section, the paper would improve if the authors relate their method with other very relevant papers such as [1][3] and [4]. In the latter, a method that bridges cross-entropy and the relative entropy (KL-control cost) is proposed to reduce the gradient variance in path-integral control problems.

[1] Improved sampling via learned diffusions. Richter et al. ICLR 2024

[2] Beyond ELBOs: A Large-Scale Evaluation of Variational Methods for Sampling. Blessing et al. 2024

[3] Denoising diffusion samplers. Vargas et al. ICLR 2023

[4] Adaptive Smoothing for Path-Integral Control. Thalmeier et al. JMLR 2020